# Mechanism for controlled assembly of transcriptional condensates by Aire

Yu-San Huoh[1,2,5], Qianxia Zhang ●[1,2,5], Ricarda Törner[2,3], Sylvan C. Baca[4], Haribabu Arthanari ●[2,3] & Sun Hur ●[1,2] ✉

Transcriptional condensates play a crucial role in gene expression and regulation, yet their assembly mechanisms remain poorly understood. Here, we report a multi-layered mechanism for condensate assembly by autoimmune regulator (Aire), an essential transcriptional regulator that orchestrates gene expression reprogramming for central T cell tolerance. Aire condensates assemble on enhancers, stimulating local transcriptional activities and connecting disparate inter-chromosomal loci. This functional condensate formation hinges upon the coordination between three Aire domains: polymerization domain caspase activation recruitment domain (CARD), histone-binding domain (first plant homeodomain (PHD1)), and C-terminal tail (CTT). Specifically, CTT binds coactivators CBP/p300, recruiting Aire to CBP/p300-rich enhancers and promoting CARD-mediated condensate assembly. Conversely, PHD1 binds to the ubiquitous histone mark H3K4me0, keeping Aire dispersed throughout the genome until Aire nucleates on enhancers. Our findings showed that the balance between PHD1-mediated suppression and CTT-mediated stimulation of Aire polymerization is crucial to form transcriptionally active condensates at target sites, providing new insights into controlled polymerization of transcriptional regulators.

Controlled protein polymerization underlies a diverse range of biological processes[1,2]. One family of protein domains that mediate polymerization is the CARD domains, which play important roles in cell death and inflammatory signaling[3]. CARD domains often self-polymerize into filaments, serving as a key mechanism to amplify upstream immune signals[3]. However, aberrant polymerization of CARD can lead to undesirable consequences such as chronic inflammation or toxicity[4,5]. Thus, tight regulatory mechanisms are required to ensure that CARD polymerization occurs only under specific conditions[3,4]. Despite their well-known roles in cytosolic signaling pathways, CARDs are understudied in transcriptional regulators (TRs) such as Aire and Speckle proteins[6,7]. Consequently, specific functions and mechanisms regulating CARD polymerization in transcription remain poorly understood.

Aire plays a critical role in central T cell tolerance[8]. Aire orchestrates the expression of thousands of peripheral tissue antigens (PTAs) in medullary thymic epithelial cells (mTECs)[8,9]. These PTAs are displayed on the mTEC cell surface to recognize auto-reactive T cells for their negative selection or diversion into regulatory T cells[8,10]. Consequently, mutations in human *AIRE* or knockout of mouse *Aire* result in multi-organ autoimmunity, including autoimmune polyendocrinopathy syndrome type 1 (APS-1)[8,11]. Initially, Aire was regarded as a conventional transcription factor (TF), directly binding PTA gene promoters to induce expression[12]. Recent studies, however, propose that Aire largely upregulates PTA expression indirectly by amplifying the actions of various lineage-defining TFs ectopically expressed in mTECs[13–15]. Precise modes of action

[1]Howard Hughes Medical Institute and Program in Cellular and Molecular Medicine, Boston Children's Hospital, Boston, MA, USA. [2]Department of Biological Chemistry and Molecular Pharmacology, Harvard Medical School, Boston, MA, USA. [3]Department of Cancer Biology, Dana-Farber Cancer Institute, Boston, MA, USA. [4]Department of Medical Oncology, Dana-Farber Cancer Institute, Boston, MA, USA. [5]These authors contributed equally: Yu-San Huoh, Qianxia Zhang. ✉e-mail: Sun.Hur@crystal.harvard.edu

and the underlying mechanisms for Aire's synergy with these TFs remain unclear.

Aire is a chromatin-binding TR, rather than a sequence-specific, direct DNA binder[16]. Although Aire possesses a putative DNA-binding domain, SAND (named after Sp100, AIRE-1, NucP41/75, DEAF-1), this domain lacks essential DNA-binding residues[17]. Instead, Aire features two potential chromatin reader domains—two plant homeo-domains (PHDs). PHD1 binds histone H3 (H3) with nonmethylated lysine 4 (H3K4me0), a histone state that is depleted from active loci but is abundantly present elsewhere[18,19]. Conversely, the second PHD (PHD2) exhibits little histone-binding activity, and its functions remain unknown[16,20]. At first, PHD1's specificity for H3K4me0 was thought to guide Aire to inactive PTA gene loci[21]. However, a study showed that Aire primarily binds genomic sites pre-enriched with the permissive histone mark, H3K27ac, notably at super-enhancers (SEs)[22]. Confoundingly, regions rich in active H3K27ac marks tend to lack H3K4me0[23], raising questions about how Aire specifically targets H3K27ac-rich sites and the precise role of PHD1 in this context.

One of the most intriguing properties of Aire as a TR is its ability to form nuclear condensates. These condensates are easily visualized by diffraction-limited light microscopy in both human and murine mTECs as well as in model cell lines[6,24–26]. A previous study showed that Aire forms homopolymers using its N-terminal CARD[24]. This homopolymerization correlates with Aire nuclear condensate formation[24], suggesting Aire homopolymers manifest as nuclear condensates. Additionally, Aire CARD can be functionally substituted with an orthogonal, chemically inducible multimerizing domain, preserving both condensate formation and transcriptional activity[24]. Notably, merely substituting with a dimerization or tetramerization domain is insufficient[24]. These observations, in combination with other reports[6,27], demonstrate the importance of Aire polymerization in condensate formation and transcriptional activity. However, while Aire condensate formation is necessary, it is not sufficient for transcriptional function. Equally critical appears to be the localization of Aire condensates, as Aire condensates associated with promyelocytic leukemia protein bodies lead to the loss of Aire transcriptional activity[24]. In fact, the precise locations and functions of Aire condensates remain unclear; whether Aire condensates form at Aire-bound genomic loci and serve as active transcription sites, inactive storage depots or suppressive compartments is unknown[22,28,29].

We here demonstrate that Aire condensates assemble on enhancers, serving as sites for transcriptional activation. Moreover, we reveal that these condensates are subject to intricate regulatory mechanisms ensuring tight coordination of CARD polymerization with genomic target recognition.

## Aire condensates form on enhancers to activate transcription

Aire is known to be expressed in a miniscule subset of mTECs with temporal dynamics[14,15,30], which has made mechanistic studies using mTECs challenging. Therefore, we generated a doxycycline (Dox)-inducible model system where Aire is ectopically expressed in a human thymic epithelial cell line (4D6) at levels matching human mTECs (Fig. 1a). In mTECs, the messenger RNA (mRNA) level of endogenous *AIRE* matches highly abundant mRNAs, such as ribosomal genes, *ACTB* and *GAPDH*, which we reproduced in our 4D6 cells. Our 4D6 system also recapitulated Aire localization to H3K27ac-enriched sites including SEs, CARD-dependent nuclear condensate formation, Aire-induced broad transcriptomic changes and the impact of loss-of-function APS-1 mutations (Fig. 1b,d,e and Extended Data Fig. 1a–c)—all characteristics previously observed in mTECs[12,22,26,30–32]. We thus utilized 4D6 cells to investigate the mechanism of Aire polymerization in the nucleus.

To characterize Aire's molecular functions, we first examined the impact of Aire at Aire-bound genomic regions by assessing the changes in both the steady-state (by bulk RNA sequencing (RNA-seq)) and nascent (by 5′-ethynyl uridine RNA sequencing (5EU-seq)) RNA levels upon Aire expression. Aire-bound versus Aire-free regions were defined as nucleosome-free regions (NFRs) (by assay for transposase-accessible chromatin with sequencing (ATAC-seq)) that were highly occupied by Aire versus regions lacking Aire chromatin immunoprecipitation followed by sequencing (ChIP–seq) signals (Supplementary Table 1). These Aire-bound and Aire-free regions were chosen to ensure comparable chromatin accessibility (Extended Data Fig. 1d). There was a global increase in both nascent and steady-state RNAs at Aire-bound loci upon Aire expression, but not in Aire-free regions (Fig. 1c, top two panels). Notably, Aire-mediated transcriptional induction was more pronounced and more focused around Aire-bound sites when examining the nascent RNAs compared to steady-state RNAs. Aire localization was primarily at active enhancers (Fig. 1b and Extended Data Fig. 1a), suggesting that Aire-mediated transcriptional activation occurs largely in the form of short-lived enhancer RNAs (eRNAs). Analyses of several loci, such as *RIC8A*, *SETD1B*, *UBTF* and *STX10*, which were all upregulated and bound by Aire, also showed marked increase in nascent transcripts upon Aire expression (Fig. 1d and Extended Data Fig. 1f). Intriguingly, Aire-dependent transcriptional activation at Aire-bound loci was not always accompanied by an increase in H3K4me1 or H3K27ac level as assessed by histone mark ChIP–seq. At the *RIC8A* locus, Aire induced spreading of H3K4me1 and H3K27ac, whereas at *SETD1B*, *UBTF* and *STX10* loci, no such changes were observed (Fig. 1d and Extended Data Fig. 1f). Global analyses of H3K4me1, H3K27ac and H3K4me3 ChIP–seq and ATAC-seq signals aggregated over Aire-bound sites also showed minimal increases in these histone

**Fig. 1 | Aire condensates form on enhancers and activate transcription.**
**a**, *AIRE* transcripts relative to other genes (*X*) in human *AIRE*⁺ mTECs (top) and Dox-inducible *AIRE*-expressing 4D6 cells (bottom). Bulk RNA-seq was performed on 4D6 cells 24 h post-*AIRE* induction and compared with previous single-cell RNA-seq data (*n* = 477 *AIRE*⁺ mTECs). Horizontal dashed lines denote relative expression = 1; horizontal solid red lines denote the median. **b**, Heatmaps of normalized Aire and H3K27ac ChIP–seq signals in 4D6 cells. Heatmaps are centered on Aire peaks (*n* = 1,363) and ranked by Aire ChIP–seq intensity. H3K27ac and Aire ChIP–seq experiments were performed on 4D6 cells without (pre-Aire) or with expression of WT human Aire-FLAG (+Aire), respectively. Unless mentioned otherwise, Aire indicates human Aire throughout the manuscript. **c**, Aire-induced changes in transcription or histone marks ±20 kb of Aire-bound and Aire-free NFRs (*n* = 542 and 658, respectively). **d**, Genome browser views of normalized ChIP–seq and RNA-seq profiles at exemplar Aire-bound sites in 4D6 cells. **e**, Nascent RNA-FISH coupled with IF images of 4D6 cells expressing Aire-FLAG. Cells were stained with anti-FLAG along with FISH probes targeting *RIC8A* and *SETD1B*. Yellow outlines represent nuclei boundaries. Left panels: nuclei containing Aire condensates with either *RIC8A* (top) or *SETD1B*

(bottom) RNA-FISH foci. Right panels: nuclei containing Aire condensates along with both RNA-FISH foci. See also Extended Data Fig. 2a. **f**, Quantitation of nuclei that have 0–3 RNA-FISH foci in 4D6 cells ±Aire as shown in **e** and Extended Data Fig. 2a. *n* indicates number of nuclei examined. **g**, Spatial relationship between RNA-FISH foci and Aire condensates. Shown are average signals of RNA-FISH (left), Aire IF centered on indicated FISH foci (center) or on random nuclear positions (right). The heatmap scale bars refer to intensity as arbitrary units (a.u.). *r*ₛ, Spearman's correlation coefficient between RNA-FISH and Aire IF signals. **h**, IF images of endogenous p300 in 4D6 cells that were not (−Aire) or were Dox-induced (+Aire) for 24 h before immunostaining. Right: line-analysis performed on a magnified nucleus enclosed by a white-dashed box. See also Extended Data Fig. 3a. Fluorescent intensity profiles (top right) correspond to the white line drawn across multiple Aire condensates. These profiles show that p300 and Aire intensities increase in the same locations, which are observed as yellow spots in the magnified merged image (bottom right). All data are representative of at least three independent experiments. CPM, counts per million; FC, fold-change.

marks or chromatin accessibility upon Aire occupancy (Fig. 1c and Extended Data Fig. 1e).

To examine whether the transcriptional activation of Aire-bound genomic regions occurs within Aire condensates, we performed immunofluorescence (IF) combined with nascent RNA-fluorescence in situ hybridization (FISH) to examine *RIC8A*, *SETD1B* and *UBTF* loci (Supplementary Table 2). In ~80% of the cells examined, Aire expression resulted in the appearance of one or two nascent RNA-FISH foci for

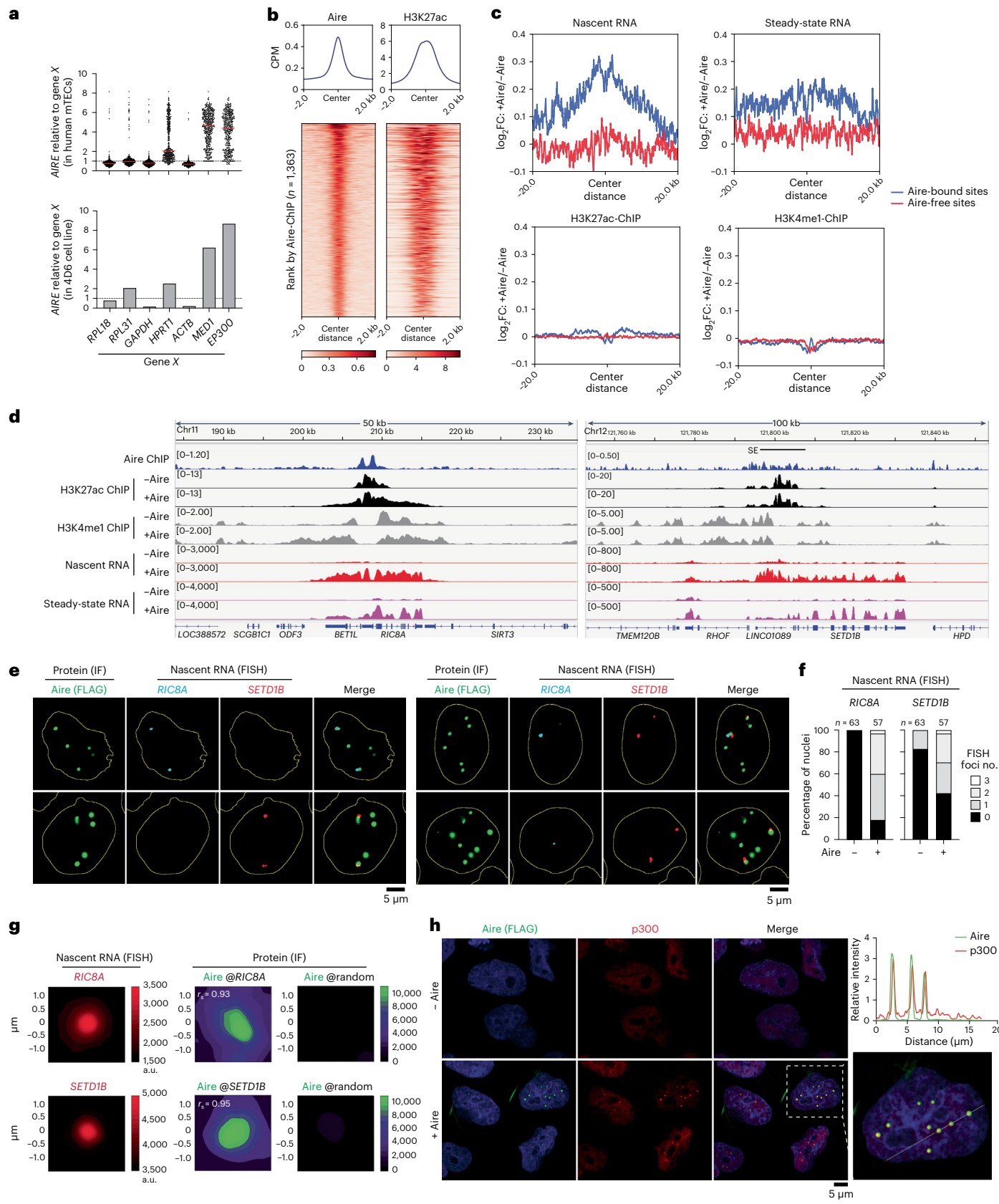

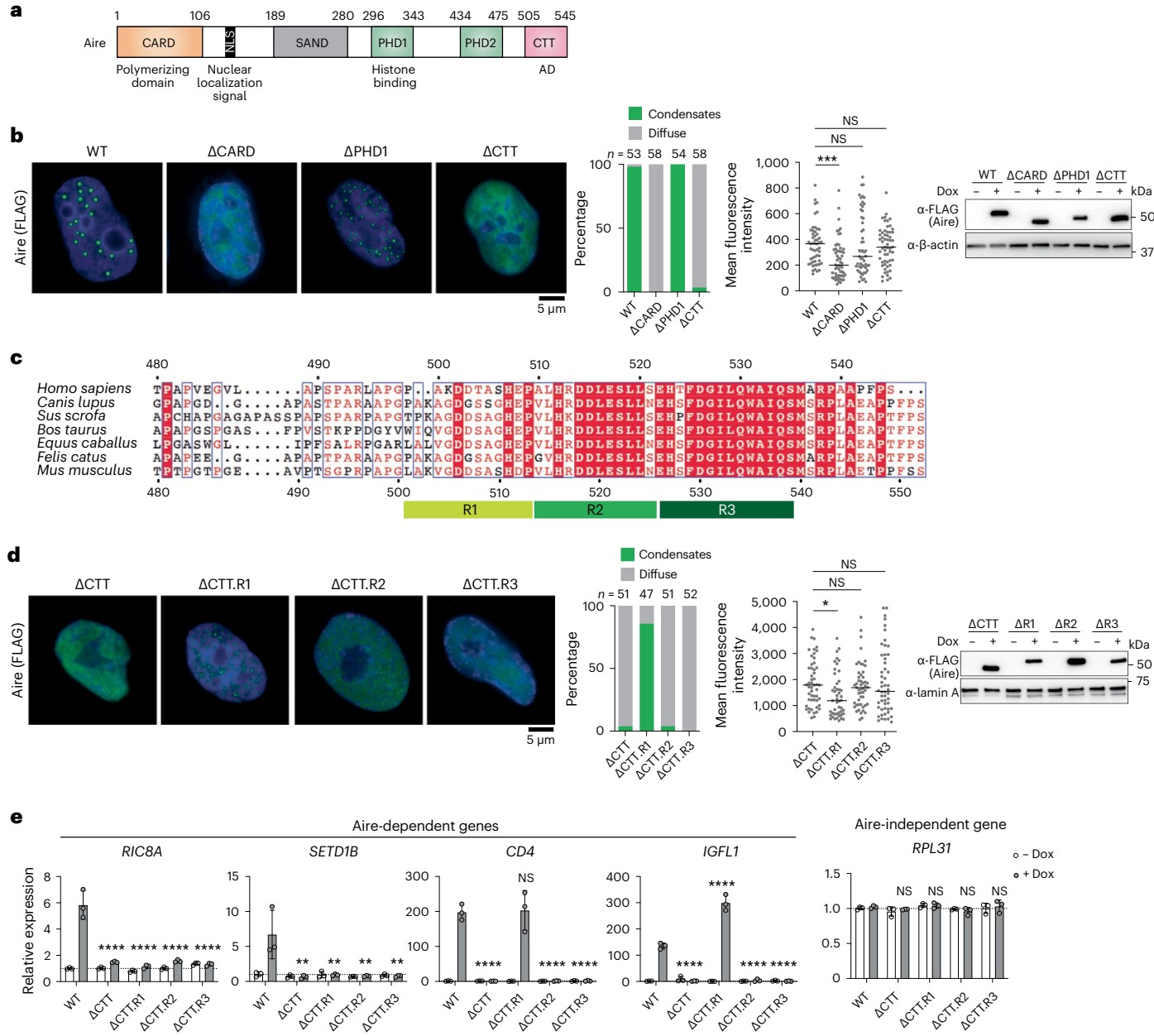

**Fig. 2 | Aire condensate formation requires the AD-like CTT. a**, Schematic of Aire domain architecture with previously characterized functions of individual domains. The numbers above denote amino acid residues of human Aire. **b**, Left: representative IF images of Aire WT, ΔCARD, ΔPHD1 and ΔCTT in 4D6 cells expressing Aire variants under a Dox-inducible promoter. Middle: percentage of nuclei with Aire condensates versus diffuse Aire staining, and mean fluorescence intensity of nuclei examined. *n* represents the number of nuclei examined. *P* values (Kruskal–Wallis test with Dunn's multiple comparisons test) were calculated in comparison with WT Aire. ***P < 0.001; P > 0.05 is NS. Horizontal lines indicate the median. Right: western blot (WB) showing the levels of FLAG-tagged Aire variants. **c**, Sequence alignment of Aire CTT domains from various species. CTT.R1, R2 and R3 correspond to amino acids 499–509, 510–521, 522–535 in human Aire, respectively. Numbers indicate amino acid residues of human Aire (numbering on the top) and mouse Aire (numbering on the bottom). **d**, Left: representative IF images of Aire ΔCTT, ΔCTT.R1, ΔCTT. R2 and ΔCTT.R3 in 4D6 cells expressing Aire variants under a Dox-inducible

promoter. Middle: percentage of nuclei with Aire condensates versus diffuse Aire staining and mean fluorescence intensity of nuclei examined. *n* represents the number of nuclei examined. *P* values (Kruskal–Wallis test with Dunn's multiple comparisons test) were calculated in comparison with AireΔCTT. *P < 0.05; NS *P > 0.05. Horizontal lines indicate the median. Right: WB showing the nuclear expression levels of FLAG-tagged Aire variants. **e**, Transcriptional activity of WT Aire or various CTT deletion mutants, as measured by the relative mRNA levels of Aire-dependent genes, *RIC8A*, *SETD1B*, *CD4* and *IGFL1*, in Dox-inducible 4D6 cells. An Aire-independent gene, *RPL31*, was examined as a negative control. All genes were normalized against the internal control *RPL18*. Horizontal dotted lines denote relative expression = 1. Data are presented as mean ± s.d. (three biological replicates). *P* values (one-way analysis of variance (ANOVA) with Dunnett's multiple comparisons test) were calculated in comparison with WT Aire (+Dox). **P < 0.01; ****P < 0.0001; NS *P > 0.05. All data are representative of at least three independent experiments. NS, not significant.

*RIC8A*, *SETD1B* and *UBTF* (Fig. 1e,f and Extended Data Fig. 2a,b). This is consistent with the transcriptional activation of these genes by Aire as measured with 5EU-seq and total RNA-seq (Fig. 1d and Extended Data

Fig. 1f). RNA-FISH foci and Aire condensates had significant overlap, as demonstrated by the averaged Aire fluorescence intensities centered on RNA-FISH foci and individual distance between the centers of the

closest RNA-FISH foci with Aire condensate (Fig. 1g and Extended Data Fig. 2c,d). Additionally, Aire condensates colocalized with coactivators p300, CBP and MED1 (Fig. 1h and Extended Data Fig. 3a). Similar condensates of p300 and CBP were not detected in the absence of Aire, although MED1 condensates were seen even without Aire (Fig. 1h and Extended Data Fig. 3a). These results together suggest that Aire condensates are indeed composed of genomic sites bound and activated by Aire.

Among all Aire-positive nuclei, 49% had both *RIC8A* FISH foci and *SETD1B* FISH foci within the same nucleus (Fig. 1e, right panels); within these nuclei, ~20% of the *RIC8A* and *SETD1B* FISH foci shared an Aire condensate with each other (Extended Data Fig. 2e). We observed a similar frequency of shared Aire condensate between *RIC8A* and *UBTF* FISH foci (Extended Data Fig. 2f). Although not obligatory pairings, the frequent contacts of *RIC8A*, *SETD1B* and *UBTF*, despite being located on different chromosomes, suggest that Aire clusters likely connect distinct inter-chromosomal loci to form transcriptional condensates.

## Aire condensate formation requires the activation domain CTT

We next examined the mechanism by which Aire forms condensates. Systematic domain truncation analysis showed that CARD was indispensable for Aire condensate formation (consistent with previous reports[6,24]), whereas SAND, PHD1 and PHD2 were dispensable (Fig. 2a,b and Extended Data Fig. 3b). Intriguingly, deletion of CTT (residues 482–545 in human and 480–552 in mouse) completely abrogated condensate formation for both human and mouse Aire (Fig. 2b and Extended Data Fig. 3b). While Aire CARD spontaneously polymerizes in vitro[24], isolated CTT behaves as a monomer. Solution nuclear magnetic resonance (NMR) of isolated CTT showed a $^{15}N$ $T_2$ relaxation time (Extended Data Fig. 3c) comparable to another monomeric activation domain (AD) of similar size[33]. Furthermore, unlike isolated CARD which forms nuclear condensates[24], isolated CTT (tagged with an artificial protein, APEX2-GST) did not show condensates in 293T cells (Extended Data Fig. 3d). These observations indicate that CTT may be modulating CARD polymerization, rather than directly participating in Aire condensate formation.

To determine which region within CTT is important for Aire condensate formation, we generated 4D6 cell lines expressing Aire CTT truncation mutants under a Dox-inducible promoter, as was done for wild-type (WT) Aire and other domain deletion mutants. Further truncation analysis suggests that residues 510–521 (R2, human residue numbering) and 522–535 (R3) within CTT were required for Aire condensate formation, but residues 499–509 (R1) of CTT were not (Fig. 2c,d). R2 and R3 also displayed higher sequence conservation than R1 and the rest of the CTT (Fig. 2c), suggesting that R2–R3 may have different functions from R1.

Aire CTT is known to have transcriptional AD-like activity[34]. We thus asked how the same CTT truncations affect AD-like activity. We measured the CTT variants' transcriptional activities by quantitative PCR (qPCR) with reverse transcription (RT–qPCR) of several Aire target genes. Deletion of R2 or R3 (ΔCTT.R2 and ΔCTT.R3) completely abolished Aire's transcriptional activity regardless of the examined target genes (Fig. 2e), recapitulating the loss-of-function phenotype observed with the complete deletion of CTT (ΔCTT). On the other hand, ΔCTT.R1 showed gene-specific behaviors, suggesting a more nuanced function for R1 (Fig. 2e). An AD reporter assay using CTT fused with Gal4 DNA-binding domain (Gal4[DBD]) also highlighted the importance of R2 and R3 in CTT's AD-like activity (Extended Data Fig. 3e), although R1 was also important in this reporter assay.

Collectively, these results suggest distinct functions for CTT R1 and R2–R3. R2–R3 contribute to both condensate formation and AD-like activity, whereas CTT R1 is involved in transcriptional activation of a subset of target genes, with minimal impact on condensate formation.

## Aire CTT binds CBP/p300 for functional condensate formation

To elucidate how CTT promotes Aire condensate formation and transcriptional activity, we investigated both the genetic and physical interaction partners of CTT. Based on our Gal4-based Aire CTT reporter assay, we designed a genome-wide CRISPR screen. This screen utilized a 4D6 cell line with the stable incorporation of the fluorescent reporter mKate2 under the control of upstream activation sequences (UASs). The mKate2 reporter was induced upon the expression of Gal4[DBD]-CTT fused with the expression reporter GFP via a self-cleavable peptide 2A (Fig. 3a and Extended Data Fig. 4a). We then collected GFP-positive cells that had decreased or increased mKate2 expression after transducing lentiviral single guide RNA (sgRNA) libraries and compared sgRNA enrichment between these two populations (Fig. 3b and Extended Data Fig. 4b). In parallel, we looked for physical Aire CTT interaction partners by performing GST pull-downs of 293T lysate using purified GST-tagged CTT (GST-CTT) and analyzed the co-purified proteins by mass spectrometry (MS) (Fig. 3c). From these two independent analyses, we identified the transcriptional coactivators CBP and p300 as the most significant common hits (Fig. 3b,c, Extended Data Fig. 4c and Supplementary Tables 3 and 4). CBP and p300 are highly homologous histone acetyltransferases (HATs) responsible for producing a large portion of H3K27ac in cells[35]. CBP/p300 colocalize with Aire condensates in both mTECs[6,36] and 4D6 cell line (Fig. 1h and Extended Data Fig. 3a). No other HATs were found in either screen.

In comparison with intact CTT, ΔCTT.R2 and ΔCTT.R3 showed reduced CTT–CBP/p300 interaction, whereas ΔCTT.R1 maintained similar binding levels (Fig. 3d). Thus, unlike R1, the importance of R2 and R3 in CBP/p300 binding correlated with their significance in Aire condensate formation. Using recombinant CTT and p300, we verified that the interaction was direct (Fig. 3e–g). Domain truncation analysis of p300 revealed that CTT interacted with the TAZ2 and IBiD domains of p300 (Extended Data Fig. 4d,e). Isolated CBP TAZ2 could be recombinantly purified and showed direct binding to purified CTT (Extended Data Fig. 4f), while isolated IBiD was insoluble, precluding more detailed binding analysis. Isothermal titration calorimetry (ITC) analysis showed that Aire CTT binds CBP TAZ2 with a $K_d$ of 0.26 μM (Fig. 3f), consistent with the TAZ2 affinity of other ADs[37,38]. Note that ITC also detected a second, low-affinity binding site (of 40 μM, Fig. 3f), which is presumably not as important as the high-affinity site.

To further characterize the TAZ2–CTT interaction, we performed $^1H$-$^{15}N$ heteronuclear single quantum coherence (HSQC) NMR spectroscopy on CTT with and without TAZ2. In the absence of TAZ2, isolated CTT displayed largely disordered characteristics (Extended Data Fig. 5a), although R2 and R3 are predicted to have a moderate tendency to form alpha helices (Fig. 3e and Extended Data Fig. 5b). Upon incubation with CBP TAZ2, R2 and R3 underwent substantial chemical shifts and peak broadening (Extended Data Fig. 5c,d), suggesting that R2 and R3 bind TAZ2. The utilization of R2 and R3 for TAZ2 binding was further supported by AlphaFold prediction (Fig. 3e). The AlphaFold model also predicted alpha helical conformations of R2 and R3, akin to the binding mode observed for other ADs[38,39]. Furthermore, mutations in the putative CTT interface (mouse Aire D530K and A536D, corresponding to human Aire D526 and A532) markedly reduced the CTT–CBP interaction, as measured by pull-down assay using purified proteins (Fig. 3g). This mutational analysis provides additional support for the NMR data and AlphaFold model.

We next examined whether CBP/p300 play an important role in Aire transcriptional activity and condensate formation. Transient expression of WT versus D530K or A536D showed that the mutations significantly lowered the transcriptional activities of Aire (Fig. 3h) and the AD reporter activity of Gal4[DBD]-CTT (Extended Data Fig. 6a). IF analysis also showed that A536D was diffuse in most cells (Fig. 3i). D530K formed condensates at a similar frequency as WT Aire, but

D530K condensates were smaller than WT condensates, irrespective of whether cells had similar Aire expression levels (as defined by similar nuclear intensities, Fig. 3i) or not (Extended Data Fig. 6b). These results show that both A536D and D530K are impaired in Aire condensate formation, albeit to differing degrees.

To further test the role of CBP/p300 in Aire functions, we used two CBP/p300 pharmacological inhibitors, the catalytic inhibitor A-485[40] and the degrader dCBP-1[41]. As expected, treatment with A-485 for 4 h reduced the levels of H3K27ac without diminishing the CBP/p300 protein levels, whereas dCBP-1 notably reduced the levels of both H3K27ac and CBP/p300 (Extended Data Fig. 6d). Global analysis of nascent RNAs (by 5EU-seq) showed that A-485 substantially reduced transcriptional activity at Aire-bound genomic regions, but not at Aire-free regions with comparable chromatin accessibilities (Fig. 3j and Extended Data Fig. 1d). 5′-ethynyl uridine quantitative PCR (5EU–qPCR) of Aire target genes showed that dCBP-1 also had a similar negative impact on transcription of Aire-bound loci (Extended Data Fig. 6c). IF of Aire with and without CBP/p300 inhibitors showed that both dCBP-1 and A-485 treatment significantly decreased the size of Aire condensates, all without affecting Aire expression level (Fig. 3k and Extended Data Fig. 6d). Furthermore, the smaller condensates visible with dCBP-1 and A-485 were transcriptionally inactive, as evidenced by the lack of nascent RNA-FISH foci of the Aire target gene, *RIC8A* (Extended Data Fig. 6e). Similarly, dCBP-1 and A-485 decreased the AD reporter activity of Gal4[DBD]-CTT, despite slightly increasing the level of Gal4[DBD]-CTT (Extended Data Fig. 6f). This inhibition was specific to CBP/p300, as inhibition of Brd4, which often cooperates with CBP/p300 for enhancer functions[42], did not inhibit Aire's transcriptional activity or condensate formation (Extended Data Fig. 6g,h), consistent with a previous report[43].

Overall, these results suggest that Aire CTT's role in Aire condensate formation and transcriptional activity are mediated by CBP/p300.

## CTT directs Aire to enhancers pre-enriched with CBP/p300

We next aimed to elucidate how the CTT–CBP/p300 interaction facilitates Aire condensate formation and transcriptional functions. We focused on Aire's binding to H3K27ac-rich enhancer loci, which are also known to have high CBP/p300 occupancy. We hypothesized that CBP/p300 clustering recruits Aire to these enhancers and nucleates Aire polymerization by increasing the local concentration of Aire molecules. To test this, we performed p300 ChIP–seq on Aire-inducible 4D6 cells before Aire expression to compare the 'pre-Aire' distribution of p300 with Aire distribution patterns. The result revealed a prominent enrichment of p300 at Aire ChIP–seq peaks, even before Aire expression (Fig. 4a). When we compared Aire ChIP–seq signals with pre-Aire CBP/p300 or pre-Aire H3K27ac ChIP–seq signals, we observed a closer

correlation between Aire and CBP/p300 occupancies than between Aire and H3K27ac occupancies (Fig. 4b, left panels). The discrepancy between CBP/p300 and H3K27ac can be explained by other HATs generating H3K27ac[35,44] and the fact that not all CBP/p300 occupancy leads to histone acetylation[45]. These observations suggest that CBP/p300 likely play a role in Aire's genomic targeting.

To investigate whether CBP/p300 guide Aire's target site selection, we performed ChIP–seq of ΔCTT.R3, which is deficient in CBP/ p300 binding and Aire condensate formation. ΔCTT.R3 strikingly exhibited a more dispersed genomic occupancy pattern than WT Aire, with approximately tenfold more Aire-bound peaks (1,363 for WT Aire versus 12,893 for ΔCTT.R3; Extended Data Fig. 7a) and relatively uniform ChIP–seq signal intensities (Fig. 4a). Moreover, the ChIP–seq signals of ΔCTT.R3 displayed a weaker correlation to CBP/ p300 ChIP–seq signals compared with those of WT Aire (Fig. 4b, middle panels). These distinctions became more apparent when examining the top 500 WT-preferred versus ΔCTT.R3-preferred sites (as defined by the WT-to-ΔCTT.R3 ChIP–seq signal ratio in Fig. 4a and Supplementary Table 5); WT-preferred sites exhibited a pre-existing high density of CBP/p300 occupancy, whereas ΔCTT. R3-preferred sites were largely devoid of CBP/p300 (Fig. 4c–e). Analysis of Aire occupancy at SEs also showed more enrichment of WT Aire compared with ΔCTT.R3 (Extended Data Fig. 7b), indicating that CTT contributes to SE preference. Furthermore, both CBP/ p300 degrader dCBP-1 and catalytic inhibitor A-485 impaired WT Aire binding to SEs (Fig. 4f,g), suggesting that Aire may preferentially home in on the catalytically active form of CBP/p300 that clusters at active genetic loci[46,47].

Altogether, these results demonstrate that Aire's localization to H3K27ac-rich regions is mediated by Aire CTT interaction with CBP/ p300. Moreover, our findings revealed that Aire condensate formation, transcriptional activity and genomic targeting all depend on CTT and CBP/p300. This strong association suggests a close coupling between Aire's genomic targeting and polymerization for functional condensate formation.

## CARD polymerization amplifies Aire preference for enhancers

To further investigate the relationship between Aire polymerization and genomic targeting, we performed ChIP–seq of ΔCARD, a variant with an intact CTT but impaired condensate formation and transcriptional functions (Fig. 2b and Extended Data Fig. 7c). We reasoned that if both Aire polymerization and CBP/p300 binding are essential for proper targeting, ΔCARD would fail to accumulate at CBP/p300-rich loci. Conversely, if polymerization is a consequence of genomic targeting, ΔCARD would show similar preference for CBP/p300-rich loci as WT Aire.

**Fig. 3 | Aire CTT directly binds CBP/p300 for transcriptional activity and condensate formation. a**, Top: schematic of Gal4-reporter-based CRISPR screening system in 4D6 cells. Bottom: system validation by flow cytometry. **b**, sgRNA enrichment in 4D6 cells with decreased (red) versus increased (blue) mKate2 expression (Supplementary Table 3 and Extended Data Fig. 4b). **c**, Purified His₆-GST and His₆-GST-CTT pull-downs of 293T nuclear extracts. Red arrow, location of ~250-kDa SDS–PAGE gel band cut and analyzed by MS (Extended Data Fig. 4c and Supplementary Table 4). **d**, His₆-GST and His₆-GST-CTT pull-downs of HA-p300 from 293T nuclear extracts. **e**, NMR results incorporated into the top AlphaFold-Multimer model using mouse CBP TAZ2 (aa 1764–1855) and mouse Aire CTT (aa 480–552). CTT is colored by NMR chemical shift perturbation with TAZ2. TAZ2 is colored by electrostatic potential (PyMol, APBS). **f**, ITC thermograms of CTT titrated into TAZ2 fit to a two-sites binding model. **g**, His₆-GST-CTT variant pull-downs of purified FLAG-p300. **h**, Transcriptional activity of mouse Aire variants in transfected 4D6 cells measured by RT–qPCR (mean ± s.d., three biological replicates). *P* values (one-way ANOVA with Dunnett's multiple comparisons test) were calculated versus

WT Aire. ****P < 0.0001; NS P > 0.05. **i**, Representative IF images of 4D6 cells expressing mouse Aire-FLAG variants. Bottom left: percentage of nuclei with Aire condensates versus diffuse Aire (*n*, nuclei). Bottom middle: mean fluorescence of Aire-expressing nuclei. Red-boxed nuclei (≤550 a.u.) were selected to compare condensate sizes with similar Aire expression. Bottom right: Aire condensates volume quantification in nuclei with similar mean fluorescence (*n*, condensates). *P* values (Kruskal–Wallis and Dunn's multiple comparisons test (bottom middle); two-tailed Mann–Whitney test (bottom right)) were calculated versus WT Aire. ***P < 0.001; ****P < 0.0001; NS P > 0.05 (Extended Data Fig. 6b). **j**, Aire-induced transcriptional changes in the presence of A-485 versus dimethylsulfoxide (DMSO) within ±20 kb of Aire-bound versus Aire-free NFRs measured by 5EU-seq. **k**, Aire condensate volume quantification with DMSO, dCBP-1 or A-485 (*n*, condensates). *P* values (Kruskal–Wallis test with Dunn's multiple comparisons test), ****P < 0.0001; NS P > 0.05 (Extended Data Fig. 6d). Horizontal lines in **i** and **k** indicate the median. **a**,**d**,**g**–**i**,**k** are representative of at least three independent experiments. FDR, false discovery rate.

Compared with WT and ΔCTT.R3, ΔCARD showed intermediate genomic targeting behavior (Fig. 4a–d). Specifically, ΔCARD's genomic occupancy correlated with p300 occupancy less than WT Aire, but more than ΔCTT.R3 (Fig. 4b, middle versus right panels). While ΔCARD favored WT-preferred sites over ΔCTT.R3-preferred sites, this preference was less pronounced than with WT Aire (Fig. 4c,d). Similarly, ΔCARD's SE localization was between that of WT and ΔCTT.

R3 (Extended Data Fig. 7b). Since both ΔCARD and ΔCTT lack condensate formation, but only ΔCTT lacks the preference for CBP/p300-rich sites, we infer that Aire's target site selection is primarily driven by CTT–CBP/p300 interaction. However, the stronger target bias exhibited by WT Aire compared with ΔCARD suggests that CARD-mediated polymerization amplifies the CTT-driven genomic preference for CBP/p300-rich sites.

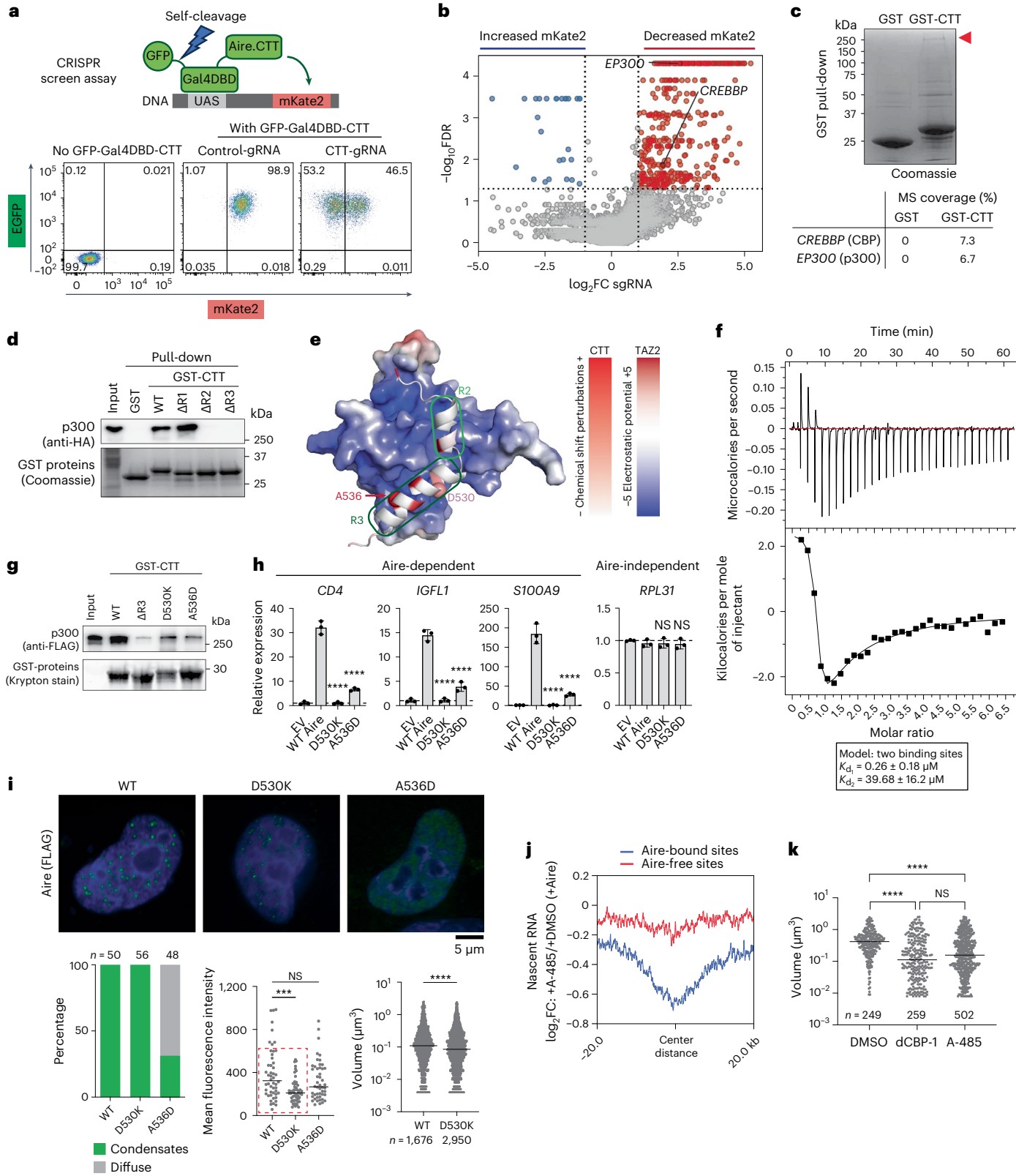

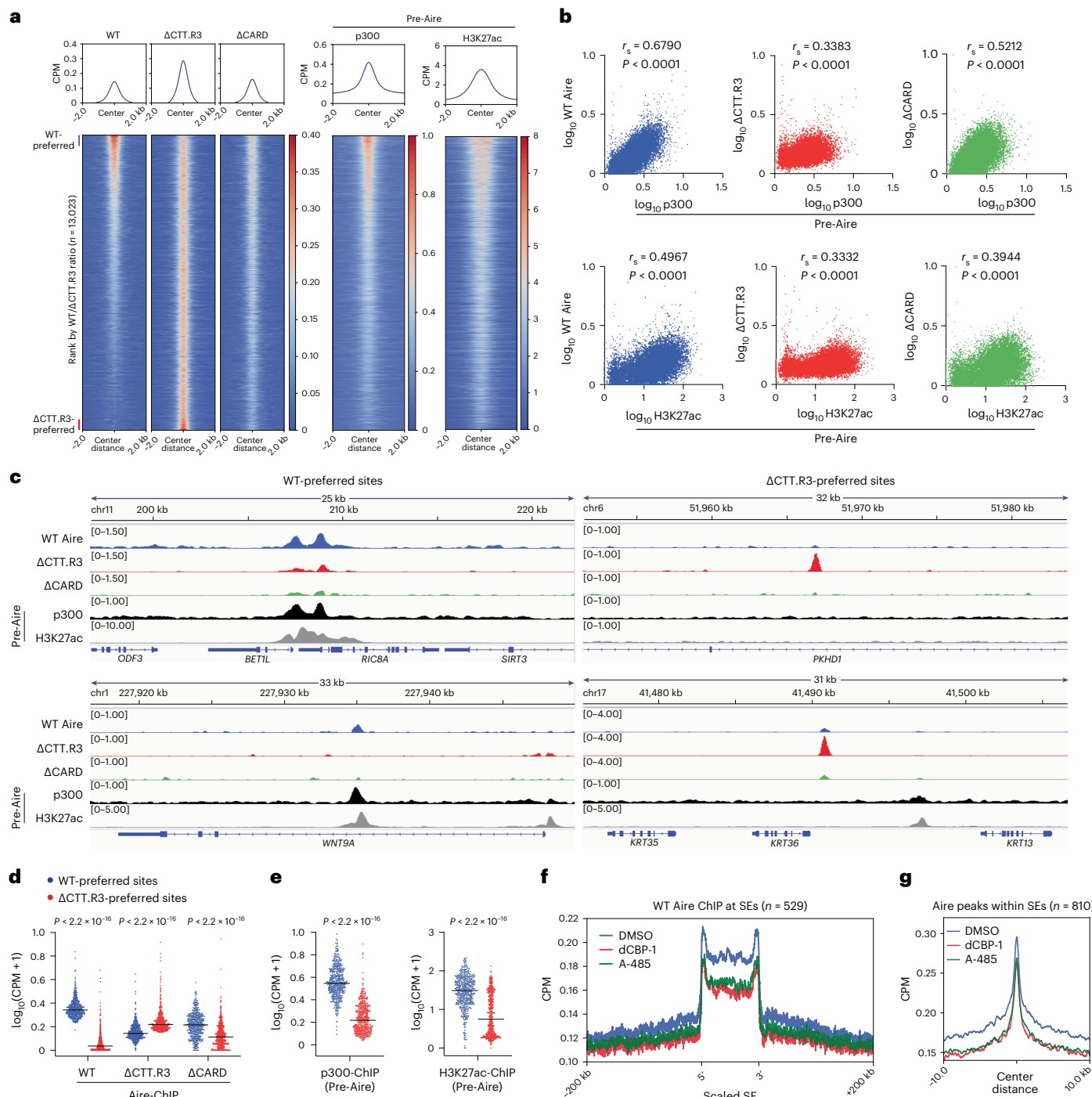

**Fig. 4 | CTT and CARD domains bias Aire towards CBP/p300-enriched enhancers. a**, Heatmaps of normalized ChIP–seq signals (CPM) for indicated proteins in Dox-inducible 4D6 cells. Heatmaps are centered on all Aire peaks (*n* = 13,023) and ranked by the ratio of WT Aire to ΔCTT.R3 ChIP–seq signals. Aire ChIP–seq signals were subtracted of background noise from corresponding input controls. p300 and H3K27ac ChIP–seq signals are from 4D6 cells before Aire expression (pre-Aire). **b**, Correlation between WT Aire, ΔCTT.R3 or ΔCARD with p300 (top panels) or H3K27ac (bottom panels) ChIP–seq signals at all Aire peaks (*n* = 13,023). $r_s$, Spearman's correlation coefficient. **c**, Genome browser views of normalized ChIP–seq profiles for indicated proteins at exemplar WT-preferred sites versus ΔCTT.R3-preferred sites in Dox-inducible 4D6 cells. Aire ChIP–seq signals shown were subtracted of background noise from corresponding input

controls. Numbers to the left of each panel indicate the ranges of normalized CPM for ChIP–seq. **d,e**, Quantification of normalized ChIP–seq signals (CPM) for Aire variants (**d**), p300 and H3K27ac (**e**) at WT-preferred sites versus ΔCTT. R3-preferred sites (*n* represents top and bottom 500 peaks from **a**, respectively). ChIP–seq signals were normalized using Trimmed Mean of M-values (Methods). *P* values were calculated using Wilcoxon rank sum test. Horizontal lines indicate the median. **f,g**, Contribution of CBP/p300 on Aire localization at SEs. 4D6 cells were treated with Dox for 24 h, and DMSO, dCBP-1 or A-485 was added 4 h before collection. Average Aire ChIP–seq profiles (normalized CPM) spanning ±200 kb of SEs (*n* = 529, defined in Extended Data Fig. 1a) are shown in **f**, or centered at Aire peaks located within SEs (*n* = 810) are shown in **g**.

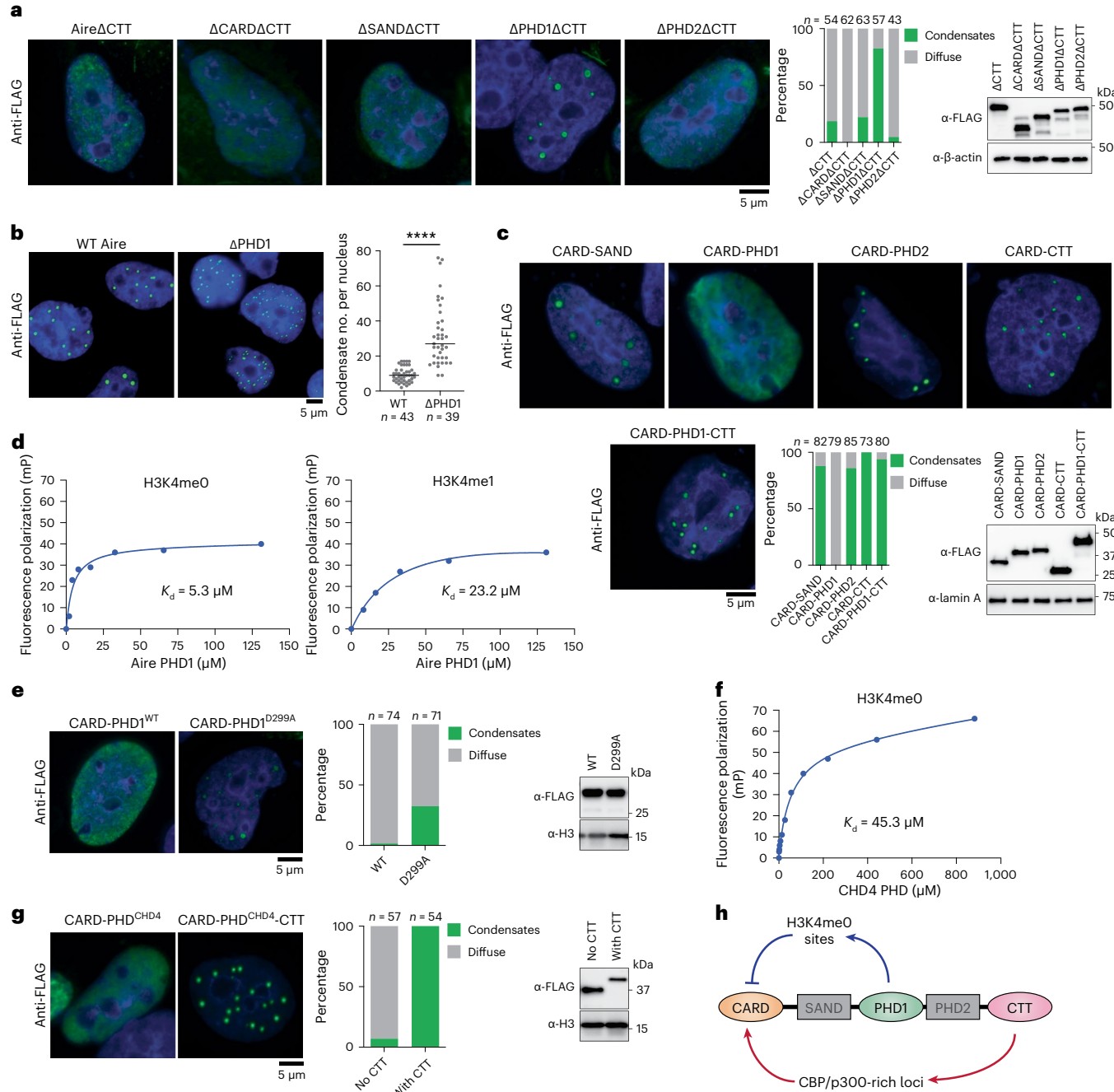

**Fig. 5 | PHD1-mediated chromatin interaction suppresses CARD polymerization. a**, Left: representative IF images of Aire domain deletion mutants in 4D6 cells. Cells were transfected with mouse Aire-FLAG expression plasmids 24 h before fixation. Middle: percentage of nuclei with Aire condensates versus diffuse Aire staining. Right: WB of FLAG-tagged Aire variants. **b**, Left: representative IF images of WT Aire and ΔPHD1 in Dox-inducible 4D6 cells. Experiments were done as in Fig. 2b. Right: quantification of number of Aire condensates per nuclei (*n*, nuclei). *P* value (two-tailed Mann–Whitney test) was calculated versus WT Aire. \*\*\*\**P* < 0.0001. Horizontal lines indicate the median. **c**, Top and bottom left: representative IF images of mouse Aire CARD fusion variants. Bottom middle: percentage of nuclei with Aire condensates versus diffuse Aire staining (*n*, nuclei). Bottom right: WB of FLAG-tagged Aire CARD fusion variants. Experiments and analyses were done as in **a**. **d**, Binding of indicated fluorescein-labeled H3 tail-derived peptides to increasing concentrations of mouse Aire PHD1. Binding is detected by change in fluorescence polarization (mP) of fluorescein. A representative experiment is shown for each peptide. Curves indicate the fit to a simple binding isotherm for the datasets shown. Data are presented as mean ± s.d.; H3K4me0

$K_d$ = 5.25 ± 2.59 μM; H3K4me1 $K_d$ = 23.18 ± 11.8 μM. **e**, Left: representative IF images of mouse Aire CARD fusion variants. Experiments and analyses were done as in **a**. Middle: percentage of nuclei with Aire CARD-PHD1 condensates versus diffuse staining (*n*, number of nuclei examined). Right: WB of FLAG-tagged Aire CARD-PHD1 variants. **f**, Binding of fluorescein-labeled H3 tail-derived peptide (H3K4me0) to increasing concentrations of the CHD4 PHD2. Data are presented as mean ± s.d.; $K_d$ = 45.3 ± 10.2 μM. **g**, Left: representative IF images of FLAG-tagged mouse Aire CARD fused with CHD4 PHD2 with and without Aire CTT. Experiments and analyses were done as in **a**. Middle: percentage of nuclei with Aire-CHD4 condensates versus diffuse staining (*n*, number of nuclei examined). Right: WB of FLAG-tagged Aire-CHD4 variants. **h**, Schematic showing the tight coordination between Aire's CARD, histone-binding domain PHD1 and AD CTT. PHD1 acts as a negative regulator by dispersing Aire to H3K4me0 sites across the entire genome, thereby 'diluting' Aire and preventing CARD polymerization at inappropriate locations. At Aire target sites, CTT recognizes CBP/p300-rich loci, concentrating Aire and promoting Aire CARD polymerization. All data are representative of at least three independent experiments.

A model to explain these findings is that Aire recruitment to CBP/p300-rich loci initiates CARD polymerization, attracting additional Aire molecules and thereby enhancing Aire's presence at the correct target site. The high density of CBP/p300 at Aire condensates and its absence without Aire (Fig. 1h and Extended Data Fig. 3a) suggests a feedback amplification that recruits additional CBP/p300 molecules to Aire nucleation sites. Altogether, while Aire's targeting to CBP/p300-rich loci does not strictly require CARD polymerization, polymerization amplifies the target preference.

## PHD1 suppresses CARD polymerization

Our results show that Aire requires CTT for condensate formation, likely by directing Aire to CBP/p300-rich enhancers. However, ΔCTT, despite having an intact CARD, is unable to form condensates. Since Aire CARD can spontaneously polymerize in vitro and form condensates in cells[24], yet cannot polymerize in the context of ΔCTT, we hypothesize that Aire CARD polymerization is subject to autoinhibitory mechanisms; CTT-mediated targeting then likely alleviates this suppression, ensuring controlled polymerization.

To identify the potential domain responsible for CARD regulation, we deleted individual domains in the AireΔCTT background, expecting that deletion of the CARD-suppressive domain would restore condensate formation. Only deleting PHD1 restored condensate formation of AireΔCTT (Fig. 5a and Extended Data Fig. 8a), suggesting that PHD1 is the domain responsible for suppressing Aire polymerization in the absence of CTT. In corroboration of PHD1 being the suppressor of Aire polymerization, deletion of PHD1 in full-length Aire significantly increased the number of Aire condensates per nucleus (Fig. 5b).

To examine whether PHD1 is sufficient to suppress Aire CARD polymerization, we directly fused PHD1 with CARD and compared CARD–PHD1 with CARD fused with other domains of Aire. Only CARD–PHD1 showed diffuse staining, whereas other fusion constructs formed condensates (Fig. 5c and Extended Data Fig. 8b). Furthermore, PHD1-mediated suppression in the CARD–PHD1 construct was relieved by additional fusion of CTT (Fig. 5c), recapitulating the suppressive effect of PHD1 and stimulatory effect of CTT in full-length Aire condensate formation. These results suggest that PHD1 is the suppressor domain for CARD polymerization and that the requirement of CTT in Aire condensate formation is due solely to PHD1-mediated suppression.

We next asked how PHD1 suppresses CARD polymerization. Because PHD1 is a histone-binding domain that specifically prefers H3K4me0[16,48] ($K_d$ = 5.25 ± 2.59 μM, Fig. 5d), we investigated whether PHD1 binding to H3K4me0 is important for inhibiting CARD polymerization. The D299A mutation in Aire PHD1, known to reduce the affinity for H3K4me0 and transcriptional activity of Aire[16,48,49], compromised PHD1's ability to suppress Aire CARD polymerization (Fig. 5e and Extended Data Fig. 8c). Additionally, fusing Aire CARD with another H3K4me0-specific PHD from an unrelated protein, CHD4[50] (Fig. 5f), suppressed Aire CARD polymerization, while CTT restored it (Fig. 5g and Extended Data Fig. 8d). Finally, the polymerization-suppressive effect of PHD1 was also conferred to an unrelated CARD from another TR, Sp110 (Extended Data Fig. 8e). These results suggest that Aire PHD1 suppresses CARD polymerization indirectly through H3K4me0 binding, rather than a direct PHD1–CARD interaction. Since PHD1's specificity for H3K4me0 is both necessary and sufficient for CARD suppression and H3K4me0 is widely distributed throughout the genome, we propose that PHD1 suppresses Aire polymerization by dispersing Aire across numerous genomic sites bearing H3K4me0. This dispersion would effectively dilute Aire and prevent its spontaneous nucleation. CTT, on the other hand, may counter this dilution effect by concentrating Aire at CBP/p300-rich loci, allowing target-specific polymerization (Fig. 5h).

## PHD1 suppression is required for functional Aire condensates

The role of PHD1 in suppressing CARD polymerization raised the question of whether PHD1 also negatively regulates Aire's transcriptional function. Since condensate formation is necessary, more frequent condensate formation by ΔPHD1 could amplify or augment Aire's function. Alternatively, given that condensate formation is not sufficient for transcriptional function, it is also possible that ΔPHD1 condensates are transcriptionally inactive. RT–qPCR of Aire target genes showed that ΔPHD1 was transcriptionally inactive (Fig. 6a). Nascent RNA-FISH also showed notably reduced transcriptional activity of ΔPHD1 despite forming more condensates per nucleus (Fig. 6b). Furthermore, ΔPHD1 condensates showed lower density of colocalized MED1 and CBP than WT Aire condensates (Fig. 6c). Similarly, other variants that bypassed PHD1-mediated regulation to form condensates (for example, ΔPHD1ΔCTT, CARD-CTT) were transcriptionally inactive (Extended Data Fig. 9a,b). These results show that Aire condensates formed in the absence of PHD1-mediated suppression are transcriptionally inactive.

To understand why ΔPHD1 is transcriptionally inactive, we analyzed ΔPHD1's interaction with chromatin. ΔPHD1 ChIP–seq revealed only seven ΔPHD1-occupied peaks, compared with 1,363 peaks for WT Aire, using the same peak calling and analysis pipeline (Extended Data Fig. 7a). When comparing ChIP–seq signals of ΔPHD1 versus WT Aire, ΔPHD1 showed remarkably lower signal at nearly all Aire-bound sites (Fig. 6d), indicating that PHD1 deletion completely abolished Aire's chromatin binding.

To confirm ΔPHD1's impaired chromatin interaction, we performed nuclear fractionation analysis, monitoring the solubility of chromatin and chromatin-associated proteins with and without MNase. In this assay, chromatin-binding proteins remain in the insoluble pellet (P) fraction with histones until MNase releases nucleosomes and associated factors into the soluble fraction (S). Both WT and ΔPHD1 co-fractionated with histones in the P fraction without MNase treatment (Extended Data Fig. 9c). Upon MNase treatment, a subset of WT Aire was released to the S fraction, indicating chromatin association. However, ΔPHD1 remained exclusively in the P fraction regardless of MNase treatment, suggesting impaired chromatin binding and formation of insoluble aggregates detached from chromatin (Extended Data Fig. 9c). Thus, the defect in transcriptional activity of ΔPHD1 condensates likely stems from a lack of chromatin tethering. These results show that while CTT and CARD modulate Aire's target specificity, PHD1 is necessary for Aire's association with chromatin.

How does PHD1, which prefers H3K4me0, promote Aire's interaction with chromatin at CBP/p300-rich enhancers typically marked by H3K4 methylation[23]? One possibility is that PHD1 facilitates Aire's chromatin localization indirectly by suppressing aberrant polymerization, which would render Aire incapable of chromatin binding. Alternatively, PHD1 may directly participate in target recognition by binding to lower affinity substrates, such as H3K4me1 near CBP/p300 clusters (Fig. 5d). Indeed, histone ChIP–seq showed that Aire's target sites (WT-preferred) displayed high levels of H3K4me1 nearby and low levels of H3K4me0 (Fig. 6e and Extended Data Fig. 9d for other histone marks). In contrast, nontarget sites, such as ΔCTT.R3-preferred sites, had higher levels of H3K4me0, but lower levels of H3K4me1 (Fig. 6e). Regardless of the specific mechanism, our results unambiguously demonstrate that PHD1-mediated suppression of spontaneous CARD polymerization is a critical intermediate step for the eventual formation of transcriptionally active Aire condensates.

## Discussion

Recent studies have highlighted the importance of biomolecular condensates in transcriptional regulation[47,51]. While studies on transcriptional condensates have largely emphasized the identification of multimerization domains, understanding of how multimerization is regulated to ensure proper localization and prevention of aberrant

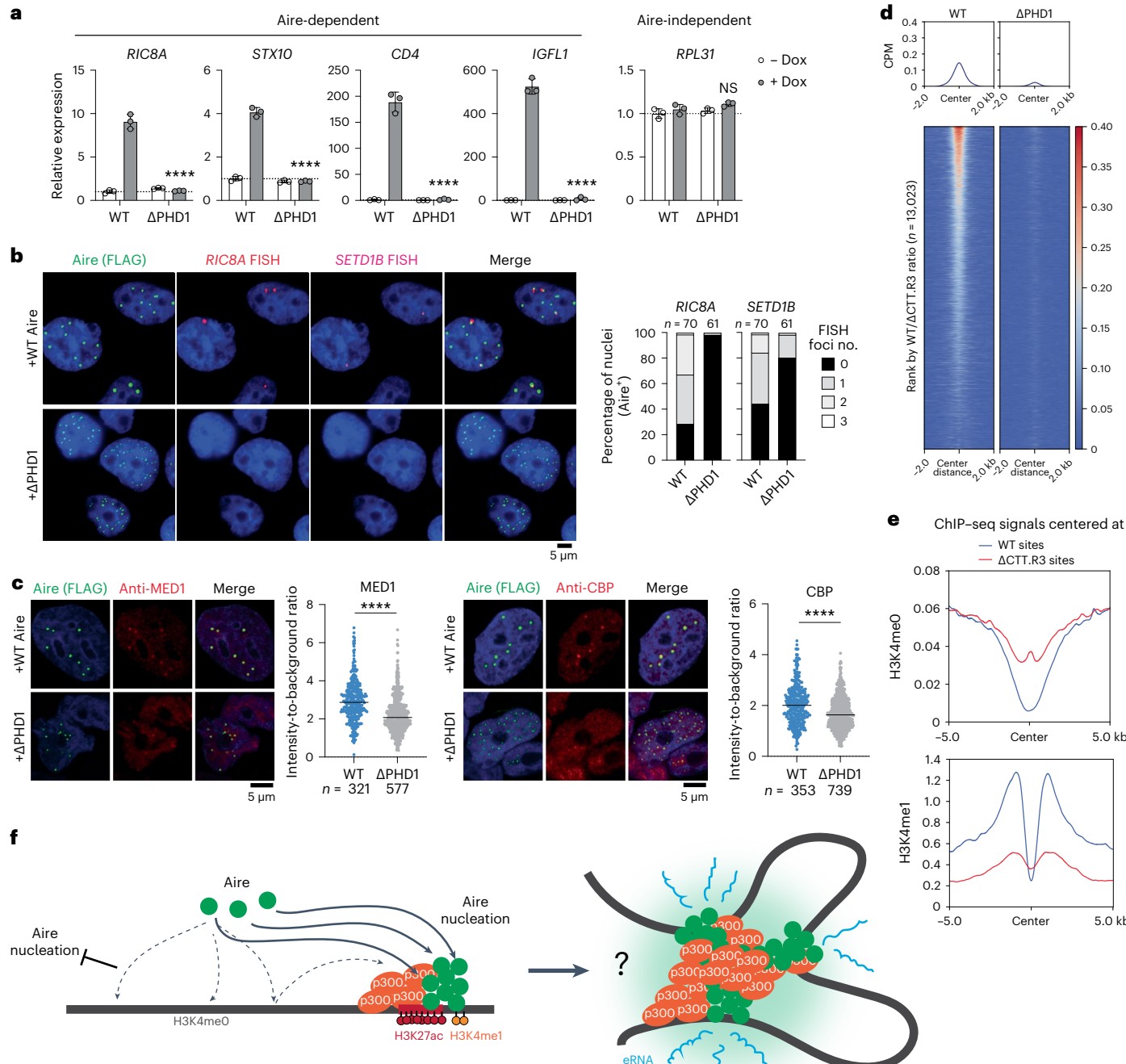

**Fig. 6 | PHD1-mediated CARD suppression is necessary for transcriptionally active condensate formation. a**, Transcriptional activity of Aire WT or ΔPHD1 in 4D6 cells as measured by RT–qPCR (mean ± s.d., three biological replicates). *P* values (two-tailed unpaired *t*-test) were calculated versus WT Aire (+Dox). ****$P < 0.0001$; NS $P > 0.05$. Horizontal dotted lines denote relative expression = 1. **b**, Left: images of WT Aire or ΔPHD1 IF with nascent RNA-FISH in 4D6 cells as in Fig. 1e. Right: percentage of nuclei that have 0–3 RNA-FISH foci (*n*, nuclei). **c**, Left and middle-right: IF images of endogenous MED1 and CBP in 4D6 cells expressing WT Aire or ΔPHD1. Right and middle-left: average intensities of MED1/ CBP at Aire condensates normalized to the average MED1/CBP intensity within the entire nucleus (*n*, condensates). ****$P < 0.0001$ (two-tailed Mann–Whitney test). Horizontal lines indicate the median. **d**, Heatmaps of normalized ChIP– seq signal (CPM) for Aire WT versus ΔPHD1 in 4D6 cells centered on all Aire peaks (*n* = 13,023) and ranked as in Fig. 4a. For clarity, WT ChIP–seq heatmap is reproduced from Fig. 4a. **e**, Average H3K4me0 and H3K4me1 ChIP–seq profiles (normalized CPM) centered at WT-preferred sites versus ΔCTT.R3-preferred sites (*n* represents top and bottom 500 peaks from Fig. 4a, respectively). Both H3K4me0 and H3K4me1 ChIP–seq experiments were performed on

4D6 cells before Aire expression. **f**, Proposed mechanism for controlled Aire condensate assembly. Aire employs PHD1's affinity for ubiquitous unmethylated H3K4 to interact distributively with chromatin, preventing inappropriate CARD polymerization outside target enhancers. Aire then utilizes CTT for specific targeting to CBP/p300-rich loci, countering PHD1's dispersion effect. Targeted accumulation of Aire at CBP/p300-rich loci initiates a positive feedback loop, nucleating Aire CARD polymerization and recruiting additional Aire molecules, thereby reducing the amount of dispersed Aire throughout the genome. Aire polymerization also recruits additional CBP/p300 molecules and connects multiple Aire-bound loci, ultimately leading to the formation of condensates highly enriched with Aire and CBP/p300. This conglomerate of Aire and CBP/p300 creates a potent environment capable of activating eRNA transcription without necessarily altering the levels of H3K27ac or chromatin accessibility. In summary, the coordinated action of PHD1-mediated 'regulation-by-dispersion' and CTT-mediated targeting is essential for functional Aire condensate formation. Data in **a**–**c** are representative of at least three independent experiments.

condensate formation remains limited. Such regulation is particularly important for CARD-containing TRs, which can spontaneously polymerize. Controlled polymerization is also crucial for CARD-containing proteins involved in cytoplasmic cell death and inflammatory signaling pathways, yet these regulatory mechanisms are still poorly understood.

We here reveal a multi-layered regulatory mechanism enabling Aire to form condensates at appropriate target sites, namely CBP/p300-rich enhancers. Aire first utilizes PHD1 to limit spontaneous polymerization by binding to H3K4me0. Loss of PHD1 results in more frequent and uncontrolled formation of condensates that are detached from chromatin, thereby rendering Aire transcriptionally inactive. By interacting with ubiquitously present H3K4me0 in the genome, PHD1 allows Aire to bind chromatin in a distributive manner and prevents CARD polymerization until the appropriate nucleation signal is recognized at correct target sites. This nucleation signal is provided by CBP/p300 clusters pre-localized at enhancers, which recruit Aire through Aire CTT, countering PHD1-mediated suppression and enabling target site-specific Aire polymerization. Once CARD polymerizes, an Aire nucleation site acts as a sink to recruit more Aire and CBP/p300 molecules, creating a positive feedback loop for transcriptional condensate assembly and consolidating multiple enhancers into a more potent transcriptional 'hub' (Fig. 6f).

The precise coupling mechanism linking CBP/p300 to CARD polymerization remains unclear. One possibility is that Aire's focused recruitment to CBP/p300 clusters increases the local concentration of Aire, overriding the PHD1-mediated 'dilution' effect and facilitating CARD nucleation. It is important to note that CTT's role in stimulating Aire polymerization is via its interaction with CBP/p300, not by directly participating in polymerization as for other ADs or intrinsically disordered regions[47,52]. Additionally, CTT is necessary for Aire polymerization only in the presence of PHD1-mediated suppression. PHD1-mediated chromatin anchoring makes Aire polymerization strictly dependent on CTT and restricted to activated CBP/p300-rich environments, such as SEs.

Questions arise as to how Aire's role in consolidating and potentiating already active enhancers contributes to its known biological function of inducing PTAs, which are silent without Aire and are not strongly occupied by Aire. We suspect that Aire's actions of forming transcriptional hubs (as described in this paper) are mechanistically distinct from its effect on PTAs. That is, Aire may access PTAs, either directly or indirectly, through mechanisms independent of CBP/p300, but dependent on other factors, such as Z-DNA or DNA breaks at PTA promoters[53], or other tissue-specific TFs[14,15]. This process may bring inactive PTA loci to Aire hubs by chance, thereby leading to the stochastic activation of PTAs[54]. Alternatively, Aire's ability to bolster mTEC enhancers may affect the differentiation of Aire-positive mTECs into mimetic cells, thereby indirectly promoting PTA expression[14,15]. More studies are needed to understand the precise functions of Aire in mTEC development and PTA expression. In summary, our mechanistic studies lay the groundwork for investigating how Aire utilizes controlled polymerization to establish central tolerance in T cell immunity.

## Online content

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

## Methods

### Expression vectors

Throughout the Methods, Aire indicates human Aire, unless mentioned otherwise. See the Supplementary Note for information on all expression vectors used in this study.

### Cell culture and transfection

Sources of cell lines were generous gifts from Dan Stetson (University of Washington) and Diane Mathis (Harvard Medical School). 293T cells were maintained in DMEM supplemented with 10% FBS, 1% L-glutamine. 293T cells were transfected with polyethyleneimine (3.75 µg per well of a 6-well plate with 1.5 µg of DNA) or Lipofectamine 2000 (Invitrogen; 1 µg of DNA per well of a 12-well plate) according to the manufacturer's protocol. 4D6 cells were maintained in RPMI supplemented with 10% FBS, 1% L-glutamine, and transfected with Lipofectamine 2000 or 3000 (Invitrogen) according to the manufacturer's protocol (1 µg of DNA per well of a 12-well plate).

For inhibitor treatments, 4D6 cells were Dox-induced for indicated amounts of time. At 4 h before collection, Dox-treated cells for different assays and an equal volume of DMSO, A-485 (3 µM final concentration, Tocris Bioscience), dCBP-1 (0.25 µM final concentration, MedChemExpress), dBET6 (100 nM final concentration, MedChemExpress) or JQ1 (1 µM final concentration, Selleck Chemical) were added.

Stable 4D6 cell line generation is described in detail within the Supplementary Note. For each 4D6 clone, a Dox titration curve was used to determine the appropriate Dox concentrations to use to have similar expression levels of Aire-FLAG variants compared with WT Aire-FLAG expression with 1 µg ml⁻¹ Dox. We used 1, 0.1, 0.1, 0.1, 0.05, 1 µg ml⁻¹ Dox on 4D6 clones expressing Aire-FLAG K83E, G228W, C311Y, ΔCARD, ΔPHD1, ΔCTT, respectively. We used 1, 1 and 0.1 µg ml⁻¹ Dox on 4D6 clones expressing Aire-FLAG ΔCTT.R1–R3, respectively. For ChIP–seq experiments with 4D6 cells treated with dCBP-1 for 4 h, 0.55 µg ml⁻¹ Dox was used to induce the same expression levels of WT Aire as 4D6 cells treated with 1 µg ml⁻¹ Dox + DMSO.

### Antibodies

Antibodies used for IF microscopy were mouse anti-FLAG (M2, Sigma-Aldrich, cat. no. F1804), mouse anti-FLAG conjugated with FITC (M2, Sigma-Aldrich, cat. no. F4049), rabbit anti-p300 (D8Z4E, Cell Signaling Technology, cat. no. 86377S), rabbit anti-CBP (D6C5, Cell Signaling Technology, cat. no. 7389S), rabbit anti-MED1 (Novus Biologicals, cat. no. NB100-2574), Alexa Fluor 488 AffiniPure donkey anti-mouse IgG (Jackson ImmunoResearch, cat. no. 715-545-151), Alexa Fluor 647 AffiniPure donkey anti-rabbit IgG (Jackson ImmunoResearch, cat. no. 711-605-152). Antibodies used for immunoblotting were rabbit anti-beta-actin (Cell Signaling Technology, cat. no. 8457S), rabbit anti-HA (C29F4, Cell Signaling Technology, cat. no. 3724S), mouse anti-FLAG-HRP (M2, Sigma-Aldrich, cat. no. A8592), mouse anti-Lamin A (133A2, Cell Signaling Technology, cat. no. 86846), mouse anti-Histone H3 (Cell Signaling Technology, cat. no. 14269S), rabbit anti-Histone H3K27ac (D5E4, Cell Signaling Technology, cat. no. 8173S), rabbit anti-Histone H3K18ac (D8Z5H, Cell Signaling Technology, cat. no. 13998), rabbit anti-p300 (D8Z4E, Cell Signaling Technology, cat. no. 86377S), rabbit anti-CBP (D6C5, Cell Signaling Technology, cat. no. 7389S), rabbit anti-actyl-p300/CBP (Cell Signaling Technology, cat. no. 4771S), anti-rabbit IgG-HRP (Cell Signaling Technology, cat. no. 7074), anti-mouse IgG-HRP (Cell Signaling Technology, cat. no. 7076). Antibodies used for ChIP–seq were mouse anti-FLAG (M2, Sigma-Aldrich, cat. no. F1804), rabbit anti-Histone H3K27ac (D5E4, Cell Signaling Technology, cat. no. 8173S), rabbit anti-Histone H3K4me1 (D1A9, Cell Signaling Technology, cat. no. 5326S), rabbit anti-Histone H3K27me3 (C36B11, Cell Signaling Technology, cat. no. 9733S), rabbit anti-Histone H3K4me0 (Active Motif, cat. no. 91317), rabbit anti-p300 (D2X6N, Cell Signaling Technology, cat. no. 54062), spike-in antibody (Active Motif, cat. no. 61686).

### CRISPR–Cas9 screening and analysis

The 4D6 cell line was transduced with lentiCas9-Blast (Addgene plasmid no. 52962)[55], selected under 10 mg ml⁻¹ blasticidin, and clones were picked for homogeneous Cas9 expression. Cas9-expressing 4D6 cells were transduced with FuGW-G5p-mKate2 (Addgene plasmid no. 105183)[56] and selected clones were verified for UAS-mKate2 genomic insertion by PCR. Then, the Cas9 + UAS-mKate2-expressing cells were transduced with pInducer20-EGFP-P2A-Gal4DBD-mouse Aire CTT and picked clones with homogenous EGFP expression after 1 µg ml⁻¹ Dox treatment were further verified for Aire CTT-dependent mKate2 expression by expressing sgRNAs that target mouse Aire CTT. See Supplementary Table 2 for sequences of PCR primers and sgRNAs used.

The engineered 4D6 cells were transduced with a lentiviral Human Brunello CRISPR knockout pooled sgRNA library (a gift from David Root and John Doench (Addgene no. 73178-LV)[57]) at a multiplicity of infection of 0.4, aiming for 500-fold representation of each sgRNA. Library-transduced cells were selected under 1 mg ml⁻¹ puromycin for 2 d and further expanded for another 7 or 10 d. Cells were treated with 1 µg ml⁻¹ Dox 24 h before sorting to induce EGFP-P2A-Gal4DBD-Aire CTT expression; then the top 5% and bottom 5% of the population were sorted based on mKate2/EGFP ratios on the SH800S Cell Sorter (Sony Biotechnology). Genomic DNA was extracted using DNeasy Blood and Tissue Kit (QIAGEN, cat. no. 69504) and cleaned using OneStep PCR Inhibitor Removal Kit (Zymo Research, cat. no. D6030). Sequencing libraries were generated by PCR amplification as described[57], pooled at an equal molar concentration, purified using MinElute Reaction Cleanup Kit (Qiagen, cat. no. 28204) to enrich for 350–360-base pair (bp) amplicons and subsequently sequenced on an Illumina sequencing platform (GENEWIZ). Demultiplexed sequencing reads were trimmed using Cutadapt (v.2.5)[58], yielding only 20-bp sequences corresponding to sgRNAs. The statistical analysis of sgRNA enrichment (Supplementary Table 3) was performed using MAGeCK-VISPR (v.0.5.6) with the 'MAGeCK-RRA' experimental configuration and visualized using the MAGeCKFlute R package (v.2.0.0)[59].

### RNA-seq and analysis

Dox-inducible Aire-expressing 4D6 cells were cultured in the absence or presence of 1 µg ml⁻¹ Dox for 24 h. Total RNA was extracted with Direct-zol RNA Miniprep Kit (Zymo Research, cat. no. R2052). RNA-seq libraries were prepared using NEBNext Ultra II RNA Library Prep Kit (New England Biolabs, cat. no. E7775S) with ribosomal RNA depletion using NEBNext rRNA Depletion Kit v2 (New England Biolabs, cat. no. E7405L) and sequenced on an Illumina NovaSeq 6000 (Novogene) with 150-bp paired-end reads.

Quality control (QC) was performed on demultiplexed sequencing files using FASTQC (v.0.11.3)[60]. Sequencing reads were trimmed using Trimmomatic (v.0.36) and aligned to reference genome (GRCh38 primary assembly, release v.43) using STAR (v.2.7.0a)[61]. Read counting across genomic features was performed using the featureCounts function within Rsubread R package (v.2.12.3) with duplicated reads ignored (ignorDup = T)[62]. Differential gene expression analysis was performed using the DEseq2 R package (v.1.38.3) and visualized using the ggplot2 R package (v.3.4.1)[63,64]. BAM files generated during the STAR alignment were converted to bigwig files using deepTools (v.3.5.1)[65] with the following settings: bamCoverage --scaleFactor "scale factor" --smoothLength 150 --binSize 50 -e 200. "scale factor" was calculated as 1 divided by the size factor that was obtained during DEseq2 analysis. After verification of consistency between replicates, bigwig files were averaged using WiggleTools (v.1.2.2) and bedGraphToBigWig (v.366) and imported into Integrative Genomics Viewer (IGV, v.2.15.1) for visualization at specific loci[66–68].

For Fig. 1a, bulk RNA-seq performed on 4D6 cells was compared with previously published single-cell RNA-seq data of human thymic epithelial cells (n = 477 AIRE⁺ mTECs)[30].

## 5EU-seq and analysis

Dox-inducible Aire-expressing 4D6 cells were cultured in the absence or presence of 1 µg ml$^{-1}$ dox for 24 h. DMSO or 3 µM A-485 was added to cells 4 h before, and 0.5 mM 5′-ethynyl uridine (5′-EU) was added to the cell culture 30 min before RNA extraction. Total RNA was immediately extracted by using Direct-zol RNA Miniprep Kit. Ribosomal RNA was depleted using NEBNext rRNA Depletion Kit v2 (New England Biolabs, cat. no. E7405L). 5′-EU-labeled nascent RNA was biotinylated and pulled down using Click-iT Nascent RNA Capture Kit (Invitrogen, cat. no. C10365) according to the manufacturer's protocol. 5′-EU-labeled RNA captured on streptavidin-beads was then immediately used for sequencing library preparation with NEBNext Ultra II RNA Library Prep Kit (New England Biolabs, cat. no. E7775S). Libraries were sequenced on Illumina NovaSeq 6000 (Novogene) with 150-bp paired-end reads.

QC, trimming, reference genome alignment, BAM file conversion, average bigwig file generation and visualization were performed as described for bulk RNA-seq. Bigwig files showing log$_2$ fold-changes between two groups were generated using deepTools (v.3.5.1) with the bigwigCompare function.

## ChIP–seq and analysis

4D6 cells were seeded on 150-mm plates with or without Dox and grown for 24 h. For drug-treated samples, an equal volume of DMSO, 3 µM A-485 or 0.25 µM dCBP-1 was added to cells 4 h before collection. For anti-FLAG and anti-p300 ChIP–seq, cells were washed three times with PBS and then crosslinked with 2 mM disuccinimidyl glutarate in PBS for 45 min at 22 °C. Cells were then washed again three times with PBS and crosslinked with 1% formaldehyde (Sigma, Thermofisher and Electron Microscopy Sciences) in PBS for 10 min at 22 °C. For anti-histone mark ChIP–seq, cells were crosslinked with 1% formaldehyde in fresh media for 10 min at 22 °C. After formaldehyde crosslinking, all cells were washed once with PBS and quenched with 0.125 M glycine in PBS for 5 min at 22 °C. Quenched cells were washed with ice-cold PBS, and then collected in ice-cold PBS supplemented with 0.5 mM PMSF. Cells were spun down at 500g for 5 min and cell pellets were supplemented with 1 µl of 100 mM PMSF and 1 µl of 1 × mammalian protease inhibitor cocktail (G-Biosciences), and then flash-frozen in liquid nitrogen and stored at −80 °C until ready to use. For one ChIP–seq pull-down, ~15 × 10$^6$ cells were used. See the Supplementary Note for more details on ChIP–seq pull-downs, library generation, and sequencing and data analyses.

## ATAC-seq and analysis

4D6 cells ±Dox were collected at 24 h post-induction. A total of 100,000 cells were used for each replicate. ATAC-seq libraries were prepared using the ATAC-seq library prep kit (Active Motif) according to the manufacturer's protocol. Deep sequencing was performed using a NovaSeq sequencer (Illumina) with paired-end 150-bp reads.

QC and trimming were performed as for bulk RNA-seq. Sequencing reads were aligned to reference genome (GRCh38 primary assembly) using bwa (v.0.7.17). The resulting SAM files were converted to BAM files, sorted and indexed, and reads mapped to mitochondrial DNA were removed using Samtools (v.1.6). Read fragment sizes were checked using ATACseqQC R package (v.3.19)[69]. Post PCR duplicate removal, reads were shifted +4 bp and −5 bp for positive and negative strands, respectively, and were further split into NFRs or mono- or di-nucleosome regions using the alignmentSieve function in deepTools (v.3.5.1). Peak calling was performed on bam files containing reads mapped to NFRs using MACS2 (v.2.2.7.1). NFRs that overlapped with Aire peaks and showed strong Aire ChIP–seq signals were selected as Aire-bound NFRs (n = 542, Supplementary Table 1), whereas NFRs that had similar ATAC-seq read pileups as those in Aire-bound NFRs but showed no Aire ChIP–seq signals were selected as Aire-free NFRs (n = 658, Supplementary Table 1).

## IF microscopy

4D6 or 293T cells were seeded onto glass coverslips in 12-well plate format. Cells at ~70% confluence were transiently transfected with indicated plasmids. 4D6 cells were seeded in the presence or absence of Dox. For inhibitor treatments, 4D6 cells were first induced with Dox for 4 h; then DMSO, A-485 (3 µM final concentration), dCBP-1 (0.25 µM final concentration), dBET6 (100 nM final concentration) or JQ1 (1 µM final concentration) was added to medium for a total of 4 h of inhibitor treatment and 8 h of Dox induction. At 24 h post-transfection or after Dox treatment for indicated amounts of time, cells were washed with PBS, and then fixed with 2% paraformaldehyde in PBS for 10 min. Cells were washed again with PBS and then permeabilized with 0.5% Triton X-100 in PBS for 10 min. Cells were blocked with 1% BSA in PBS-T (PBS + 0.2% Tween-20) for 15 min at 22 °C or 16 h at 4 °C, and then probed with antibodies. Cells were then counterstained with DAPI (Life Technologies). Coverslips were mounted using Fluoromount-G (SouthernBiotech) or Vectashield (Vector Laboratories, cat. no. H-1000-10). Two-dimensional images were captured on a wide-field Zeiss Axio Imager M1. Image z-stacks were captured on a wide-field Nikon Ti2 equipped with a Nikon DS-Qi2 large-format CMOS camera (11 frames per stack, 0.3-µm z-step), or a Yokogawa spinning disk confocal Nikon Ti equipped with a Hamamatsu ORCA-Fusion BT sCMOS camera (9 frames per stack, 0.3-µm z-step).

All quantitative IF imaging analyses were performed with Fiji Is Just ImageJ (FIJI, ImageJ2 v.2.14.0/1.54f). See the Supplementary Note for details of quantitative imaging analyses.

## IF with RNA-FISH

Cells were plated on coverslips in 12-well tissue culture plates and grown for 24 h. For inhibitor treatment experiments, Dox and DMSO, A-485 or dCBP-1 were supplemented as described for IF microscopy. For all other experiments, Dox was supplemented to media for 24 h. Cells were washed with PBS and fixed using 4% paraformaldehyde (Electron Microscopy Sciences, cat. no. 15714) in PBS for 10 min. After washing cells twice in PBS, permeabilization of cells was performed using 0.5% Triton X-100 in PBS for 10 min, followed by washing with PBS-T twice. Cells were blocked with 1% RNase-free BSA (Sigma-Aldrich, cat. no. 126609) in PBS-T for 30 min, and then incubated with the FITC-conjugated anti-FLAG antibody (10 µg ml$^{-1}$ in PBS-T with 1% BSA) at 22 °C for 1 h. After washing in PBS-T twice and PBS once, cells were re-fixed using 4% PFA in PBS for 10 min. After two PBS washes, cells were pre-incubated in Buffer A (20% Stellaris RNA-FISH buffer A (Biosearch Technologies, cat. no. SMF-WA1-60) and 10% deionized formamide (Millipore, cat. no. S4117) in RNase-free water (Thermofisher, cat. no. 10977023)) for 5 min. Cells were then incubated with 125 nM nascent RNA probes in hybridization buffer (90% Stellaris RNA-FISH hybridization buffer (Biosearch Technologies, cat. no. SMF HB1-10) and 10% deionized formamide) overnight in a humidified chamber at 37 °C. After washing with Buffer A for 30 min at 37 °C, nuclei were stained with DAPI in Buffer A for 5 min, followed by a Stellaris RNA-FISH buffer B (Biosearch Technologies, cat. no. SMF-WB1-20) wash for 5 min. Coverslips were mounted onto glass slides with Vectashield and sealed with nail polish. Nascent RNA-FISH probes were custom-designed to target *RIC8A*, *SETD1B* and *UBTF* intronic regions using Stellaris probe designer and manufactured at Biosearch Technologies. Sequences of RNA-FISH probes are listed in Supplementary Table 2.

Three-dimensional images were acquired on a Nikon Ti2 fluorescence microscope with ×60 objective using NIS-Elements acquisition software, at a resolution of 9.2308 pixels per µm and voxel size of 0.1083 × 0.1083 × 0.3 µm$^3$. Microscope specifications can be found at https://nic.med.harvard.edu/microscopes/george_michael/.

Images were post-processed using FIJI or MATLAB for further analyses. See the Supplementary Note for details of quantitative imaging analyses.

## Aire protein expression and chromatin fractionation assays

4D6 and 293T cells were transfected with plasmids expressing indicated proteins for assaying expression levels and chromatin fractionation assays in 12-well and 6-well plate formats, respectively. 4D6 cells induced with 0.1–1 µg ml$^{-1}$ Dox were also seeded in 12-well format for expression level determination. At 24 h after transfection or Dox induction, cells were collected in PBS and washed one time with PBS. For samples expressing mouse Aire variants that were sensitive to protein degradation and/or histone deacetylation, washed cells were immediately lysed in 1% SDS Buffer (50 mM Tris pH 7.5, 150 mM NaCl, 1% SDS, 0.3 mM dithiothreitol; 75–100 µl per sample), boiled for 5 min and processed for western analyses as described below. For all other samples, washed cells were incubated in Hypotonic Buffer (20 mM HEPES pH 7.5, 0.05% IGEPAL, 1.5 mM MgCl$_2$, 10 mM KCl, 5 mM EDTA and 1 × mammalian protease inhibitor cocktail; 50 µl and 100 µl per sample for 12-well and 6-well plate formats, respectively) for 15 min at 4 °C. The lysed cells were spun down at 500$g$ for 5 min at 4 °C and the supernatant (cytoplasmic fraction) was removed. The pellet (nuclear fraction) was washed two times with ice-cold PBS.

For comparing mouse Aire variant expression levels, the PBS-washed nuclear fraction was lysed in 1% SDS Buffer and boiled for 5 min. BCA assay was used to determine the total protein concentration of lysates. See the Supplementary Note for Aire chromatin fractionation assay details.

## Luciferase reporter assay

4D6 cells (~80% confluence) were transfected with 200 ng of pGL4.31 (Firefly luciferase reporter plasmid under 5XUAS box promoter), 1 ng of phRLCMV (a constitutive Renilla luciferase reporter plasmid) and 25 ng of plasmid expressing Gal4-DBD fusion variants by using Lipofectamine 2000 in a 48-well plate format. At 24 h post-transfection, Gal4$^{DBD}$-CTT transcriptional activity was measured by using Dual Luciferase Reporter assay (Promega) with a Synergy2 plate reader (BioTek). Firefly luciferase activity was normalized against Renilla luciferase activity.

## RT–qPCR and 5EU–qPCR

4D6 cell lines expressing Aire variants were collected for RNA extraction 24 h post Dox treatment or transfection, respectively. For RT–qPCR, total RNA was extracted using Direct-zol RNA Miniprep Kit with DNase I digestion and reverse-transcribed using SuperScript II (Life Technologies) with oligo(dT$_{18}$).

For 5EU–qPCR, 0.5 mM 5′-EU was added to cells 30 min before Direct-zol RNA extraction. Extracted 5′-EU-labeled RNA was biotinylated and pulled down using the Click-iT Nascent RNA Capture Kit (Invitrogen, cat. no. C10365) according to the manufacturer's protocol. 5′-EU-labeled RNA captured on streptavidin-beads was then reversed transcribed using SuperScript II with oligo(dT$_{18}$).

qPCR was performed using Power SYBR Green PCR Master Mix (Invitrogen) on a CFX-Connect detection system (Bio-Rad). The expression levels of Aire-induced genes and *RPL31* (Aire-independent gene control) were normalized against the Aire-independent gene *RPL18* using the ΔΔCt method. qPCR primer sequences are listed in Supplementary Table 2. All statistical analyses were performed in Prism (GraphPad).

## *Escherichia coli* expression and purification of recombinant proteins

See the Supplementary Note for methods used to prepare recombinant proteins in this study.

## MS of Aire CTT binding partners

Equal amounts of His$_6$-GST and His$_6$-GST-mouse Aire CTT in MS Lysis Buffer (50 mM Tris pH 8, 300 mM NaCl and 10% glycerol, 1 mM PMSF) were captured onto glutathione Sepharose beads (Cytiva) for 1 h at

4 °C. GST-protein-bound beads were washed three times with MS Lysis Buffer. 293T cells were lysed in Hypotonic Buffer and Nuclear Extraction Buffer to obtain nuclear 'soluble' extracts as described above for chromatin fractionation assays. 293T nuclear extracts were incubated with GST-protein-bound beads for 16 h at 4 °C. Beads were washed with ice-cold PBS + 0.05% IGEPAL and three times with ice-cold PBS. Bound proteins were eluted with Laemmli sample buffer and boiled for 5 min. The eluted proteins were run on SDS–PAGE gel and stained with Coomassie Blue. A gel band around ~250 kDa was extracted for each lane containing either proteins interacting with His$_6$-GST-CTT or His$_6$-GST (negative control). Extracted gel bands were sent to Taplin Mass Spectrometry Facility at Harvard Medical School for in-gel protease digestion and micro-capillary liquid chromatography with tandem MS analysis. Protein interaction candidates identified from MS were ranked from high to low based on their coverages (%) in His$_6$-GST-CTT pull-down samples where there were no coverages in His$_6$-GST pull-down samples (Supplementary Table 4).

## GST-Aire CTT-FLAG pull-downs

All pull-downs were performed at 4 °C unless specified. Equal amounts of His$_6$-GST-FLAG and His$_6$-GST-mouse Aire CTT-FLAG variants in SEC Buffer (25 mM HEPES pH 7.5, 100 mM NaCl, 5 mM BME) were captured onto glutathione Sepharose beads for 30 min. GST-protein-bound beads were washed three times with SEC Buffer, and then blocked with 0.1% BSA in SEC Buffer for 30 min until ready to mix with 293T nuclear extracts or purified recombinant proteins.

For 293T lysate pull-downs, 293T cells were transiently transfected with HA-p300 variant expression vectors in a 6-well plate format. At 48 h post-transfection, cells were lysed in Hypotonic Buffer and Nuclear Extraction Buffer to obtain nuclear 'soluble' extracts as described above for chromatin fractionation assays. Nuclear fractions were mixed with BSA-blocked GST-protein-bound beads for 1 h. For recombinant FLAG-tagged full-length p300 (Active Motif) pull-downs, p300 was diluted into 25 mM HEPES pH 7.5, 100 mM NaCl, 1.5 mM MgCl$_2$, 5 mM BME and added to BSA-blocked GST-protein-bound beads for 1 h. Pull-downs were eluted by boiling beads in Laemmli sample buffer for 5 min.

For pull-downs with recombinant CBP TAZ2, CBP TAZ2 was mixed with BSA-blocked GST-protein-bound beads in SEC Buffer for 1 h; then beads were washed three times with SEC Buffer. Bound CBP TAZ2 was eluted with the addition of 25 mM glutathione (Sigma) in SEC Buffer. Elutions were passed through a Costar Spin-X centrifuge tube 0.22-µm filter (Millipore) to prevent contaminating beads from entering samples.

## NMR spectroscopy

Please see the Supplementary Note for details on the NMR spectroscopy methods used in this study.

## Fluorescence polarization peptide binding assay

N-terminal 5,6-carboxyfluorescein (5,6-FAM) FAM-labeled H3 peptide (aa 1–21) with no modification or K4me1 was purchased from Anaspec. Binding reaction mixtures (150 µl) contained 50 nM fluorescein-labeled peptide and increasing concentrations of PHDs (0–900 µM) in SEC Buffer. Fluorescence polarization was measured at 25 °C using a Synergy H1 (BioTek). Data were fit to a simple binding isotherm using Prism. Polarization values were normalized to the value without added PHD. All binding experiments were performed at least three times.

## ITC

Protein samples were dialyzed in 20 mM HEPES pH 7.5, 100 mM NaCl, 1 mM dithiothreitol for 16 h at 4 °C. ITC experiments were performed using a VP-ITC calorimeter (MicroCal) and were conducted at 20 °C with 19–251–2-µl injections, 60-s delay between injections with 760 µM of mouse Aire CTT in the syringe and 19–76 µM TAZ2 in the cell. As a

control, 760 μM mouse Aire CTT was titrated into the cell containing dialysis buffer only. Thermograms were fit to a two-site binding model using MicroCal Origin 7.0 software. Experiments were collected in duplicate.

## Reporting summary

Further information on research design is available in the Nature Portfolio Reporting Summary linked to this article.

## Data availability

The Gene Expression Omnibus accession number for next-generation sequencing data reported in this paper is GSE243825. Source data are provided with this paper. All other data supporting the findings of this study are present in the article and the Supplementary Information.

## Code availability

All code used in this study was previously reported and cited in the Methods. Any additional information required to reanalyze the data from this paper is available from the lead contact upon request.

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

## Acknowledgements

We thank K. Adelman (Harvard Medical School, HMS) and B. Bernstein (Dana Farber Cancer Institute, DFCI) for helpful discussions. We also thank S. Bevill (DFCI), B. Martin (HMS) and A. Dall'Agnese (Whitehead Institute) for technical guidance. IF and RNA-FISH microscopy images were collected at the Core for Imaging Technology & Education (CITE) at HMS. ITC experiments were performed at the Center for Macromolecular Interactions (CMI) at HMS. Next-generation sequencing library construction quality control was performed by the Molecular Genomics Core at Boston Children's Hospital. This work was supported by the Jeffrey Modell Foundation Postdoctoral Award (Y.-S.H.), the Cancer Research Institute Irvington Postdoctoral Fellowship no. CRI4120 (Q.Z.), the Damon Runyon Cancer Research Foundation (S.C.B.), and NIH grants no. GM136859 (H.A.) and nos AI147099 and AI180137 (S.H.).

## Author contributions

Y.-S.H., Q.Z. and R.T. performed the experiments. Y.-S.H., Q.Z., H.A. and S.H. designed and coordinated the study. Y.-S.H., Q.Z., R.T., S.C.B., H.A. and S.H. interpreted the results. Y.-S.H., Q.Z. and S.H. wrote the manuscript. All authors discussed the results and commented on the manuscript.

## Competing interests

The authors declare no competing interests.

## Additional information

**Extended data** is available for this paper at https://doi.org/10.1038/s41590-024-01922-w.

**Correspondence and requests for materials** should be addressed to Sun Hur.

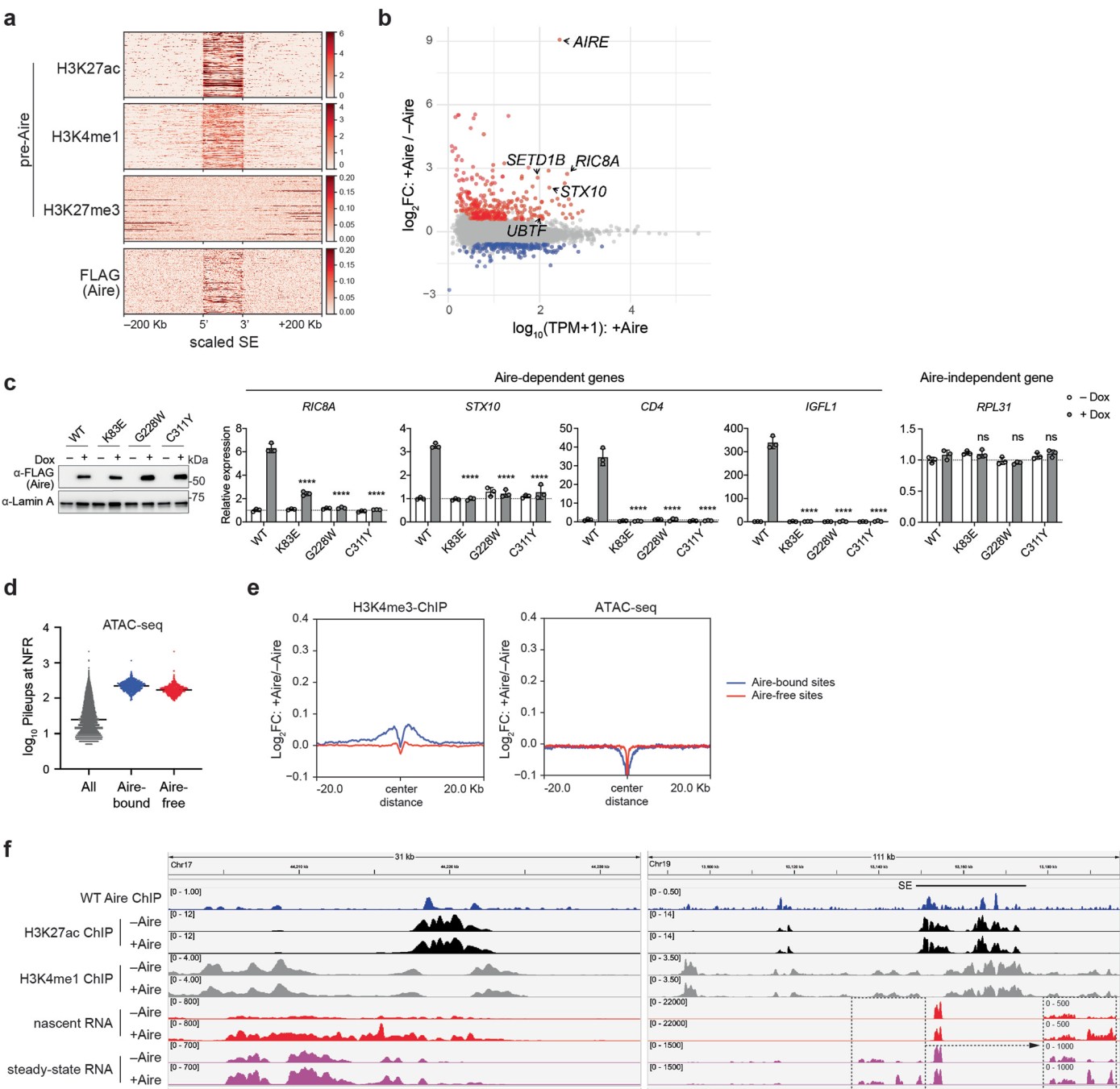

**Extended Data Fig. 1 | Thymic epithelial cell line (4D6) with Dox-inducible Aire recapitulates Aire transcriptional activity and functions. a**, Heatmaps of normalized ChIP-seq signals (Counts Per Million, CPM) for indicated proteins spanning 200Kb up- or downstream of H3K27ac-delimited super-enhancers (*n* = 529). Histone marks are shown from 4D6 cells prior to Aire expression (pre-Aire). **b**, Gene expression (transcripts per million, TPM) in Aire-expressing 4D6 cells (+Aire) vs. expression changes between cells with and without Aire (+Aire/−Aire). Steady-state bulk RNA-seq was performed on Dox-inducible Aire-expressing 4D6 cells without or with 24 h Dox treatment. Red, genes upregulated by Aire (>1.5-fold, FDR < 0.05); blue, genes downregulated by Aire (<1.5-fold, FDR < 0.05). FC, fold-change. FDR, false discovery rate. **c**, Transcriptional activity of Aire WT or APS-1 mutants as measured by the relative mRNA levels of Aire-dependent genes, *RIC8A*, *STX10*, *CD4* and *IGFL1*, in Dox-inducible 4D6 cells. An Aire-independent gene, *RPL31*, was also examined as a negative control. All genes

were normalized against the internal control *RPL18*. Data are representative of three independent experiments and presented as mean ± s.d. (three biological replicates). *P*-values (one-way ANOVA with Dunnett's multiple comparisons test) were calculated vs. WT Aire (+Dox). ****$P$ < 0.0001; $P$ > 0.05 is not significant (ns). Left: western blot (WB) showing the nuclear levels of FLAG-tagged Aire proteins compared to endogenous levels of Lamin A. **d**, Pileup counts at all nucleosome-free regions (NFRs), Aire-bound or Aire-free NFRs that showed similar ATAC-seq signals. *n* = 215039, 542 and 658, respectively. **e**, Aire-induced changes in H3K4me3 ChIP or ATAC-seq 20 Kb upstream or downstream of Aire-bound and Aire-free NFRs, *n* = 542 and 658, respectively. **f**, Genome browser views of normalized ChIP-seq and RNA-seq profiles at exemplar Aire-bound sites in 4D6 cells. Numbers to the left indicate the ranges of normalized reads for RNA-seq or CPM for ChIP-seq. SE, super-enhancer.

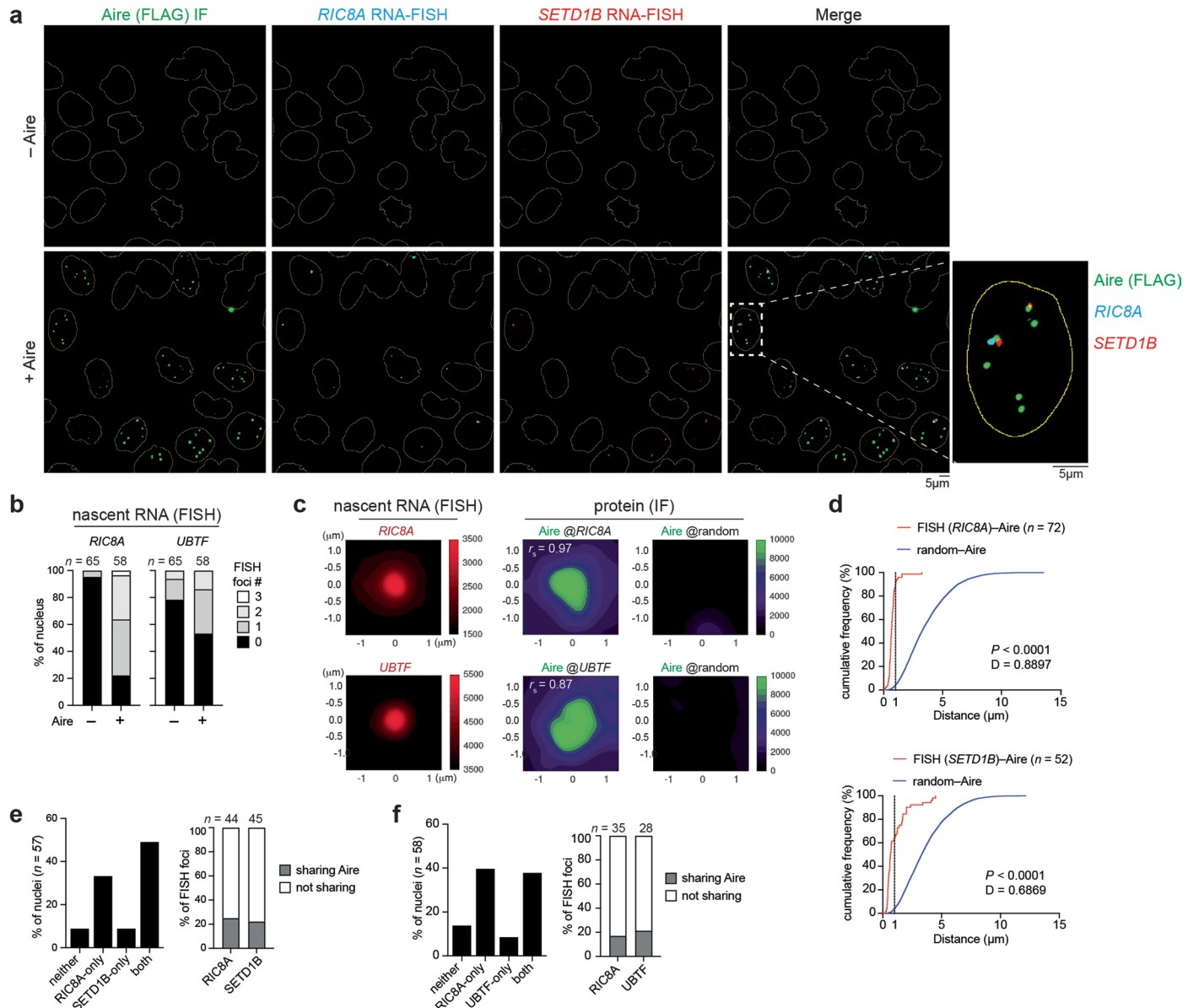

**Extended Data Fig. 2 | Aire links inter-chromosomal genomic loci into transcriptionally active condensates. a**, Nascent RNA-FISH coupled with IF images of 4D6 cells expressing Aire-FLAG. Cells were not (−Aire) or were Dox-induced (+Aire) for 24 h before staining. RNA-FISH probes hybridized to intronic regions of Aire-dependent targets. Yellow outlines represent nuclei boundaries. Bottom right: zoomed-in view of the white-dashed box. For clarity, the zoomed-in image of a nucleus with *RIC8A* and *SETD1B* RNA-FISH foci sharing the same Aire condensate was reproduced from Fig. 1b. **b**, Quantitation of percent of nuclei that have 0-3 RNA-FISH foci (left: *RIC8A*; right: *UBTF*). *n*, number of nuclei. **c**, Quantitation of average intensity signals from RNA-FISH combined with IF of Aire-FLAG in 4D6 cells. Left: average signals of RNA-FISH. Center: Aire IF centered on FISH foci. Right, Aire IF centered on random nuclear positions. *r*ₛ, Spearman's correlation coefficient between RNA-FISH and IF signals. **d**, Distribution of the minimal center-to-center distances between Aire condensates and RNA-FISH

foci (FISH−Aire) or simulated random positions (random−Aire). D, Kolmogorov-Smirnov test D statistic. *P*, *P*-value. RNA-FISH and Aire condensates often showed partial (within ~1 μm) rather than complete overlap, consistent with recent microscopy studies on active genetic elements. The close association between Aire condensates and RNA-FISH foci supports Aire's role in directly activating transcription. **e**, Quantitation of nuclei harboring RNA-FISH foci. Left: percentage of nuclei containing *RIC8A* and/or *SETD1B* RNA-FISH foci from **a** and Fig. 1f. Right: frequency of RNA-FISH foci sharing Aire condensates in nuclei containing both *RIC8A* and *SETD1B* RNA-FISH foci. **f**, Quantitation of nuclei harboring RNA-FISH foci. Left: percentage of nuclei containing *RIC8A* and/or *UBTF* RNA-FISH foci from **b**. Right: frequency of RNA-FISH foci sharing Aire condensates in nuclei containing both *RIC8A* and *UBTF* RNA-FISH foci. *n*, number of nuclei examined (left) or number of RNA-FISH spots (right). All data are representative of at least three independent experiments.

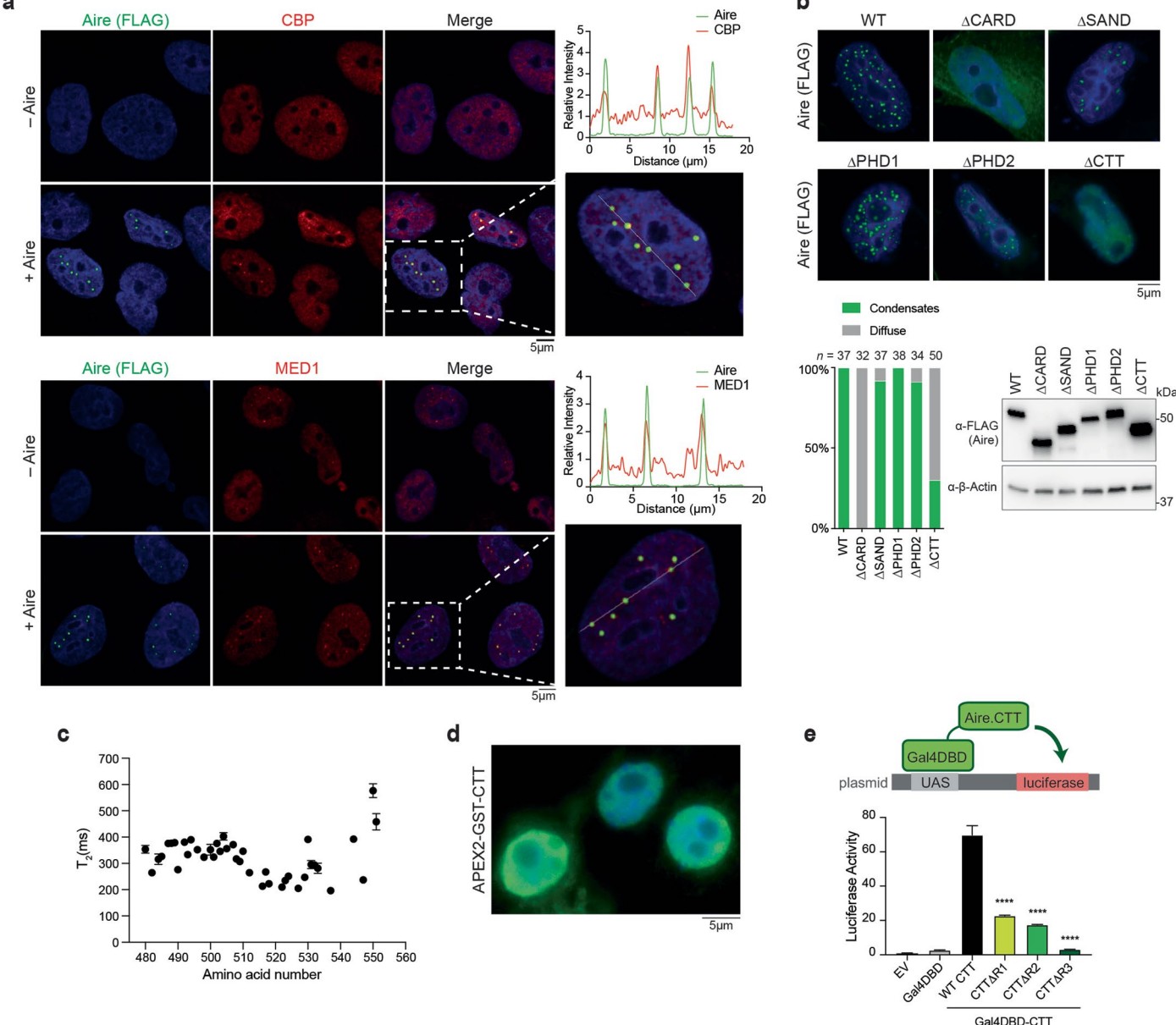

**Extended Data Fig. 3 | Aire condensates represent sites of active transcription and require Aire CTT. a**, Representative IF images of endogenous CBP and MED1 in 4D6 cells before and after Aire-FLAG expression. Cells were not (−Aire) or were induced with Dox (+Aire) for 24 h before immunostaining with anti-FLAG, anti-CBP and anti-MED1. Right: zoomed-in views of nuclei enclosed within white-dashed boxes along with measured fluorescence intensities across drawn solid white lines. **b**, Representative IF images of Aire WT and domain deletion variants in 4D6 cells. Cells were transfected with mouse Aire-FLAG expression plasmids 24 h prior to fixation. Bottom left: percentage of nuclei with Aire condensates vs. diffuse Aire staining. *n*, number of nuclei examined. Bottom right: WB showing the levels Aire variants. **c**, 15 N relaxation time ($T_2$) of Aire CTT (340 μM) recorded at 700 MHz and 15 °C. The averaged $T_2$ is 0.319 s

[corresponding relaxation rate ($R_2$) of 3.13 s$^{-1}$]. **d**, Representative IF image of 293 T transiently expressing mouse Aire CTT fused with APEX2-GST (APEX2-GST-CTT) for 24 h prior to fixation and staining with anti-FLAG. **e**, Activation domain (AD)-like activity of mouse Aire CTT and various CTT deletion mutants as measured in 4D6 cells. Top: schematic of the AD-reporter assay. CTT was fused with Gal4 DNA-binding domain (Gal4DBD), which binds upstream activation sequences (UAS) and controls the expression level of the reporter luciferase. Bottom: luciferase activities were shown relative to that of empty vector (EV)-transfected 4D6 cells. Data are representative of three independent experiments and presented as mean ± s.d. (three biological replicates). *P*-values (one-way ANOVA with Dunnett's multiple comparisons test) were calculated vs. WT CTT, ****$P < 0.0001$.

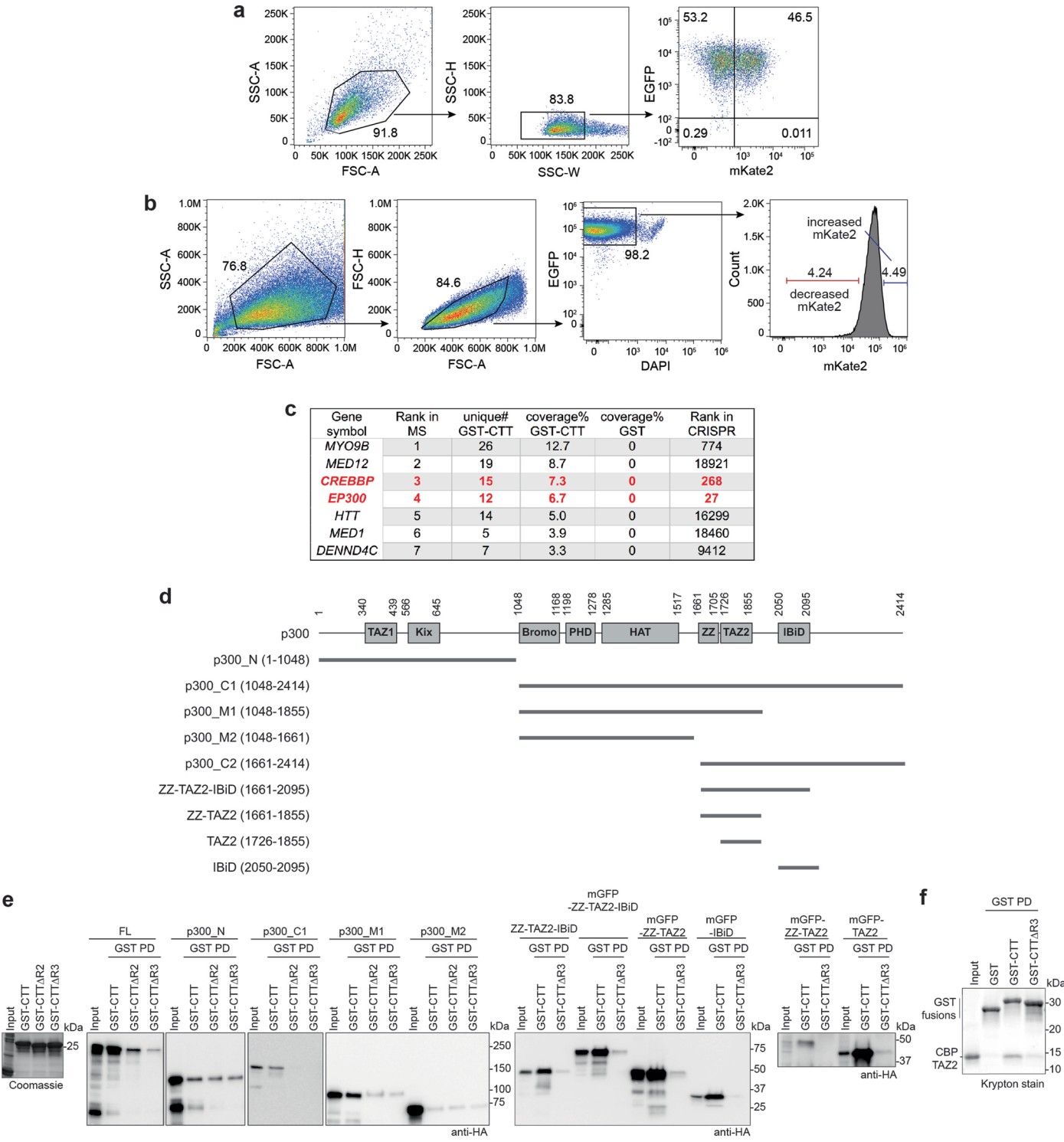

**Extended Data Fig. 4 | CBP/p300 directly interact with Aire CTT. a**, Gating strategy of flow cytometric analyses shown in Fig. 3a. Sample with GFP-P2A-Gal4DBD-CTT expression (+ CTT-gRNA) was shown as an example. **b**, Sorting strategy of CRISPR screening shown in Fig. 3b. **c**, LC-MS/MS analysis results of the GST-CTT pull-down in Fig. 3c. A ~ 250 kDa band in the GST-CTT-bound fraction and the equivalent region in the GST control were analyzed. See also Supplementary Table 4. **d**, Domain architecture of p300 and the truncation variants used in **e**. Note that the paralog CBP has the same domain architecture.

**e**, His₆-GST-CTT variant pull-downs of HA-tagged p300 transiently expressed in 293 T nuclear extracts. Far left: SDS-PAGE gel of GST-CTT variant proteins captured onto glutathione beads used for the pull-down assay. Right: WBs of HA-p300 variants pulled-down with GST-CTT WT vs. GST-CTTΔR2 or GST-CTTΔR3. Data shown are of at least two independent experiments. **f**, His₆-GST-CTT variant pull-downs of recombinantly expressed and purified CBP TAZ2. Data is representative of three independent experiments.

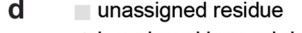

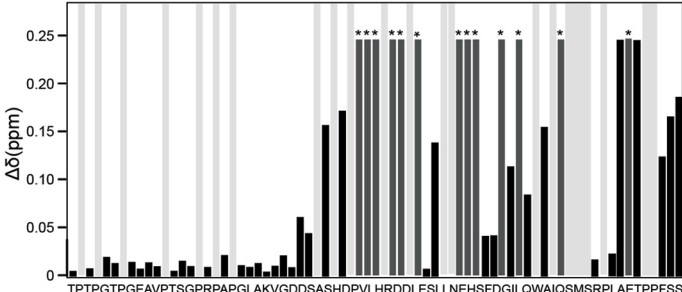

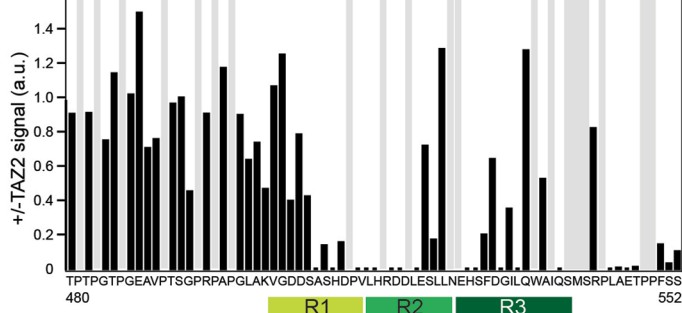

**Extended Data Fig. 5 | Aire CTT R2 and R3 bind CBP TAZ2. a**, NMR spectroscopy of mouse Aire CTT (aa 480-552). 2D-¹⁵N-HSQC of ¹⁵N¹³C-labeled Aire CTT was recorded at 15 °C (288 K) on an 800 MHz spectrometer equipped with a cryoprobe. The chaotrope urea was used to identify regions with structural propensity, as measured by exchange broadening. Aire residues in both panels are severely exchange broadened in the absence of urea (blue), suggesting secondary structure propensity. Upon addition of urea (green), this structural propensity is diminished; the highlighted peaks gained intensity, allowing for their assignment. Isolated CTT is largely disordered, as evident from low dispersion in the proton chemical shift and secondary chemical shifts. R2 and R3, on the other hand, alternate between disordered and alpha helical characteristics, as evidenced by strong peak broadening. The helical propensities of R2 and R3 are further supported by secondary structure predictions in **b** and

AlphaFold modeling (Fig. 3e). **b**, Helical propensity of CTT from human and mouse Aire (ProtScale, using Chou & Fasman parameters). Aire CARD, which has high α-helical propensity, was used in comparison. **c**, NMR spectroscopy of ¹⁵N¹³C-labeled mouse Aire CTT (aa 480-552) in the absence or presence of CBP TAZ2. 2D-¹⁵N-HSQC of ¹⁵N¹³C-labeled Aire CTT alone (blue) and with 1:1.75 molar ratio of ¹⁵N¹³C-labeled Aire CTT:CBP TAZ2 (red), recorded at 15 °C (288 K) on an 800 MHz spectrometer equipped with a cryoprobe. Close-ups at a lower contour level show the extent of peak broadening and chemical shift perturbation upon complex formation in select regions. **d**, Plots of detected chemical shift perturbations (CSPs) Δδ [calculated as $\Delta\delta = \sqrt{(0.14 \cdot \Delta N^2 + \Delta H^2)}$] and the signal loss upon addition of CBP TAZ2 (1:1.75 molar ratio of Aire CTT:CBP TAZ2). Mouse Aire CTT residues marked with an asterisk were broadened beyond detection in the Aire−CBP complex.

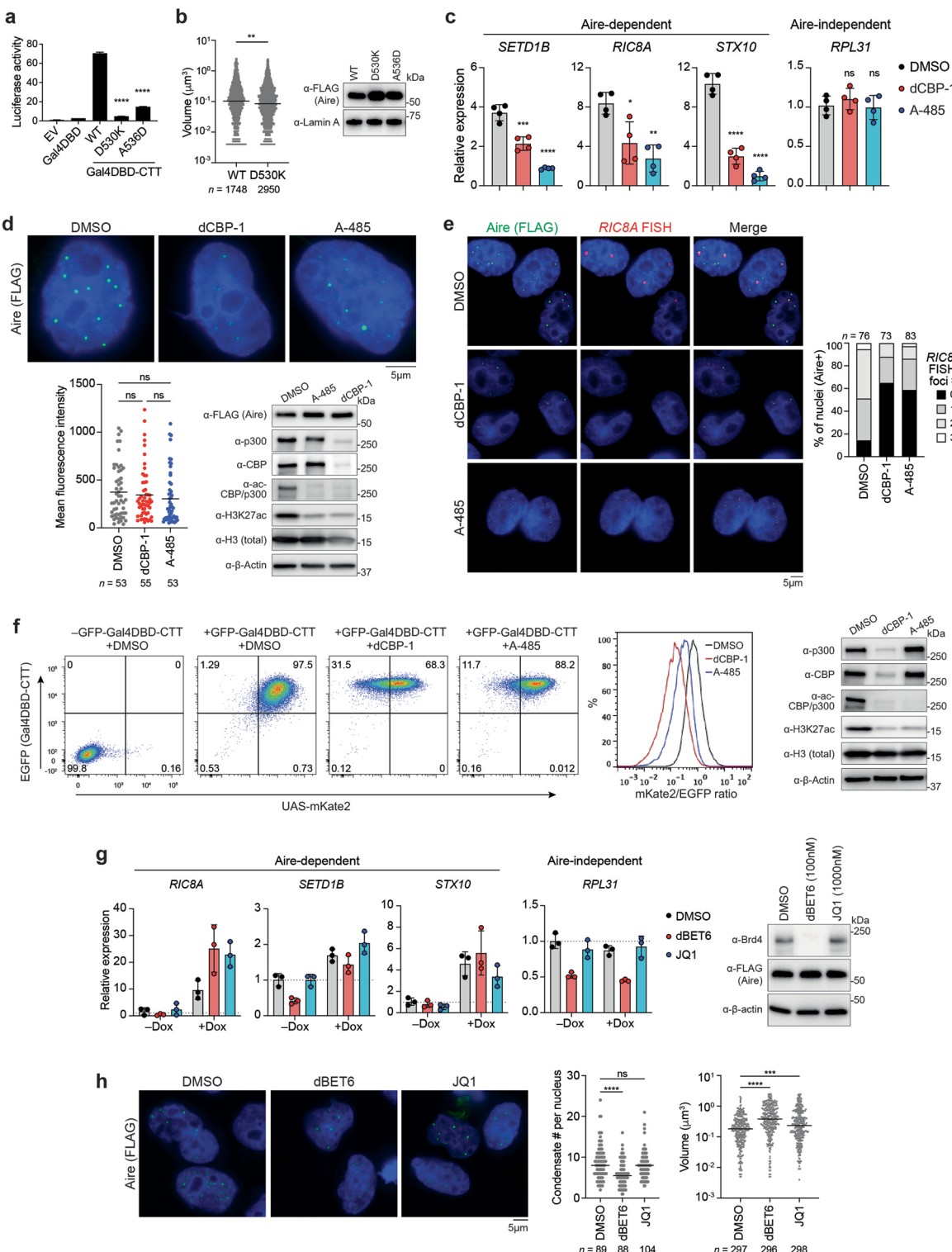

**Extended Data Fig. 6 | See next page for caption.**

**Extended Data Fig. 6 | CBP/p300 are essential for Aire transcriptional condensate formation. a**, AD-like activity of mouse Aire CTT variants as in Extended Data Fig. 3e (mean ± s.d.). *P*-values (one-way ANOVA with Dunnett's multiple comparisons test) vs. Gal4DBD-CTT WT. **\*\*P* < 0.01; \*\*\*\*P* < 0.0001. **b**, Quantification of WT vs. D530K Aire condensate volumes within mouse Aire-expressing nuclei examined in Fig. 3i. *n*, condensates. \*\*P* < 0.01 (two-tailed Mann-Whitney test). Right: WB of Aire variants. **c**, Aire transcriptional activity measured by 5EU-qPCR (mean ± s.d.). Dox, dCBP-1 or A-485, and 5′-EU were added to cells 24, 4, and 0.5 h before harvest, respectively. *P*-values (one-way ANOVA with Dunnett's multiple comparisons test) vs. DMSO-treated cells. \*\*P* < 0.01; \*\*\*P* < 0.001; \*\*\*\*P* < 0.0001; ns *P* > 0.05. **d**, IF images of Aire-FLAG in 4D6 cells as in Fig. 3k. Bottom left: mean fluorescence of nuclei. ns *P* > 0.05, Kruskal-Wallis test with Dunn's multiple comparisons test. Bottom right: WB of indicated proteins. **e**, Images of Aire-FLAG IF with nascent RNA-FISH + dCBP-1 or A-485 in 4D6 cells.

Right: percent of nuclei with 0-3 RNA-FISH foci. *n*, nuclei. **f**, Flow cytometric analyses of UAS-mKate2 expression vs. GFP-P2A-Gal4DBD-CTT expression (left) or their ratios (middle) in 4D6 cells. Cells expressing GFP-P2A-Gal4DBD-CTT (as in Fig. 3a) were treated with DMSO, dCBP-1 or A-485 for 24 h. See Extended Data Fig. 4a for gating strategy. Right, WB of indicated proteins. **g**, Aire transcriptional activity measured by 5EU-qPCR (mean ± s.d.). Dox, dBET6 or JQ1, and 5′-EU were added to cells 24, 4, and 0.5 h prior to harvest, respectively. Right: WB of indicated proteins. **h**, IF images of WT Aire-FLAG in 4D6 + DMSO, dBET6 or JQ1. Middle: Aire condensate quantification per nuclei (*n*, nuclei). Right: Aire condensate volume quantification within cells treated with Brd4 inhibitors (*n*, condensates). *P*-values (Kruskal-Wallis test with Dunn's multiple comparisons test) vs. DMSO-treated cells. \*\*\*P* < 0.001; \*\*\*\*P* < 0.0001; ns *P* > 0.05. All data are representative of at least three independent experiments.

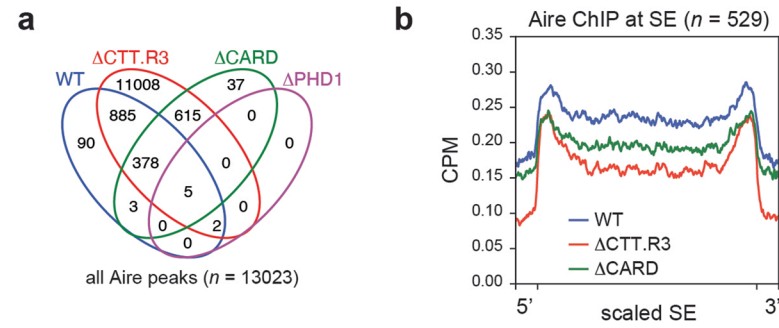

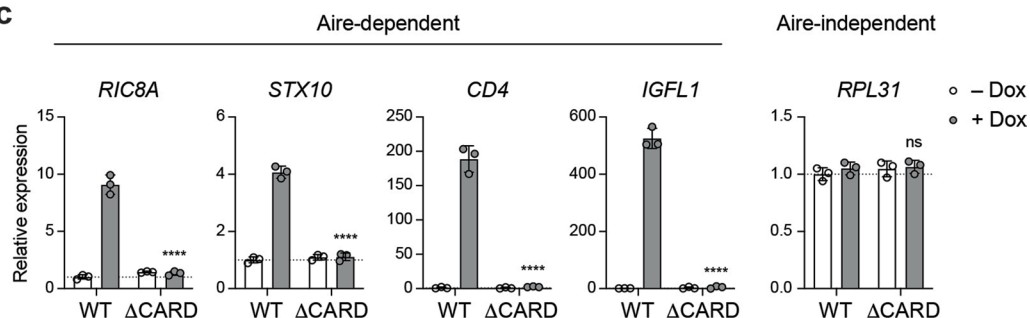

**Extended Data Fig. 7 | CARD is essential for robust Aire localization to super enhancers and mediating transcriptional activity. a**, Venn diagram showing all peaks identified from WT vs. mutant Aire ChIP-seq. **b**, Average Aire WT, ΔCTT. R3, and ΔCARD ChIP-seq profiles (CPM) at H3K27ac-delimited super-enhancers (SEs, *n* = 529, defined in Extended Data Fig. 1a). **c**, Transcriptional activity of Aire WT or ΔCARD mutant in 4D6 cells as done in Fig. 6a. Data are presented as mean ± s.d. (three biological replicates). *P*-values (two-tailed unpaired t-test) were calculated vs. WT Aire (+Dox). ****$P$ < 0.0001; ns $P$ > 0.05. For clarity, data for WT Aire transcriptional activity are reproduced from Fig. 6a.

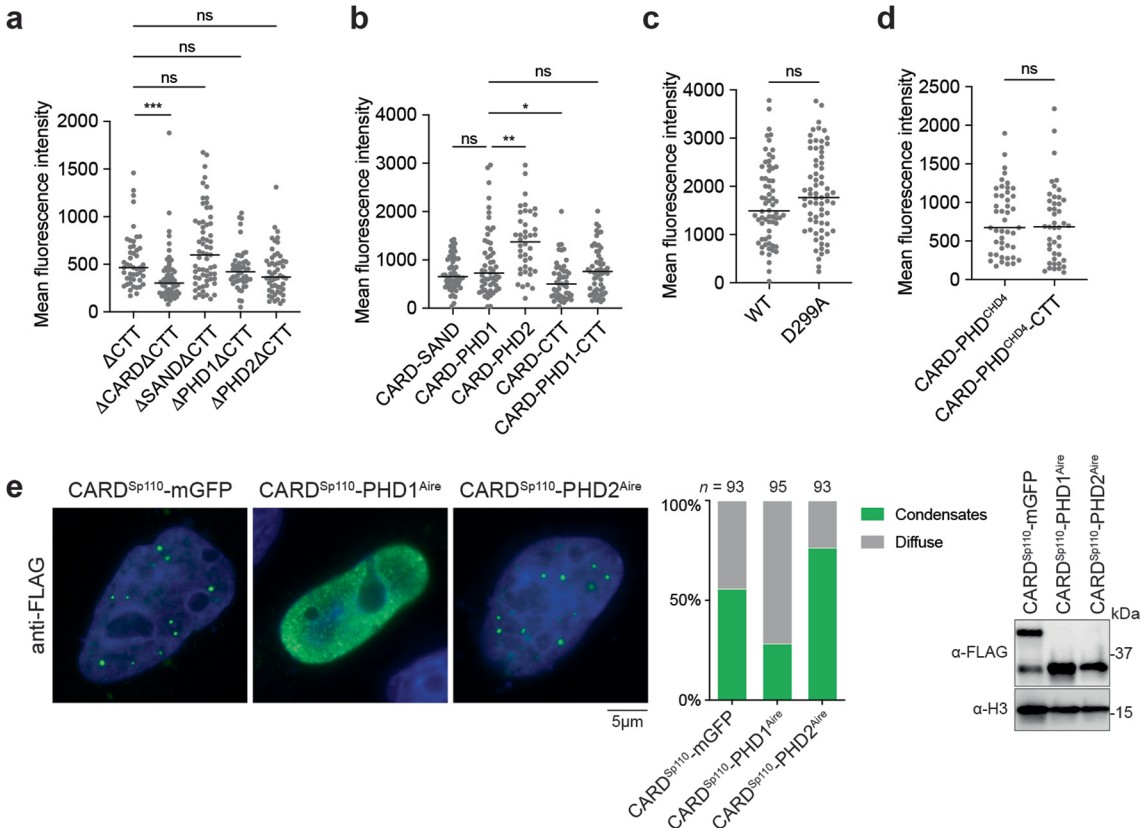

**Extended Data Fig. 8 | PHD1 binding to H3K4me0 is sufficient to inhibit Aire CARD polymerization. a-d**, Mean fluorescence intensity of nuclei examined from Fig. 5a,c,e,g, respectively. *P*-values (Kruskal-Wallis test with Dunn's multiple comparisons test for **a** and **b** and two-tailed Mann-Whitney test for **c** and **d**) were calculated in comparison to the control groups indicated in each graph. *P < 0.05;

**P < 0.01; ***P < 0.001; ns P > 0.05. **e**, Representative IF images of FLAG-tagged Sp110 CARD fused with mGFP (monomeric GFP), PHD1 or PHD2 from mouse Aire. Experiments and analyses were done as in Fig. 5a and are representative of three independent experiments.

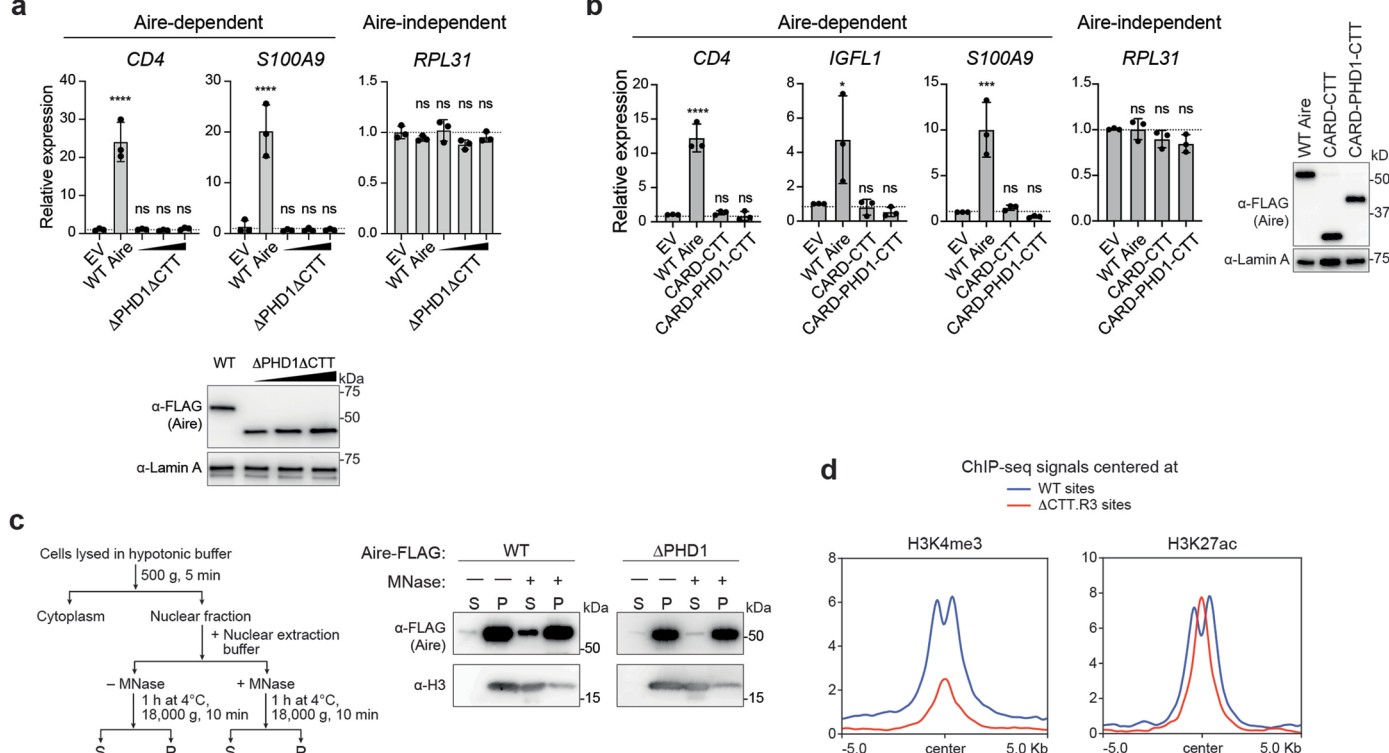

**Extended Data Fig. 9 | PHD1 is required for the chromatin anchoring of Aire.**
**a**, Transcriptional activity of mouse Aire WT or ΔPHD1ΔCTT mutant measured by RT-qPCR. Data are presented as mean ± s.d. (three biological replicates). *P*-values (one-way ANOVA with Dunnett's multiple comparisons test) were calculated vs. empty-vector transfected cells (EV). ****$P < 0.0001$; ns $P > 0.05$. **b**, Transcriptional activity of mouse Aire WT or CARD-fusions measured by RT-qPCR. Data are presented as mean ± s.d. (three biological replicates). *P*-values (one-way ANOVA with Dunnett's multiple comparisons test) were calculated vs. empty-vector transfected cells (EV). ****$P < 0.0001$; ***$P < 0.001$; ns $P > 0.05$. **c**, Chromatin fractionation analysis of Aire WT and ΔPHD1. Left: schematic of chromatin fractionation analysis. 293T cells were transfected with mouse Aire expressing plasmids for 48 h before harvesting. Using a nuclear extraction buffer to preserve Aire-interaction partners, we examined Aire solubility with or without micrococcal nuclease (MNase) treatment. MNase solubilizes chromatin

along with proteins associated with chromatin. Chromatin (inferred by levels of histone H3) exclusively partitions in the insoluble (P) fraction of nuclear extracts until MNase is added, releasing chromatin into the soluble (S) fraction. WT Aire remains mostly insoluble until chromatin is solubilized with MNase treatment, which releases a portion of Aire. A substantial portion of WT Aire remained in the P fraction after MNase digestion, possibly due to large Aire polymers that cannot be dissolved in our nuclear extraction buffer. In contrast, ΔPHD1 showed no interaction with chromatin, existing solely in the P fractions regardless of MNase treatment, indicating the formation of insoluble aggregates that are not associated with chromatin. **d**, Average H3K27ac and H3K4me3 ChIP-seq profiles (normalized CPM) centered at WT-preferred sites vs. ΔCTT.R3-preferred sites ($n$ = top and bottom 500 peaks from Fig. 4a, respectively). Both H3K27ac and H3K4me3 ChIPs were performed in 4D6 cells prior to Aire expression. Data from **a-c** are representative of at least three independent experiments.

# Reporting Summary

## Statistics

For all statistical analyses, confirm that the following items are present in the figure legend, table legend, main text, or Methods section.

| n/a | Confirmed | |
|---|---|---|
| ☐ | ☒ | The exact sample size (*n*) for each experimental group/condition, given as a discrete number and unit of measurement |
| ☐ | ☒ | A statement on whether measurements were taken from distinct samples or whether the same sample was measured repeatedly |
| ☐ | ☒ | The statistical test(s) used AND whether they are one- or two-sided *Only common tests should be described solely by name; describe more complex techniques in the Methods section.* |
| ☒ | ☐ | A description of all covariates tested |
| ☐ | ☒ | A description of any assumptions or corrections, such as tests of normality and adjustment for multiple comparisons |
| ☐ | ☒ | A full description of the statistical parameters including central tendency (e.g. means) or other basic estimates (e.g. regression coefficient) AND variation (e.g. standard deviation) or associated estimates of uncertainty (e.g. confidence intervals) |
| ☐ | ☒ | For null hypothesis testing, the test statistic (e.g. *F*, *t*, *r*) with confidence intervals, effect sizes, degrees of freedom and *P* value noted *Give P values as exact values whenever suitable.* |
| ☒ | ☐ | For Bayesian analysis, information on the choice of priors and Markov chain Monte Carlo settings |
| ☒ | ☐ | For hierarchical and complex designs, identification of the appropriate level for tests and full reporting of outcomes |
| ☐ | ☒ | Estimates of effect sizes (e.g. Cohen's *d*, Pearson's *r*), indicating how they were calculated |

*Our web collection on statistics for biologists contains articles on many of the points above.*

## Software and code

Policy information about availability of computer code

| Data collection | No custom computer codes were used. Please see Methods section for details of all modes of data collection for experiments. |
|---|---|
| Data analysis | No custom computer codes were used. Please see Methods section for details of all data analyses performed for experiments. |

For manuscripts utilizing custom algorithms or software that are central to the research but not yet described in published literature, software must be made available to editors and reviewers. We strongly encourage code deposition in a community repository (e.g. GitHub). See the Nature Portfolio guidelines for submitting code & software for further information.

## Data

Policy information about availability of data

All manuscripts must include a data availability statement. This statement should provide the following information, where applicable:
- Accession codes, unique identifiers, or web links for publicly available datasets
- A description of any restrictions on data availability
- For clinical datasets or third party data, please ensure that the statement adheres to our policy

The accession numbers for the next generation sequencing data reported in this paper is Gene Expression Omnibus: GSE243825. Any additional information required to reanalyze the data reported in this paper is available from the lead contact corresponding author upon request.

# Research involving human participants, their data, or biological material

Policy information about studies with human participants or human data. See also policy information about sex, gender (identity/presentation), and sexual orientation and race, ethnicity and racism.

| | |
|---|---|
| Reporting on sex and gender | *Use the terms sex (biological attribute) and gender (shaped by social and cultural circumstances) carefully in order to avoid confusing both terms. Indicate if findings apply to only one sex or gender; describe whether sex and gender were considered in study design; whether sex and/or gender was determined based on self-reporting or assigned and methods used.*<br>*Provide in the source data disaggregated sex and gender data, where this information has been collected, and if consent has been obtained for sharing of individual-level data; provide overall numbers in this Reporting Summary. Please state if this information has not been collected.*<br>*Report sex- and gender-based analyses where performed, justify reasons for lack of sex- and gender-based analysis.* |
| Reporting on race, ethnicity, or other socially relevant groupings | *Please specify the socially constructed or socially relevant categorization variable(s) used in your manuscript and explain why they were used. Please note that such variables should not be used as proxies for other socially constructed/relevant variables (for example, race or ethnicity should not be used as a proxy for socioeconomic status).*<br>*Provide clear definitions of the relevant terms used, how they were provided (by the participants/respondents, the researchers, or third parties), and the method(s) used to classify people into the different categories (e.g. self-report, census or administrative data, social media data, etc.)*<br>*Please provide details about how you controlled for confounding variables in your analyses.* |
| Population characteristics | *Describe the covariate-relevant population characteristics of the human research participants (e.g. age, genotypic information, past and current diagnosis and treatment categories). If you filled out the behavioural & social sciences study design questions and have nothing to add here, write "See above."* |
| Recruitment | *Describe how participants were recruited. Outline any potential self-selection bias or other biases that may be present and how these are likely to impact results.* |
| Ethics oversight | *Identify the organization(s) that approved the study protocol.* |

Note that full information on the approval of the study protocol must also be provided in the manuscript.

# Field-specific reporting

Please select the one below that is the best fit for your research. If you are not sure, read the appropriate sections before making your selection.

☒ Life sciences          ☐ Behavioural & social sciences          ☐ Ecological, evolutionary & environmental sciences

For a reference copy of the document with all sections, see nature.com/documents/nr-reporting-summary-flat.pdf

# Life sciences study design

All studies must disclose on these points even when the disclosure is negative.

| | |
|---|---|
| Sample size | Sample sizes varied depending on experiments and are noted in the figure legends and Methods. |
| Data exclusions | For microscopy experiments examining Aire localization, we excluded cells that did not appear to express Aire (cells with fluorescence signals dimmer than background levels). For next-generation sequencing analyses, please see Methods for details of excluding background noises and blacklisted regions. |
| Replication | All experiments were performed at least 3 times. For qPCR data, in addition to 3 biological replicates, 2-3 technical replicates were performed. Only experimental data that were successfully replicated in all attempts are reported. See Figure legends for details. |
| Randomization | For microscopy, images were taken at random locations on the cover slips. For RNA-FISH coupled to IF experiments, random nuclear positions were generated to determine whether Aire foci was significantly associated with RNA FISH spots. Other experiments in this study were not subjected to randomization as the identity of the samples are predetermined during experiments; the experimental results would not be interpretable if these samples were randomized. |
| Blinding | For microscopy of cells treated with p300/CBP inhibitors or BRD4 inhibitors, samples were blinded while imaging. For all other experiments, there were no samples that could be blinded as the identity of the samples are predetermined during experiments. |

# Reporting for specific materials, systems and methods

We require information from authors about some types of materials, experimental systems and methods used in many studies. Here, indicate whether each material, system or method listed is relevant to your study. If you are not sure if a list item applies to your research, read the appropriate section before selecting a response.

## Materials & experimental systems

| n/a | Involved in the study |
|---|---|
| ☐ | ☒ Antibodies |
| ☐ | ☒ Eukaryotic cell lines |
| ☒ | ☐ Palaeontology and archaeology |
| ☒ | ☐ Animals and other organisms |
| ☒ | ☐ Clinical data |
| ☒ | ☐ Dual use research of concern |
| ☒ | ☐ Plants |

## Methods

| n/a | Involved in the study |
|---|---|
| ☐ | ☒ ChIP-seq |
| ☐ | ☒ Flow cytometry |
| ☒ | ☐ MRI-based neuroimaging |

## Antibodies

| | |
|---|---|
| Antibodies used | Antibodies used for immunofluorescence (IF) microscopy were mouse anti-FLAG (M2, Sigma-Aldrich, F1804), mouse anti-FLAG conjugated with FITC (M2, Sigma-Aldrich, F4049), rabbit anti-p300 (D8Z4E, Cell Signaling Technology, 86377S), rabbit anti-CBP (D6C5, Cell Signaling Technology, 7389S), rabbit anti-MED1 (Novus Biologicals, NB100-2574), Alexa Fluor® 488 AffiniPure donkey anti-mouse IgG (Jackson ImmunoResearch, 715-545-151), Alexa Fluor 647 AffiniPure donkey anti-rabbit IgG (Jackson ImmunoResearch, 711-605-152). Antibodies used for immunoblotting were rabbit anti-beta-actin (Cell Signaling Technology, 8457S), rabbit anti-HA (C29F4, Cell Signaling Technology, 3724S), mouse anti-FLAG-HRP (M2, Sigma-Aldrich, A8592), mouse anti-Lamin A (133A2, Cell Signaling Technology, 86846), mouse anti-Histone H3 (Cell Signaling Technology, 14269S), rabbit anti-Histone H3K27ac (D5E4, Cell Signaling Technology, 8173S), rabbit anti-Histone H3K18ac (D8Z5H, Cell Signaling Technology, 13998), rabbit anti-p300 (D8Z4E, Cell Signaling Technology, 86377S), rabbit anti-CBP (D6C5, Cell Signaling Technology, 7389S), rabbit anti-actyl-p300/CBP (Cell Signaling Technology, 4771S), anti-rabbit IgG-HRP (Cell Signaling Technology, 7074), anti-mouse IgG-HRP (Cell Signaling Technology, 7076). Antibodies used for chromatin immunoprecipitation mouse anti-FLAG (M2, Sigma-Aldrich, F1804), rabbit anti-Histone H3K27ac (D5E4, Cell Signaling Technology, 8173S), rabbit anti-Histone H3K4me1 (D1A9, Cell Signaling Technology, 5326S), rabbit anti-Histone H3K27me3 (C36B11, Cell Signaling Technology, 9733S), rabbit anti-Histone H3K4me0 (Active Motif, 91317), rabbit anti-p300 (D2X6N, Cell Signaling Technology, 54062), rabbit anti-BRD4 (E2A7X, Cell Signaling Technology, 13440), spike-in antibody (Active Motif, 61686). |
| Validation | All primary antibodies were validated previously by the manufacturer. Citations of studies using these antibodies and user ratings are provided on the manufacturer's websites |

## Eukaryotic cell lines

Policy information about cell lines and Sex and Gender in Research

| | |
|---|---|
| Cell line source(s) | Human embryonic kidney cells 293T were a generous gift Dr. Dan Stetson, University of Washington; Seattle, WA. 4D6 cells were originally derived from human thymic epithelium from children undergoing cardiac surgery and provided to the laboratory of Diane Mathis. |
| Authentication | No form of authentication was used for these cell lines. |
| Mycoplasma contamination | These cells were verified to be mycoplasma free by using the MycoAlert Mycoplasma Detection Kit (Lonza, Cat. No. LT07-318). |
| Commonly misidentified lines (See ICLAC register) | Unfortunately, we were not aware that 4D6 cell line was on the "Cross Contaminations Distribution List" as contaminated with human liver carcinoma cell line until we saw version 13 from April 2024 in the Author Guidance. We received these as a gift from Dr. Diane Mathis, who's lab has used this cell line routinely to study Aire. Many of our results have corroborated with the Mathis lab's previous findings and can be recapitulated in other cell lines including 293T. |

## Plants

| | |
|---|---|
| Seed stocks | *Report on the source of all seed stocks or other plant material used. If applicable, state the seed stock centre and catalogue number. If plant specimens were collected from the field, describe the collection location, date and sampling procedures.* |
| Novel plant genotypes | *Describe the methods by which all novel plant genotypes were produced. This includes those generated by transgenic approaches, gene editing, chemical/radiation-based mutagenesis and hybridization. For transgenic lines, describe the transformation method, the number of independent lines analyzed and the generation upon which experiments were performed. For gene-edited lines, describe the editor used, the endogenous sequence targeted for editing, the targeting guide RNA sequence (if applicable) and how the editor was applied.* |
| Authentication | *Describe any authentication procedures for each seed stock used or novel genotype generated. Describe any experiments used to assess the effect of a mutation and, where applicable, how potential secondary effects (e.g. second site T-DNA insertions, mosaicism, off-target gene editing) were examined.* |

# ChIP-seq

## Data deposition

☒ Confirm that both raw and final processed data have been deposited in a public database such as GEO.

☒ Confirm that you have deposited or provided access to graph files (e.g. BED files) for the called peaks.

| | |
|---|---|
| Data access links<br>*May remain private before publication.* | The accession numbers for the ChIP-seq data reported in this paper is Gene Expression Omnibus: GSE243825 |
| Files in database submission | Raw and processed data (bigwig) were uploaded. |
| Genome browser session<br>(e.g. UCSC) | No longer applicable. |

## Methodology

| | |
|---|---|
| Replicates | All experiments were performed 2-3 times. |
| Sequencing depth | Pair-end 150bp reads, 30-40M reads per sample. |
| Antibodies | mouse anti-FLAG (M2, Sigma-Aldrich, F1804), rabbit anti-Histone H3K27ac (D5E4, Cell Signaling Technology, 8173S), rabbit anti-Histone H3K4me1 (D1A9, Cell Signaling Technology, 5326S), rabbit anti-Histone H3K27me3 (C36B11, Cell Signaling Technology, 9733S), rabbit anti-Histone H3K4me0 (Active Motif, 91317), rabbit anti-p300 (D2X6N, Cell Signaling Technology, 54062), spike-in antibody (Active Motif, 61686). |
| Peak calling parameters | Peak calling was performed using MACS2 (v2.2.7.1) with the following parameters: macs2 callpeak -f BAMPE -B -g 3.2e+9 --keep-dup 1 --SPMR --nomodel --extsize 250 -q 0.05 --cutoff-analysis. See Methods for more details. |
| Data quality | Alignments with MAPQ>1 were filtered. Replicates were examined to ensure reproducibility. Peaks were called using the filter FDR<0.05. |
| Software | FASTQC (v0.11.3), Trimmomatic (v0.36), deepTools (v3.5.1), WiggleTools (v1.2.2), bedGraphToBigWig (v366), Integrative Genomics Viewer (IGV, v2.15.1), bwa (v0.7.17), Samtools (v1.6), MACS2 (v2.2.7.1), DiffBind R package (v3.8.4) |

# Flow Cytometry

## Plots

Confirm that:

☒ The axis labels state the marker and fluorochrome used (e.g. CD4-FITC).

☒ The axis scales are clearly visible. Include numbers along axes only for bottom left plot of group (a 'group' is an analysis of identical markers).

☒ All plots are contour plots with outliers or pseudocolor plots.

☒ A numerical value for number of cells or percentage (with statistics) is provided.

## Methodology

| | |
|---|---|
| Sample preparation | Cells were treated with 1 µg/ml Dox 24 hrs prior to sorting or acquisition to induce EGFP-P2A-Gal4DBD-AireCTT expression. In some experiments, cells were also treated with DMSO, A-485 (3 µM), or dCBP-1 (0.25 µM) for 24hrs prior to acquisition. |
| Instrument | Cells for CRISPR screening were sorted on SH800S Cell Sorter (Sony Biotechnology). Cells for flow cytometric analyses were acquired on FACSCanto II-AR (BD Biosciences). |
| Software | Flow cytometric data were analyzed using FlowJo (v10). |
| Cell population abundance | >98% cells were EGFP+ (Gal4DBD-AireCTT+) post 24 hrs of Dox treatment. |
| Gating strategy | Cells sorted for CRISPR screening were first gated on singlets based on FSC/SSC, and then gated on live (DAPI−) EGFP+ cells; top-5% and bottom-5% of the population were sorted based on mKate2 expression. Cells for flow cytometric analyses were first gated on singlets based on FSC/SSC. |

☒ Tick this box to confirm that a figure exemplifying the gating strategy is provided in the Supplementary Information.

