## [Peer Review File · Nature Immunology]

Mechanism for controlled assembly of transcriptional condensates by Aire

Corresponding Author: Dr Sun Hur

Version 0:

Decision Letter:

2nd Apr 2024

Dear Sun,

Good to hear from you and thank you for transferring your manuscript entitled "A new mode of transcriptional hub assembly by autoimmune regulator Aire". I had contacted the Nature editorial assistant to obtain the reviewer comments from the earlier round of external review when considered there; hence the delay in my getting back to you. I discussed the study with my editorial colleagues here at Nature Immunology, and I remember seeing part of this story presented on a poster at the PCMM retreat last September. We are very interested in the possibility of publishing your study in Nature Immunology, along the lines proposed in your point-by-point response for a revised manuscript.

Please note, we are over-riding the concerns posed by referees regarding the use of the 4D6 thymic epithelial cell line, as we recognize that catching Aire in the act in primary cells is actually quite difficult. We will likely take the liberty of replacing one or more of these referees with another referee who may be better versed in AIRE biology in thymic tolerance.

We therefore invite you to revise your manuscript, noting the response to that earlier round of review. Please highlight all changes in the manuscript text file in Microsoft Word format.

* If you have not done so already please begin to revise your manuscript so that it conforms to our Article format instructions at <http://www.nature.com/ni/authors/index.html>. Refer also to any guidelines provided in this letter.

* Please include a revised version of any required reporting checklist. It will be available to referees to aid in their evaluation of the manuscript goes back for peer review. They are available here:

Reporting summary:

When submitting the revised version of your manuscript, please pay close attention to our <https://www.nature.com/nature-portfolio/editorial-policies/image-integrity> Digital Image Integrity Guidelines and to the following points below:

Link Redacted

We hope to receive your revised manuscript within two months. If you cannot send it within this time, please let us know. We will be happy to consider your revision so long as nothing similar has been accepted for publication at Nature Immunology or published elsewhere.

Nature Immunology is committed to improving transparency in authorship. As part of our efforts in this direction, we are now requesting that all authors identified as 'corresponding author' on published papers create and link their Open Researcher and Contributor Identifier (ORCID) with their account on the Manuscript Tracking System (MTS), prior to acceptance. ORCID helps the scientific community achieve unambiguous attribution of all scholarly contributions. You can create and link your ORCID from the home page of the MTS by clicking on 'Modify my Springer Nature account'. For more information please visit www.springernature.com/orcid.

Kind regards,

Laurie

Laurie A. Dempsey, Ph.D.
Senior Editor
Nature Immunology
l.dempsey@us.nature.com
ORCID: 0000-0002-3304-796X

Author Rebuttal letter:

Revision Summary

We thank all three reviewers for their insightful comments and constructive criticisms. To address each and all of their concerns, we performed additional experiments and more careful analyses (resulting in 29 new subfigures and 5 new supplementary tables). To incorporate both new experimental data and reviewers's suggestions, we have also extensively edited the manuscript. In particular, the main text was expanded to clearly lay out our experimental strategies, rationales, and interpretation. We believe that these changes strengthen the manuscript and further support the model we described in the original manuscript.

Due to the extensive changes within the manuscript text, we did not mark all changes in the main text. Only the areas directly related to the reviewers's concerns are colored blue.

Summary of new and modified figures

1. Fig. 1 was significantly expanded and was thus split into Fig 1 and Fig 2. Thus, original Fig 2-5 are now new Fig 3-6.
2. Fig 1A, 1B: moved from the original Figure S1A and S1B
3. Fig 1C: new experiments and analysis (genomic data, effect of Aire on Aire-bound region)
4. Fig 1D: new experiments and analysis (genomic data, H3K27ac, H3K4me1 and nascent RNA +/- Aire were added)
5. Fig 2B: new analysis added (imaging data, quantitation of nuclear intensity)
6. Fig 2D: new analysis added (imaging data, quantitation of nuclear intensity)
7. Fig. 3F: new experiment (CTT \hat{c} CBP affinity)
8. Fig. 3G: experiment repeated with recombinant protein, replacing the pull-down result with lysate
9. Fig. 3I: new analysis added (imaging data, quantitation of nuclear intensity and foci volume)
10. Fig. 3J: new experiments and analysis (genomic data, effect of A-485)
11. Fig. 5A: experiments repeated to ensure even protein expression (imaging, western blot)
12. Fig. 5B: new analysis (imaging, WT vs. \hat{c} PHD1 foci frequency)
13. Fig. 5C: experiments repeated to ensure even protein expression (imaging, western blot)
14. Fig. 6E: new experiments and analysis (genomic data, WT vs. \hat{c} TTT site comparison)
15. Fig. 6F: model modified to improve clarity
16. Fig. S1D, S1E, S1F: new experiments & analysis (genomic data, ATAC-seq, effect of Aire on

Aire-bound region)

17. Fig. S3B: experiments repeated to ensure even protein expression (imaging, western blot)
18. Fig. S3C: new experiments & analysis (NMR data)
19. Fig. S6B: new analysis (imaging data, quantification of foci volume)
20. Fig. S6D: new analysis (imaging data, nuclear intensity)
21. Fig. S6H: new experiments and analysis (RT-qPCR, BRD4 inhibitor analysis)
22. Fig. S6I: new experiments and analysis (imaging, BRD4 inhibitor analysis)
23. Fig. S7A: new analysis (genomic data, Aire ChIP peak comparison)
24. Fig. S8A-D: new analysis (imaging data, nuclear intensity comparison)
25. Fig. S9A: new experiments (RT-qPCR of Δ PHD1 Δ CTT)
26. Fig. S9B: new experiments (RT-qPCR of CARD-CTT and CARD-PHD1-CTT)
27. Fig. S9C: trimmed down to the comparison of WT vs. Δ PHD1. Other unnecessary figures were removed for clarity.
28. Fig. S9D: new experiments & analysis (genomic data, histone mark ChIP-seq comparison at WT vs. Δ CTT sites)
29. Supp. Tables 1-5: new

In addition, there are a few figure panels that were rearranged within figures or moved between figures and supplementary figures to improve clarity.

Below, we provide detailed point-by-point response to reviewers's concerns using the following color codes.

â€¢ Green: Reviewer's comments.

â€¢ Black: Our detailed response.

â€¢ Blue: Relevant text in the revised manuscript.

Referee #1 (Remarks to the Author):

In this manuscript, Huoh et al demonstrate that the transcriptional regulator Aire forms the nuclei foci via interaction with both p300/CBP and histone tails, as well as polymerization of itself. The authors identified the CTT domain that directly interacts with p300/CBP. Disruption of the Aire-p300 interaction results in loss of nuclei foci and concomitant loss of gene activation. The authors show that the PHD1 domain, which interacts with histone H3K4me0, is required for proper foci formation at the target genes. Notably, without the PHD1 domain, Aire undergoes aggregation outside the target genes, resulting in loss of gene activation. These results support the authors's model that the nuclei foci formation involving interaction with p300/CBP and histone tails is required for activation of Aire's target genes.

Overall, the presented data supports some of the authors's claims regarding the mechanism of Aire nuclear foci formation within the cell line that they used. However, the authors have not addressed whether the proposed model is applicable to the physiological condition, such as immune tolerance, which is critical to the significance of this study. Furthermore, the manuscript requires more in-depth analyses on genomic data to provide mechanistic insights, as outlined below. Without these data and analyses, it is difficult to support the publication of this manuscript in Nature.

> We thank the reviewer for the thoughtful comments. We want to emphasize that our 4D6 cellular system faithfully recapitulates many features of Aire in its native cellular environment (mTECs), including its expression level (see Figure 1A), nuclear foci formation (PMIDs 20085707, 18414681, 19487417, 17974569, 15964547), Aire's enhancer localization (PMID 28135252; Figure 1B, S1A), Aire's stochastic nature (PMIDs 26237550, 26237553, 25224068; Figure 1E, 1F, S2B), and loss-of-function phenotypes of APS-1 patient mutations (PMIDs 18414681, 10677297; Figure S1C). We believe that these features validate our 4D6 cellular system and make our mechanistic studies physiologically relevant.

We have also performed additional genomic data analyses, as suggested by the reviewer. Our new data further strengthen our original conclusions and we have revised the manuscript accordingly.

Major concerns:

1. The authors should provide more in-depth analysis on genome-wide Aire occupancy data. For example, does the level of Aire occupancy correlate with up-regulation of the nearest genes? Are the binding sites of R3 mutant co-localized with any histone marks, such as H3K4me1, me2, and me3, which are generally the targets of the PHD domains? Are the levels of H3K27Ac increased by WT Aire but not by the R3 mutant?

> We thank the reviewer for the insightful suggestions. We have performed nascent RNA-seq and various histone mark ChIP-seq before and after Aire expression to comprehensively address the reviewer's comment. The result showed that Aire occupancy correlates with the up-regulation of enhancer RNA at Aire-bound loci (nucleosome-free regions highly occupied by WT Aire, new Fig. 1C, top panels), but not at the control sites (equally nucleosome-free but free of WT Aire). Furthermore, enhancer RNA

expression was downregulated by A-485 at Aire-bound sites, and this effect was more significant than in control sites (new Fig. 3J). This transcriptional activation is not accompanied by an increase in H3K27ac, H3K4me1, H3K4me3 and ATAC signals (new Fig. 1C, bottom panels; new Fig. S1E), suggesting that Aire's mode of action is independent of these histone marks (but dependent on CBP/p300). This was evident when examining changes in the histone marks aggregated over all Aire-bound sites (new Fig. 1C), and individual representative sites in new Fig. 1D and S1F.

We also performed systematic comparison of histone marks at Aire-preferred vs. Δ CTT.R3-preferred sites (new Fig. 6E and S9D). The results showed that Aire-preferred sites have higher levels of H3K27ac, H3K4me1, while Δ CTT.R3-preferred sites have higher levels of H3K4me0. This is consistent with our model that Aire CTT helps recruit Aire to active loci pre-occupied with CBP/p300, while PHD1's intrinsic preference for H3K4me0 (see Fig. 5D and PMID: 18292755, 18840680, 20615959, 20190764) plays a more important role in Δ CTT.R3 targeting.

2. Regarding the role of Aire at super-enhancers, it will be helpful to test whether Aire occupancy and TAD-like activity are affected by BRD4 inhibitor JQ1 or degrader dBET6, as H3K27ac-bound BRD4 plays a key role in super-enhancers (PMID: 23582323, 28673542).

> A previous report has shown that JQ1 treatment does not impair the transcriptional activity of Aire (PMID: 29463681). Consistent with this report, we also found that neither JQ1 nor dBET6 inhibits Aire's transcriptional activity (new Fig. S6H) or condensate formation (new Fig. S6I). Quantitative comparison of Aire foci volume suggests that Aire foci are slightly bigger in the presence of the Brd4 inhibitors, which may be due to the fact that Brd4 inhibition leads to increased CBP/p300 occupancy at super-enhancers (PMID: 30428346). Note that dBET6 induced near complete degradation of Brd4. Altogether, these observations suggest that, while Brd4 plays an important role in super-enhancer activity, Aire's targeting and actions are independent of Brd4. Instead, our data suggest that Aire's super-enhancer targeting is dependent on CBP/p300.

New Fig S6H

New Fig S6I

3. In Figure 5, a simple interpretation of the data would be that the PHD1 domains interact with H3K4 methylation (mono, di, or tri) near H3K27ac sites. It will be helpful to show IF of these histone marks and compare with Aire WT or Δ PHD1 mutant foci.

> Consistent with the reviewer's prediction, H3K4me1, but not H3K4me0, was highly enriched at/near Aire-bound sites, as measured by our new ChIP-seq (new Fig. 6E). However, we were unable to observe significant enrichment of H3K4me1 at Aire foci by imaging (top panel). This was also the case with H3K27ac, which was also enriched at Aire-bound sites by ChIP-seq, but not by IF (bottom panel). This is in contrast to CBP/p300, which showed strong enrichment at Aire foci by IF as well as at Aire-bound site by ChIP-seq (Fig. 1H, 4A, 4B). We suspect that the strong IF signal of CBP/p300 at Aire foci is in part due to the intrinsic ability of CBP/p300 to form condensates themselves (PMID: 33651988), which may synergize with Aire's polymerization activity. Regardless, we believe that our new ChIP-seq data address the reviewer's question on the histone marks at the Aire's genomic target sites.

4. What are the dynamics of Aire foci formation? How long do they stably form the foci, and are the foci merged temporarily?

> Despite our extensive efforts, we found that fusing small fluorescent proteins (e.g. mini GFP, monomeric GFP, Halo, SNAP) to N- or C-terminus or internal regions of Aire abrogates Aire's functions (see below for two examples). This aligns with our conclusion that the formation of functional Aire foci relies on a delicate balance between multiple regulatory mechanisms, and perturbations of any of those regulatory elements, for example by fusion of a fluorescent proteins, can upset this balance and result in dysfunctional aggregate formation. Notably, to the best of our knowledge, there is no report of successful live cell imaging of functional Aire tagged with a fluorescent protein. While we continue to explore

strategies to label Aire without compromising its functions, we believe that investigating Aire foci dynamics is outside the scope of the current manuscript.

We previously showed that GFP-Aire forms foci, but is transcriptionally inactive and has a dominant negative effect when co-expressed with WT Aire (left, PMID: 32242017).

We also attempted to add a SNAP tag to the C-terminus or within Aire to no avail. For example, inserting a SNAP-tag in between the PHD2 and CTT domain (SNAP-fusion, left) still formed foci but was also transcriptionally inactive.

Minor concerns:

1. The authors should provide the supplementary tables for the whole results of CRISPR- and proteomics-based screening shown in Figure 2B,C.

> We have included the CRISPR- and proteomics-based data in Tables S3, S4.

2. Regarding Figure 2, the authors should provide the evidence that the mutation within the CTT R2/R3 domain results in disruption of interaction with p300/CBP using a recombinant p300/CBP protein. This result will be required to claim the role of this domain in direct interaction with p300/CBP.

> We performed the binding assay using purified recombinant proteins, and confirmed the original finding based on cellular pull-down (new Fig. 3G).
Referee #2 (Remarks to the Author):

The manuscript investigates the mechanism of the assembly of biomolecular condensates by the Autoimmune Regulator Aire, a key nuclear regulator of thymus cell function. The authors performed mutagenesis of Aire, imaging of Aire condensates, and ChIP-Seq and qPCR analyses to measure Aire genomic binding and transcriptional activity. The key results of the manuscript appear the following: i) Ectopically expressed Aire forms condensates in a thymic epithelial cell line, and the condensates overlap p300 and loci controlled by Aire; ii) Aire condensate assembly requires the N-terminal CARD domain and the C-terminal CTT domain; iii) The CTT interacts with CBP/p300, and perturbation of CBP/p300 or the CTT interaction inhibits Aire condensation; iv) The CTT and CARD are also necessary for genomic recruitment of Aire, v) PHD1, another domain in Aire, suppresses condensate formation by transgenic CARD, but not by a CARD-CTT chimera; vi) deletion of the PHD1 domain leads to loss of genomic recruitment (ChIP-Seq), but has a minor to no effect on Aire condensates. Based on the results, the authors propose a model that Aire condensate assembly is controlled at multiple levels, including the recruitment through CBP/p300 association to the genome, CARD polymerization, and is counterbalanced by dilution of Aire to chromatin through the interaction between the Aire PHD1 domain with H3K4me0.

Biomolecular condensates are liquid-like protein-nucleic acid compartments implicated in virtually all cellular processes (Shin et al. Science 2017, Banani et al., NRMCB 2017). Major current frontiers include the molecular basis of condensate assembly, the molecular basis of the selective partitioning of proteins and nucleic acids into specific condensates, and understanding emergent functions of condensates. The present manuscript proposes an interesting and potentially important mechanism, whereby portions of the same protein compete in facilitating and suppressing condensate assembly. The "regulation-by-dispersion" model, if appropriately substantiated, may provide important clues into the functions of various chromatin-reader domains in condensate assembly, and is likely to be useful to many investigators.

In general, the manuscript includes several elegant and compelling perturbation approaches, including mutagenesis of domains, mutagenesis of catalytic residues, biochemical inhibition and protein degradation. The experiments together provide strong support for several of the claims. To this reviewer, the "regulation-by-dispersion" model is the most interesting and unique insight, but is also the least substantiated, and some of the presented data are inconsistent with the model. There are also technical concerns with imaging data. Considering how unique and interesting the model is, I am generally supportive of publication in Nature if the data inconsistencies are resolved, and additional controls are provided. I also have several suggestions on improving the terminology, data presentation and interpretation.

Specific comments

1. The authors propose a "regulation-by-dispersion" model, whereby the PHD1 domain of Aire facilitates chromatin association of Aire through interaction with H3K4me0, and this counterbalances condensate nucleation at sites bound by p300/CBP. The key data supporting the model include: i) deletion of the PHD1 domain rescues condensate formation of the Aire mutant lacking the CTT domain (the latter of which is necessary for condensate formation) (Figure 4A); ii) Fusing the Aire PHD1 domain to the CARD domain suppresses the formation of condensates (Figure 4B); iii) Mutation of a key residue in the PHD1-CARD fusion rescues condensate formation (Figure 4C), iv) the PHD domain of CHD4, also known to bind H3K4me0, also suppresses CARD4 condensation (Figure 4D). These results are presented in the context of compelling data that Aire condensation is driven by the CARD domain and the interaction of the CTT domain of Aire with p300/CBP.

The model predicts that Aire recruitment is driven by the CTT and CARD, consistent with ChIP experiments (Figure 3A). The major question then is, why does deletion of the PHD1 domain abolish Aire recruitment to the genome (Figure 5B), and gene induction (Figure 5A)? All the domains (CARD and CTT), which the authors demonstrate to drive genomic recruitment are present in the DeltaPHD1 Aire mutant. Moreover, the authors show that wild type Aire is associated with super-enhancers and Aire condensates overlap with highly transcribed loci, and such sites are enriched in H3K4me1, suggesting that the endogenous substrate of the PHD1 is not even present there. In other words, why would deletion of a domain effect the recruitment of the protein at sites where its substrate is not even present? The authors' model predicts that in the absence of the PHD1 domain, the recruitment of Aire is dominated by the CTT-p300/CBP interaction, so the expectation is to find the DPHD1 mutant localizing at p300/CBP-bound sites. This clearly does not seem to be the case (Figure 5D).

The authors write: "These results suggest that PHD1 is the main driver bringing Aire to chromatin, which differs from the role of CTT and CARD in modulating Aire's target specificity." This speculation does not explain why Aire is then mostly bound at sites where the substrate of the PHD1 domain is not present (as the H3K4 residues are monomethylated).

> We thank the reviewer for the thoughtful summary. As noted by the reviewer, it remains a mystery why PHD1 is detached from the chromatin. One possibility is that the role of PHD1 in facilitating Aire's chromatin localization may be an indirect consequence of suppressing aberrant polymerization of Aire. Once spontaneously polymerized, Aire may be in a conformation that is no longer competent for chromatin binding. Alternatively, PHD1 may directly participate in target recognition by binding to lower affinity substrates, such as H3K4me1 (Figure 5D), that are nearby CBP/p300 clusters. Indeed, histone ChIP-seq showed that Aire's target sites (WT-preferred) displayed high levels of H3K4me1 nearby, while being depleted of H3K4me0 (Figure 6E, also see Figure S9D for other histone marks). In contrast, Aire's non-target sites, such as CTT.R3-preferred sites, had relatively higher levels of H3K4me0, but lower levels of H3K4me1 (Figure 6E). These possibilities are now discussed in the main text (page 10).

A few suggestions to help resolve this:

i) What happens to Aire target genes in cells expressing CARD-CTT, and CARD-PHD1-CTT? Are they induced in a level comparable by wild type Aire?

> Our new RT-qPCR analyses of target genes show that neither CARD-CTT nor CARD-PHD1-CTT is transcriptionally active (new Fig. S9B). This result suggests that SAND and/or PHD2 have functions, albeit not in regulation of CARD polymerization. In other words, Aire condensate formation is necessary, but not sufficient for its transcriptional activity.

ii) What recruits DPHDDCTT? (Figure 4A)? Does this construct bind the genome and induce Aire-target expression?

> Our new RT-qPCR showed that PHD1-CTT is also transcriptionally inactive (new Fig S9A), despite forming foci. This is consistent with the result that PHD1 deletion leads to a spontaneous formation of Aire foci that are detached from the chromatin and are thus transcriptionally inactive. This result, together with the one above, suggests that controlled polymerization of Aire is important. These points are now explicitly discussed in the main text.

iii) If the authors' model is correct, PHD1 deletion should lead to higher partitioning of Aire protein into condensates. This should be quantifiable using microscopy images. This experiments would provide a strong support for the authors model.

> We thank the reviewer for the insightful suggestion. We performed the suggested analysis and found that $\hat{\alpha}$ PHD1 indeed forms more frequent foci per nucleus (new Fig. 5B).

iv) Does methyltransferase inhibition lead to hub dispersion?

> We have performed the suggested experiments using MRK-740, which inhibits PRDM9 methyltransferase (MTase), and OICR-9429, which targets the common essential subunit of all MLL-family H3K4 MTases WDR5. However, neither inhibitor significantly increased the global H3K4me0 level or decreased H3K4me3 (see below, panel a). Due to significant toxicity associated with individual inhibitors, we were unable to explore the effects of multiple MTase inhibitors in combination. Note that there is no single inhibitor, to the best of our knowledge, that can inhibit all H3K4 MTases at once. Additionally, we also employed siRNA knockdown of WDR5 and RBBP5, essential scaffolding subunits of all MLL-family H3K4 MTases. Despite successful knockdown, we again did not observe a dramatic increase in H3K4me0 (see below, panel b). Finally, we investigated the potential impact of overexpressing the histone demethylase KDM5B, which showed significant decrease in H3K4me3/me1, yet did not increase the global H3K4me0 level (see below, panel c). Additionally, overexpressing KDM5B markedly reduced Aire expression (which is dependent on doxycycline), precluding us from analyzing Aire condensate formation.

In summary, none of the methods we used (inhibitor or siRNA against histone MTase, or overexpressing histone demethylase) resulted in a significant change in the global H3K4me0 level. Despite an extensive literature search, we found no example of successful perturbation of global H3K4me0 level, which is considered abundant and widespread across the genome. Altogether, we conclude that global alteration of H3K4me0 may not be feasible.

Western blots showing the levels of H3K4me0, H3K4me3 and dox-inducible FLAG-tagged hAire after MTase inhibitor treatment (a), transfection of siRNAs targeting essential common subunits of MLL-family H3K4 MTases WDR5 or RBBP5 (b), and transfection of plasmid expressing H3K4 demethylase KDM5B (c, left). Representative images showing low/no Aire expression in KDM5B-transfected cells (c, right). Dox-inducible hAire-expressing 4D6 cells were treated with drugs or transfected with siRNAs/plasmids for time indicated above each panel before harvesting cells for WB or staining.

2. The second major issue concern appropriate quantification of proteins in cells. It is expected that condensate assembly scales with protein concentration and that ectopically expressed transgenic proteins are prone to form condensates (Pappu and Mittag, Mol Cell 2022). The authors provide Western blot quantification of several proteins to show equal expression of various protein species across cell populations. Many of the presented images however show large heterogeneity in protein levels among cells, and in several instances the expression level in cells that are being compared are clearly different. In general, the authors either need to show representative images, or describe the heterogeneity in the population. Also, typical controls here include the quantification of the total fluorescence intensities across cells that are being compared.

> We thank the reviewer for the suggestion. For experiments where the expression levels of the proteins in comparison were not equivalent, we apologize for the oversight. We have repeated experiments to enable foci comparison at equivalent expression levels. We also provide quantitation of mean Aire fluorescence intensity per nucleus as the reviewer suggested. Overall, there is cell-to-cell variability in expression levels for any given protein (as is usually the case in tissue culture experiments, and in endogenous Aire in mTECs) but expression levels are generally similar. Most importantly, changes in the nuclear foci formation do not correlate with the slight differences in the expression level. That is, impaired condensate formation cannot be explained by reduced/increased expression of Aire or its variant.

i) The quantification in Figure 1F suggests that all cells expressing WT Aire contain foci. Looking at the cells in Figure 1E (bottom right), one out of five cells show intense foci, three show fewer and smaller foci, and one does not show any. Also, in the images in Figure 1F, the level of DCARD and DCTT appear clearly different.

Per reviewer's suggestion, we measured mean nuclear fluorescence intensity, which are now shown in most of the imaging data, including new Fig. 2B, 2D (original Fig 1F, 1H). As stated above, there are cell-to-cell variabilities in the Aire expression levels but they are generally similar and the nuclear foci formation do not correlate with the expression levels. For example, the key point of the manuscript is that $\hat{\alpha}$ CTT, $\hat{\alpha}$ CTT.R2 and $\hat{\alpha}$ CTT.R3 do not form condensates, while WT and $\hat{\alpha}$ CTT.R1 do. Expression levels of WT (foci-forming) and $\hat{\alpha}$ CTT (no foci) are equivalent (left panel). Expression level of $\hat{\alpha}$ CTT.R1 (foci-forming) is slightly lower than $\hat{\alpha}$ CTT, $\hat{\alpha}$ CTT.R2 or $\hat{\alpha}$ CTT.R3 (no foci) (right panel).

ii) Figure 2H and 2I. The Western blots and images are inconsistent. The green level in IF seems much lower for the D530K mutant, and the cells treated with inhibitors.

The mean nuclear fluorescence intensity analysis in new Fig. 3I and S6D show that Aire expression levels are largely similar. While D530K is slightly lower than WT Aire, this is inconsistent with the western blot analysis of the nuclear Aire (for which, we do not have an explanation for). Even in this case, comparison between WT and D530K with equivalent nuclear intensities show reduced foci size for D530K. For your reference, old Fig 2H and 2I are now Fig 3I and 3K supplemented by Fig S6B and S6D.

iii) Figure 4. The images seem inconsistent with the Western blots. For example, in Figure 4A DCARDDCTT: there is hardly any protein on the Western blot. Figure 4B. CARD-PHD1 and CARD-PHD2 levels are wildly different in the Western blot than the other constructs. Figure 4C has similar issues.

We repeated these experiments to have more similar levels of Aire variant expression for new Fig. 5A and 5C (old Fig 4A and 4B). We also quantified the mean nuclear fluorescence intensity per nucleus (new Fig. S8A-D). The results show that the expression levels of most variants are similar, albeit not across all the samples. Importantly, key comparison pairs—between CTT (no foci) and PHD1CTT (foci); between CARD-PHD1 (no foci) and CARD-PHD1-CTT (foci)—show similar levels of expression, which support our original conclusion.

3. Consistent with the notion that CBP/p300 play important roles in Aire targeting to active loci, both degrader dCBP-1 and catalytic inhibitor A-485 impaired Aire binding to SEs (Figures 3F and 3G). The other more pertinent question is: does the inhibitor treatment prevent binding at the CTT.R3-preferred sites?

> Our new result showed that the inhibitors do not affect Aire binding to CTT-preferred sites (see right).

4. Terminology. The authors use the term *hub* throughout the manuscript. While indeed the terminology of the field may not be completely settled, the clear direction is using the term *condensate* to refer to membraneless concentrates of various cellular biopolymers (Pappu and Mittag, Mol Cell 2022).

The first sentence of the abstract reads: *Transcriptional hubs have emerged as critical structural components in gene expression and regulation 1-4*. The primary research papers referenced for this statement (Cho et al., Science 2018; Ma et al., Mol Cell 2018) both use the term *condensate* and not *hub*.

The authors are encouraged to help the field by using the terminology accepted and used by most investigators, and the papers they themselves cite.

> We appreciate the suggestion. We now use *condensate* in place of *hub* throughout the manuscript, except when we specifically refer to the hub-like structure, where multiple genomic regions come in contact with Aire. This change was not highlighted in the manuscript.

5. The manuscript is hard to read. This is in large part because the authors in many cases just appear to interpret some feature of the data, without explaining the experiment, and what part of the data/result is being interpreted.

i) The requirement of CTT for Aire hub formation mirrored CTT's TAD-like activity (Figures 1H and S3D). What is the assay? What does *mirrored* mean?

> We now write:

The importance of R2 and R3, but not R1, in CBP/p300 binding correlated with their significance in Aire condensate formation (Figure 2D). Old Figure 1H is now new Figure 2D.

ii) Mutations in the putative CTT interface (A536D and D530K) compromised Aire foci formation (Figures 2H and S6A). What is the experiment? Is it a knock-in system, is it ectopic expression? Please explain in the text.

> They were ectopically expressed by plasmid transfection. We now explain the experiment by writing *Transient expression of WT vs. A536D or D530K showed that the mutations significantly lowered the transcriptional activities of Aire (Figure 3H) and the AD reporter activity of Gal4DBD-CTT (Figure S6A). IF analysis also showed that A536D was diffuse in most cells (Figure 3I). Although D530K formed*

condensates at similar frequency as WT Aire, the sizes of individual condensates were smaller than WT Aire condensates, regardless of whether comparing cells showing equivalent total nuclear intensity (Figure 3I) or not (Figure S6B). These results show that both A536D and D530K are impaired in Aire condensate formation, albeit to differing degrees. Old Figures 2H and S6A are now new Figures 3I and S6B.

iii) Deletion of R2 or R3 abrogated Aire's transcriptional activity regardless of the target genes (Figure 1I). CTT-R1, on the other hand, showed gene-specific behaviors, suggesting more complex functions for R1 (Figure 1I). Please explain the assay in the text.

> We now added We thus asked how the same CTT truncations affect AD-like activity. We measured their transcriptional activities by RT-qPCR of a few target genes. Deletion of R2 or R3 (CTT-R2 and R3) completely abolished Aire's transcriptional activity regardless of the examined target genes (Figure 2E), recapitulating loss-of-function phenotype of the complete deletion of CTT (ΔCTT). On the other hand, CTT-R1 showed gene-specific behaviors, suggesting a more nuanced function for R1 (Figure 2E). An AD reporter assay using CTT-fused with Gal4 DNA-binding domain (Gal4DBD)⁵⁴ also highlighted the importance of R2 and R3 in CTT's AD-like activity (Figure S3E), although R1 was also important in this reporter assay. Old figure 1I is now new Figure 2E.

iv) ChIP-seq analysis of CBP/p300 revealed that Aire target sites exhibit a high density of CBP/p300 occupancy even before Aire expression (Figure 3A). What exactly is interpreted for this statement?

> We modified the text for clarification To investigate the role of CBP/p300 in Aire's enhancer targeting, we performed p300 ChIP-seq using Aire-inducible 4D6 cells, but prior to Aire expression to compare the pre-Aire distribution of p300 ChIP with Aire ChIP patterns. The result revealed a significant enrichment of p300 at Aire ChIP peaks, even before Aire expression (Figure 4A). Old Figure 3A is now new figure 4A.

v) ΔCTT on its own did not show a similar multimerization property (see Figure S5) or foci formation (Figure S3C). Figure S5 shows NMR data. Also, the authors need to explain the experimental system used in Figure S3C for readers to understand what is being interpreted.

>We included new data (new Figure S3C) and more information Unlike the CARD domain that spontaneously polymerizes in vitro⁴¹, isolated CTT behaved as a monomer. This was examined using solution NMR, specifically by measuring ¹⁵N T2 relaxation time (Figure S3C), which was consistent with those of other monomeric proteins of similar size. Furthermore, overexpression of isolated CTT tagged with an artificial protein (APEX2-GST) also did not display foci formation (Figure S3D). Old Figure S3C is now new figure S3D.

vi) Both dCBP-1 and A-485 significantly impaired Aire foci formation (Figure 2I). This statement is problematic, I do not see what is being interpreted as "impaired". Probably better quality images of representative cells, and quantification of fluorescence intensities in individual nuclei would solve this.

>We now present mean nuclear fluorescent intensity and more representative images to show their differences (old figure 2I is now expanded to new Figures 3K, S6D). We also modified the text to clarify what we meant by "impaired foci formation". It now reads "IF of Aire with and without CBP/p300 inhibitors showed that both dCBP-1 and A-485 treatment significantly decreased the size of Aire condensates, all without affecting the Aire expression level (Figure S6D)".

vii) Strikingly, ΔCTT.R3 exhibited a more dispersed genomic occupancy pattern, characterized by approximately tenfold more Aire-bound peaks and relatively uniform ChIP-seq signal intensities compared to WT Aire (Figure 3A). The heatmaps do not quantify number of peaks. If the authors want to claim "tenfold more peaks" they need to quantify the number of peaks.

> We now indicate the number of ChIP peaks in the main text, which are 1,363 and 12,893 for WT Aire and ΔCTT.R3, respectively. We also included a venn diagram showing the number of peaks identified from WT and mutant Aire ChIP (new Fig. S7A). The text now reads "Compared to WT Aire, ΔCTT.R3 strikingly exhibited a more dispersed genomic occupancy pattern, characterized by approximately tenfold more Aire-bound peaks (1,363 and 12,893 for WT Aire and ΔCTT.R3 ChIP peaks, Figure S7A) and relatively uniform ChIP-seq signal intensities compared to WT Aire (Figure 4A)".

viii) This amplification seems essential for establishing stable Aire-chromatin interactions, as shown by nuclear fractionation analysis. Both WT Aire and ΔCTT exhibited MNase-dependent solubilization, indicating chromatin binding akin to histones (Figure S7C). Please explain the data, and how the data

suggest these conclusions.

> Upon careful consideration, we have concluded that the result is not strong enough to make the argument. Thus, this sentence and corresponding data in the original Figure S7C (now Figure S9C) were removed in the revision. The main conclusion that Δ CTT binds chromatin in a more dispersed manner remains the same and is supported by ChIP-seq.

ix) Δ Strikingly, Δ CTT.R3 exhibited a more dispersed genomic occupancy pattern. What is being interpreted for this claim? What does Δ dispersed mean? That there are more peaks detected?

> We included some clarification Δ Compared to WT Aire, Δ CTT.R3 strikingly exhibited a more dispersed genomic occupancy pattern, characterized by approximately tenfold more Aire-bound peaks (1,363 and 12,893 for WT Aire and Δ CTT.R3 ChIP peaks, Figure S7A) and relatively uniform ChIP-seq signal intensities compared to WT Aire (Figure 4A).

x) Δ We thus asked whether CTT's ability to modulate CARD polymerization and thereby support foci formation stems from its TAD-like activity. Δ The experiment does not test the former.

> The text now reads Δ Aire CTT is known to have transcriptional activation domain (AD)-like activity^{1,3}. We thus asked how the same CTT truncations affect AD-like activity. We measured the transcriptional activities of R1, R2 and R3-deletion mutants by RT-qPCR of a few target genes. Deletion of R2 or R3 (CTT Δ R2 and Δ R3) completely abolished Aire's transcriptional activity regardless of the examined target genes (Figure 2E), recapitulating loss-of-function phenotype of the complete deletion of CTT (Δ CTT).

6. Minor comments

i) Figure 1B what is left and what is right? 3 color FISH? Please label the figure panel.

> Figure 1E (old Fig 1B) has four representative cells, each stained with Aire (IF) and two RNA-FISH probes (for RIC8A and SETD1B genes). The figures and legends were modified to clarify this issue.

ii) Figure S3B Western blot: why is there no signal in the CARD cells?

> We redid this experiment with adjusted DNA transfection conditions to have all variants, including Δ CARD, have similar concentration of protein expression (new Fig S3B). The data interpretation of these new results is the same.

iii) Δ A TAD-activity reporter assay using Gal4 DNA-binding domain (Gal4DBD)⁵² also showed the importance of R2 and R3 in the TAD-like activity of CTT (Figure S3D). Δ In these data, the R1 is also important for activity. Therefore, there appears to be a disconnect between activity and condensates.

> We agree with the reviewer and modified the text to Δ An AD reporter assay using CTT-fused with Gal4 DNA-binding domain (Gal4DBD)⁵⁴ also highlighted the importance of R2 and R3 in CTT's AD-like activity (Figure S3E), although R1 was also important in this reporter assay. Δ Old Figure S3D is now new Figure S3E.

iv) Δ In a departure from the conventional TAD functions⁵⁵! Actually, the authors data convincingly show that the function of the TAD (i.e. the CTT) is interaction with p300/CBP, which is the most conventional function of TADs!

> We apologized for the poor writing. We have now clarified the novelty of the manuscript, which is not that Aire TAD binds p300/CBP, but that Aire utilizes TAD to seed its polymerization and PHD1 to regulate this process. See the revised Discussion and Abstract.

v) Some referencing seems off. For example, Heintzman, Nature Genetics 2007 does not associate H3K4me0 with repressive chromatin!

> We revised the manuscript and changed the references including the one noted by the reviewer. Referee #3 (Remarks to the Author):

Huoh and Zhang et al perform a molecular dissection of Aire foci formation using an overexpression system in 4D6 cells. Aire foci overlap with nascent RNA FISH of known Aire targets and with other transcriptional coactivators. Some foci show multiple FISH foci associated with a single focus. CARD and disordered CTT are required for Aire foci formation. Using a CRISPR screen for reporter activity of CTT and a pulldown of GST-CTT, the authors highlight p300 as a functional direct interactor of CTT. The authors identify the interaction of CTT with TAZ2 in p300. Small molecule inhibition or degradation

of p300 decrease Aire focus formation. Removing the region of CTT that interacts with p300 causes a loss of correlation in Aire and p300 binding genome-wide. Deletion of CARD showed an intermediate effect on chromatin occupancy and p300 co-occupancy. Nuclear fractionation with and without MNase showed that CARD deletion caused Aire to have a weaker chromatin association compared to WT and other domain deletions. The authors had previously shown that CARD alone is sufficient for focus formation, but here show that deleting CTT prevents Aire focus formation. They find that deleting PHD1 restores focus formation but without activity. PHD1 deletion reduced chromatin occupancy by ChIP-seq and led to insoluble aggregation of Aire.

Overall, the authors correlate focus formation, chromatin occupancy, and expression for a variety of Aire domain deletions. The authors find interesting correlations in their experimental data, but overall, the conclusions are over-interpretations of their data. For example, the authors claim that they have found a role for polymerization of CARD, but they do not show any data that the foci they are observing in this study are due to polymerization of the CARD domain and not another feature of the domain. Another major deficit in this paper is the exclusive use of one model system (overexpression in 4D6 cells). The authors should confine their interpretation to the particular experimental model they are using. The authors do not provide evidence that Aire forms these foci in a physiologically relevant context. The authors also do not show that the nuclear concentration of Aire or the domain deletions tested in their model system are equivalent to nuclear concentrations of endogenous Aire. For these reasons and the specific comments provided below, the manuscript is not suitable for publication in Nature. With appropriate revisions, this manuscript might be more suitable for a specialized journal.

> We thank the reviewer for the critiques. However, we respectfully disagree with the reviewer on all five points. We acknowledge that there may have been shortcomings in the clarity of our original manuscript, and we have made every effort to address these concerns in the revised manuscript.

First, Aire expression is known to be largely limited to thymic epithelial cells (mTECs), which 4D6 is derived from. Second, Aire's nuclear foci formation has been extensively reported in native mTECs (PMIDs 20085707, 18414681, 19487417, 17974569, 15964547), which we can recapitulate in our 4D6 cell line. Third, Aire expression in our 4D6 cell line is comparable to that in mTECs (moved from Fig S1A to Fig. 1A for emphasis). Note that Aire in mTECs is highly abundant, with its mRNA level comparable to that of ribosomal proteins, ACTB and GAPDH. Fourth, our 4D6 cellular system faithfully recapitulates many other features of Aire in mTECs, including Aire's enhancer localization (PMID 28135252; Figure 1B, S1A), Aire's stochastic nature of gene activation (PMIDs 26237550, 26237553, 25224068; Figure 1E, 1F, S2B), and loss-of-function phenotypes of APS-1 patient mutations (PMIDs 18414681, 10677297; Figure S1C). We believe that these features validate our 4D6 cellular system and make our mechanistic studies physiologically relevant. Finally, the role of CARD polymerization in Aire foci formation was extensively tested in our previous paper (PMID: 32242017). We now summarize our previous findings in the Introduction. A previous mutagenesis study showed that Aire's nuclear condensate formation closely correlates with the polymerization activity of its CARD41 and that the CARD can be functionally substituted with an orthogonal, chemically-inducible multimerizing domain41.

Specific comments

1) The authors frame their study by claiming that they have discovered a new mechanism for the assembly of the transcription machinery or cluster assembly. Polymerization/oligomerization of transcription factors and other transcription regulators have been known to play a role in the assembly of the transcriptional machinery (PMIDs: 28076810, 23812588, 18406148, 27009358, 27097556). Polymerization/oligomerization domains have been shown to be required for the formation of condensates in cells or phase separation in vitro (PMIDs: 33692345, 32302570, 22579281, 29195049)

> We apologize for the lack of clarity in our writing, which might have led to the misinterpretation of our study's novelty. In this manuscript, we do not claim to be the first to report Aire's polymerization capability or Aire as the first polymerizing TR. The novelty of our manuscript lies instead in elucidating the multi-layered regulatory mechanism that governs Aire's polymerization. We also demonstrate for the first time that such regulatory mechanism not only exists but is required for Aire's transcriptional activity. The fact that there are other polymerizing TFs/TRs besides Aire, we believe, makes our findings more impactful.

2) The authors make claims that focus formation and transcription activity of Aire depends on CARD polymerization, but only show domain deletions. It is formally possible that phenotypes observed with deletion of the CARD domain are not due to loss of polymerization. There is no direct evidence that the foci observed in cells are due to polymerization of Aire.

> As stated above, this issue was extensively addressed in our previous study (PMID 32242017). We showed a close correlation between CARD polymerization in vitro and Aire foci formation in cells. We also showed that a chemically-inducible multimerization domain can substitute CARD and restore foci formation and transcriptional activity only in the presence of the chemical multimerizer, but not in its

absence. We also showed a simple dimerization or tetramerization is not sufficient, and instead, multimerization or polymerization is required to form condensates and transcriptionally active. Our extensive structure-activity-relationship analyses altogether showed that CARD polymerization is directly responsible for Aire foci formation in cells. We now include this description in the introduction.

3) There are many hits in the CRISPR screen and GST pulldown data, why do the authors focus on p300. Is this the only protein that is found by both methods? Do the other results make sense in the context of CTT function?

> We now include the comprehensive results from the CRISPR and mass-spec screens in new Table S3 and S4. CBP/p300 were the most significant hits that were identified in both screens. There were no other hits that were obviously linked to CBP/p300 functions, but we are actively following up on our screens to address this question.

4) It is well documented that acidic activation domains interact with Taz2 of p300 (PMID: 25753752). Many other proteins interact with Taz2. What is the affinity of the CTT-TAZ2 interaction? How does this compare to all the other TAZ2-mediated interactions from other transactivation domains?

> We thank the reviewer for this suggestion. We performed isothermal titration calorimetry (ITC) experiments on Aire CTT with TAZ2, which showed that CTT's affinity for TAZ2 (260 nM, new Fig. 3F) is comparable to those of other well-characterized activation domains (for example, p53 binds to TAZ2 at Kd of $\sim 1 \mu\text{M}$; PMID: 25753752, whereas β -catenin binds TAZ2 at Kd1 of 160 nM and Kd2 of $7 \mu\text{M}$; PMID: 36963539). Note that ITC also detected a second, low-affinity binding site/mode for AireCTT-TAZ2 (with Kd of $39 \mu\text{M}$), which likely has a less important role than the high-affinity site/mode.

5) The authors claim that PHD1 deletion data "explains the requirement for CTT in Aire foci formation", but there is no data that explains this requirement.

> We apologize for the lack of clarity. The reviewer is probably referring to the section with the subheading "Requirement for CTT in Aire polymerization stems from PHD1-mediated suppression.", which describes Fig 5. In the earlier section (related to Fig 4), we showed that CTT is required for Aire foci formation. Fig 5 shows that when Aire lacks PHD1, CTT requirement is no longer present. That is "PHD1+CTT can form foci, whereas "CTT cannot (Fig. 5A). The rest of the figure also supports the notion that CTT is no longer needed for Aire foci formation per se, when Aire lacks PHD1. Furthermore, we demonstrated that PHD1 is the domain that suppresses CARD-mediated foci formation. These results thus support our conclusion that "PHD1 is the suppressor domain for CARD polymerization and the requirement for CTT in Aire foci formation stems solely from PHD1-mediated suppression." We revised the subheading and text to further clarify this point.

6) It is confusing to understand why the authors sometimes use "foci" and other times use "hub". The authors also use the word "hub" to refer to multiple distinct concepts. By hub, do they mean the abstract concept of a signaling hub, the 3D organization of multiple genomic regions, or some other definition? The term hub is used to mean different things by different fields. It might be useful to come up with more precise language and avoid using hub or use it in a specifically defined way.

> We agree with the reviewer that our inconsistent terminology is confusing. As suggested by another reviewer, we have removed the term "hub" except when referring to multiple genetic loci coming in contact with a given Aire condensate. We have restricted the use of "foci" to describing RNA-FISH foci. We now use "condensates" throughout the entire manuscript.

7) The authors also claim that what they have found is a "departure from the conventional TAD functions" when what they present is the most conventional understanding of what an activation domain does, bind a co-activator. It is unclear what novelty the authors are trying to invoke here. There are now multiple examples of disorder-region mediated genomic targeting (PMID: 32553192, 31563432, 25303530, 37788668)

> We again apologize for the poor writing. We agree that there are other examples where a TAD is used to interact with coactivators or for genomic targeting. However, we believe that Aire is the first example, to the best of our knowledge, wherein TAD-mediated genomic targeting is used for TR condensate formation. This, together with PHD1-mediated suppression of TR condensate formation, ensures Aire forms condensate at the right genomic loci to activate its target genes. Our results in this manuscript thus reveal the multi-layered regulatory mechanism for transcriptional condensate formation and its functional importance, which we believe is novel.

Decision Letter:

Our ref: NI-A37665A

6th Jun 2024

Dear Sun,

Thank you for submitting your revised manuscript "Mechanism for controlled assembly of transcriptional condensates by Aire" (NI-A37665A). It has now been seen by the original referees and their comments are below. The reviewers find that the paper has improved in revision, and therefore we'll be happy in principle to publish it in Nature Immunology, pending minor revisions to satisfy the referees' final requests and to comply with our editorial and formatting guidelines.

We will now perform detailed checks on your paper and will send you a checklist detailing our editorial and formatting requirements in about a week. Please do not upload the final materials and make any revisions until you receive this additional information from us.

If you had not uploaded a Word file for the current version of the manuscript, we will need one before beginning the editing process; please email that to immunology@us.nature.com at your earliest convenience.

Thank you again for your interest in Nature Immunology Please do not hesitate to contact me if you have any questions.

Kind regards,

Laurie

Laurie A. Dempsey, Ph.D.
Senior Editor
Nature Immunology
l.dempsey@us.nature.com
ORCID: 0000-0002-3304-796X

Reviewer #1 (Remarks to the Author):

This is a MS from Houh et al. that attempts to further define the molecular details of how the AIRE transcription factor/regulator controls gene expression. The paper examines this process in a transfected cell model (4D6 cells) for many of the experiments and identify several domains that link AIRE's workings to nuclear condensates and transcriptional activation. Here they find that the C-terminal domain helps guide AIRE to enhancers that are occupied by CBP/p300 and also link this to the formation of nuclear condensates. They even perform an unbiased CRISPR screen to show CTT interactions with this complex and map out domains within CTT that define the interaction. They further identify that the PHD1 domain of AIRE which is a known H3K4Me0 binder negatively regulates nuclear condensate formation. The authors have also performed many experiments to address concerns raised in the original review of the MS at Nature.

Overall, these are some very interesting insights into what has been an enigmatic protein for the details of how it operates. The current experiments are very well done and the data here is convincing for the interactions that have been mapped out. The major limitation of the MS remains showing that this model is actually directly relevant to the in vivo situation with mTEC's (mouse or human) and connecting this to immune tolerance and mTEC activity.

The authors claim that their 4D6 Dox inducible AIRE system recapitulates how it works in mTEC's but here I found the data not to be convincing for directly correlating the same genes that it induces in vivo. Yes, AIRE does regulate gene expression in these cells and yes it forms foci and so on but I don't see convincing data that it robustly recapitulates the features of how it works in true mTEC's.

Furthermore, the MS also falls short in taking what has been found here and translating it to the broad body of work that others have performed or could be performed directly on mTEC's given their findings. For example, could one examine CBP/p300 occupied genes in Aire knockout mTEC's? Wouldn't these be enriched in genes that AIRE is about to turn on given their data? Do 4D6 cells further differentiate into mimetic cell types after AIRE induction? Why not? etc.

Despite these kind of limitations, this is a tour de force of molecular biology that provides some new insights into how AIRE potentially works and carries out its function and would be a great MS for a molecular biology journal given the in vivo limitations.

Reviewer #2 (Remarks to the Author):

The manuscript "Mechanism for controlled assembly of transcriptional condensates by Aire" is a revised version of a previous submission.

The manuscript reports insights on the mechanism of condensate formation by the transcriptional regulator Aire. The authors performed mutagenesis, imaging, genomics and biochemical assays, and found that various domains in Aire promote whereas other domains inhibit condensate formation, and Aire condensates are associated with transcriptional activity.

In the previous submission I commented on i) better articulation and support for the main model of the authors, ii) improving technical controls of imaging experiments, in particular the quantification of protein levels, and iii) terminology.

The authors have significantly improved the manuscript in all three areas in the revised version.

Overall, the authors make a compelling case in the revised version, and present substantial data that appear of high quality. The domain-balance model will be useful for investigators studying biomolecular condensates, and the results on the roles of Aire condensates in gene activity will be useful for investigators of immune cells.

I recommend acceptance for publication.

Denes Hnisz

Reviewer #3 (Remarks to the Author):

In this study, the authors dissect the molecular mechanisms underlying Aire regulation of transcription, performing mutagenesis to disrupt distinct regions of the Aire protein. The authors show that Aire formation of nuclear condensates is dependent on the CTT domain and interaction with CBP/300 and that the PHD1 Aire domain affects Aire genomic localization but not polymerization.

Mechanisms underlying Aire regulation of transcription remain highly debated and topical. This study utilizes several elegant and complementary approaches to address the molecular mechanisms underlying Aire regulation of transcription. Overall, this study represents an important advance in understanding Aire biology.

In the revised manuscript the authors have addressed most of the points raised by the reviewers, conducting additional experiments and analyses which add further rigor and, overall, the results support the main findings and conclusions. I am happy with the use of the 4D6 human epithelial cell line to investigate thymic Aire biology. In the absence of additional mouse models to study Aire mutants in physiological setting, 4D6 recapitulates many aspects of Aire-mTEC biology.

While questions remain over some aspects of the model, most notably why mutant PHD1 leads to loss of Aire chromatin binding, there are sufficient advances to make this paper highly significant and the findings will be useful to investigators studying Aire regulation of tissue specific antigens.

Author Rebuttal letter:

Reviewer #1:

Remarks to the Author:

This is a MS from Houh et al. that attempts to further define the molecular details of how the AIRE transcription factor/regulator controls gene expression. The paper examines this process in a transfected cell model (4D6 cells) for many of the experiments and identify several domains that link AIRE's workings to nuclear condensates and transcriptional activation. Here they find that the C-terminal domain helps guide AIRE to enhancers that are occupied by CBP/p300 and also link this to the formation of nuclear condensates. They even perform an unbiased CRISPR screen to show CTT interactions with this complex and map out domains within CTT that define the interaction. They further identify that the PHD1 domain of AIRE which is a known H3K4Me0 binder negatively regulates nuclear condensate formation. The authors have also performed many experiments to address concerns raised in the original review of the MS at Nature.

Overall, these are some very interesting insights into what has been an enigmatic protein for the details of how it operates. The current experiments are very well done and the data here is convincing for the interactions that have been mapped out. The major limitation of the MS remains showing that this model is actually directly relevant to the in vivo situation with mTEC's (mouse or human) and connecting this to immune tolerance and mTEC

activity.

The authors claim that their 4D6 Dox inducible AIRE system recapitulates how it works in mTECs but here I found the data not to be convincing for directly correlating the same genes that it induces in vivo. Yes, AIRE does regulate gene expression in these cells and yes it forms foci and so on but I don't see convincing data that it robustly recapitulates the features of how it works in true mTECs.

Furthermore, the MS also falls short in taking what has been found here and translating it to the broad body of work that others have performed or could be performed directly on mTECs given their findings. For example, could one examine CBP/p300 occupied genes in Aire knockout mTECs? Wouldn't these be enriched in genes that AIRE is about to turn on given their data? Do 4D6 cells further differentiate into mimetic cell types after AIRE induction? Why not? etc.

Despite these kind of limitations, this is a tour de force of molecular biology that provides some new insights into how AIRE potentially works and carries out its function and would be a great MS for a molecular biology journal given the in vivo limitations.

We thank you for the comments and we agree with the limitations of this study. To address this, we have modified the text within the Discussion:

Questions arise as to how Aire's role in consolidating and potentiating already active enhancers contributes to its known biological function of inducing PTAs, which are silent without Aire and are not strongly occupied by Aire. We suspect that Aire's actions of forming transcriptional hubs (as described in this paper) are mechanistically distinct from its effect on PTAs. That is, Aire may access PTAs, either directly or indirectly, through mechanisms independent of CBP/p300, but dependent on other factors, such as Z-DNA or DNA breaks at PTA promoters⁵³, or other tissue-specific transcription factors^{14,15}. This process may bring inactive PTA loci to Aire hubs by chance, thereby leading to the stochastic activation of PTAs⁵⁴. Alternatively, Aire's ability to bolster mTEC enhancers may affect the differentiation of Aire-positive mTEC into mimetic cells, thereby indirectly promoting PTA expression^{14,15}. More studies are needed to understand the precise functions of Aire in mTEC development and PTA expression. In summary, our mechanistic studies lay the groundwork for investigating how Aire utilizes controlled polymerization to establish central tolerance in T cell immunity.

Version 2:

Decision Letter:

In reply please quote: NI-A37665B

Dear Sun,

I am delighted to accept your manuscript entitled "Mechanism for controlled assembly of transcriptional condensates by Aire" for publication in an upcoming issue of *Nature Immunology*.

Over the next few weeks, your paper will be copyedited to ensure that it conforms to *Nature Immunology* style. Once your paper is typeset, you will receive an email with a link to choose the appropriate publishing options for your paper and our Author Services team will be in touch regarding any additional information that may be required.

Please note that *Nature Immunology* is a Transformative Journal (TJ). Authors may publish their research with us through the

traditional subscription access route or make their paper immediately open access through payment of an article-processing charge (APC). Authors will not be required to make a final decision about access to their article until it has been accepted. [Find out more about Transformational Journals](https://www.springernature.com/gp/open-research/transformational-journals).

Your paper will be published online soon after we receive your corrections and will appear in print in the next available issue.

Also, if you have any spectacular or outstanding figures or graphics associated with your manuscript - though not necessarily included with your submission - we'd be delighted to consider them as candidates for our cover. Simply send an electronic version (accompanied by a hard copy) to us with a possible cover caption enclosed.

If you have not already done so, we strongly recommend that you upload the step-by-step protocols used in this manuscript to protocols.io. protocols.io is an open online resource that allows researchers to share their detailed experimental know-how. All uploaded protocols are made freely available and are assigned DOIs for ease of citation. Protocols can be linked to any publications in which they are used and will be linked to from your article. You can also establish a dedicated workspace to collect all your lab Protocols. By uploading your Protocols to protocols.io, you are enabling researchers to more readily reproduce or adapt the methodology you use, as well as increasing the visibility of your protocols and papers. Upload your Protocols at <https://protocols.io>. Further information can be found at <https://www.protocols.io/help/publish-articles>.

Please note that we encourage the authors to self-archive their manuscript (the accepted version before copy editing) in their institutional repository, and in their funders' archives, six months after publication. Nature Portfolio recognizes the efforts of funding bodies to increase access of the research they fund, and strongly encourages authors to participate in such efforts. For information about our editorial policy, including license agreement and author copyright, please visit www.nature.com/ni/about/ed_policies/index.html

Kind regards,

Laurie

Laurie A. Dempsey, Ph.D.
Senior Editor
Nature Immunology
l.dempsey@us.nature.com
ORCID: 0000-0002-3304-796X

Click here if you would like to recommend Nature Immunology to your librarian
<http://www.nature.com/subscriptions/recommend.html#forms>

** Visit the Springer Nature Editorial and Publishing website at http://editorial-jobs.springernature.com?utm_source=ejP_NImm_email&utm_medium=ejP_NImm_email&utm_campaign=ejp_NImm for more information about our career opportunities. If you have any questions please click [here](mailto:editorial.publishing.jobs@springernature.com).
